# Average versus high surface ozone levels over the continental U.S.A.: Model bias, background influences, and interannual variability

Jean J. Guo[1], Arlene M. Fiore[1], Lee T. Murray[2,3,4], Daniel A. Jaffe[5], Jordan L. Schnell[6,7], Charles T. (Tom) Moore[8], George Milly[2]

[1]Department of Earth and Environmental Sciences and Lamont-Doherty Earth Observatory of Columbia University, Palisades, NY, U.S.A.
[2]Lamont-Doherty Earth Observatory of Columbia University, Palisades, NY, U.S.A.
[3]NASA Goddard Institute for Space Studies, New York, NY USA
[4]Now at: Department of Earth and Environmental Sciences, University of Rochester, Rochester, NY, U.S.A.
[5]University of Washington, School of STEM, Bothell, WA and Department of Atmospheric Science, Seattle, WA, U.S.A.
[6]NOAA Geophysical Fluid Dynamics Laboratory, Atmospheric and Oceanic Sciences, Princeton University, Princeton, NJ, U.S.A.
[7]Now at: Department of Earth and Planetary Sciences, Northwestern University, Chicago, IL, U.S.A.
[8]WESTAR and WRAP, Fort Collins, CO, U.S.A.

*Correspondence to*: Jean J. Guo (jean.j.guo@columbia.edu)

**Abstract.** U.S. background ozone ($O_3$) includes $O_3$ produced from anthropogenic $O_3$ precursors emitted outside of the U.S.A., from global methane, and from any natural sources. Using a suite of sensitivity simulations in the GEOS-Chem global chemistry-transport model, we estimate the influence from individual background versus U.S. anthropogenic sources on total surface $O_3$ over ten continental U.S. regions from 2004-2012. Evaluation with observations reveals model biases of +0-19 ppb in seasonal mean maximum daily 8-hour average (MDA8) $O_3$, highest in summer over the eastern U.S.A. Simulated high-$O_3$ events cluster too late in the season. We link these model biases to excessive regional $O_3$ production (e.g., U.S. anthropogenic, biogenic volatile organic compounds (BVOC), and soil $NO_x$, emissions), or coincident missing sinks. On the ten highest observed $O_3$ days during summer ($O_3$_top10obs_JJA), U.S. anthropogenic emissions enhance $O_3$ by 5-11 ppb and by less than 2 ppb in the eastern versus western U.S.A. The $O_3$ enhancement from BVOC emissions during summer is 1-7 ppb higher on $O_3$_top10obs_JJA days than on average days, while intercontinental pollution is up to 2 ppb higher on average versus on $O_3$_top10obs_JJA days. During the summers of 2004-2012, monthly regional mean U.S. background $O_3$ MDA8 levels vary by up to 15 ppb from year to year. Observed and simulated summertime total surface $O_3$ levels on $O_3$_top10obs_JJA days decline by 3 ppb (averaged over all regions) from 2004-2006 to 2010-2012, reflecting rising U.S. background (+2 ppb) and declining U.S. anthropogenic $O_3$ emissions (-6 ppb) in the model. The model attributes interannual variability in U.S. background $O_3$ on $O_3$_top10obs days to natural sources, not international pollution transport. We find that a three-year

averaging period is not long enough to eliminate interannual variability in background $O_3$ on the highest observed $O_3$ days.

## 1    Introduction

In the United States, ozone ($O_3$) is regulated as a criteria pollutant under the National Ambient Air Quality Standard (NAAQS). The current NAAQS for ground-level $O_3$, set in October 2015, states that the 4th-highest daily

maximum 8-hour average (MDA8) $O_3$, averaged across three consecutive years, cannot be 71 ppb or higher (U.S. Environmental Protection Agency, 2015). The three-year average is nominally intended to smooth out fluctuations in $O_3$ levels resulting from natural variability in meteorology within the timing constraints of the federal Clean Air Act for air quality planning. As even one ppb of excess $O_3$ may be enough to push a county out of NAAQS attainment, it is relevant to understand which sources influence the severity and timing of the highest $O_3$ events.

Since measured $O_3$ does not retain a signature of the source from which it was produced, estimates of background $O_3$ rely on models, ideally evaluated closely with observations, to build confidence in the model capability for source attribution. Here we apply a global chemistry-transport model alongside $O_3$ observations to examine which sources are influencing average versus high-$O_3$ events, and the extent to which they vary from year-to-year.

As U.S. anthropogenic emissions of $O_3$ precursors decline, the relative importance of "U.S. background" to

total surface $O_3$ rises. U.S. background $O_3$ is defined here as the $O_3$ levels that would exist in the absence of U.S. anthropogenic emissions of $O_3$ precursors, nitrogen oxide ($NO_x$) and non-methane volatile organic compound (NMVOC). U.S. background $O_3$ thus includes naturally occurring $O_3$ as well as $O_3$ produced from global methane (including U.S. anthropogenic emissions) and from $O_3$ precursor emissions outside of the U.S.A. Jaffe et al. (2018) review the current understanding on U.S. background $O_3$ from models and observations, and its relevance to air quality

standard setting and implementation. Previous studies estimating background $O_3$ over the United States found that background sources of $O_3$, including stratospheric $O_3$ intrusions (Lin et al., 2012, 2015a), increasing Asian anthropogenic emissions (Lin et al., 2015b), and more frequent wildfires in summer (Abatzoglou and Williams, 2016; Jaffe, 2011; Yang et al., 2015), may present challenges to obtaining the $O_3$ standard, especially since regional emission controls may be offset by a warming climate (Fiore et al., 2015). At high-altitude sites in the western U.S.A. (WUS)

in spring, the influence from stratospheric intrusions and foreign transport, combined with relatively deep planetary boundary layers, can lead to high background $O_3$ events (Fiore et al., 2002; Zhang et al., 2011). Lin et al. (2017)

investigated surface $O_3$ trends over the U.S.A. from 1980-2014 with the GFDL AM3 model and found that emissions controls decreased the 95[th] percentile summer $O_3$ values in the eastern U.S.A. (EUS) by 5-10 ppb over 1988-2014, but rising Asian emissions increased by 2-8 ppb at individual sites in the WUS over the period (Lin et al., 2016).

Earlier work in the GEOS-Chem model analyzing background $O_3$ during a single meteorological year noted a tendency for the model to underestimate springtime $O_3$ at high-altitude WUS sites but overestimate summertime $O_3$ over the EUS (e.g. Fiore et al., 2002, 2003; Wang et al., 2009; Zhang et al., 2011, 2014). Identifying the extent to which these biases reflect poor representation of U.S. anthropogenic versus background $O_3$ sources is relevant for assessing uncertainties in estimates of background $O_3$ on days when the $O_3$ NAAQS is exceeded. We build upon prior studies by analyzing MDA8 $O_3$ measurements and 9-year model simulations spanning 2004-2012 from the GEOS-Chem 3D global chemistry-transport model (CTM). A suite of sensitivity simulations in which different emissions of $O_3$ precursors are perturbed allows us to identify which sources are contributing the most to observed high-$O_3$ days and on the days with the highest model bias. We assess here whether biases in the model reflect problems in the modeled transported background versus $O_3$ produced within the U.S. from both background and anthropogenic sources. In addition, the availability of these simulations for 2004-2012 allow us to investigate the year-to-year variability in background sources and the extent to which this variability is relevant for observed high events, and therefore, potentially to attaining the $O_3$ NAAQS. Though coarse resolution global models such as GEOS-Chem will mix emissions into the same grid cell that may remain separate in the real atmosphere, a global model is necessary to quantify background $O_3$ transported intercontinentally, including that produced via oxidation of methane. We estimate the influence from various individual background sources on $O_3$ concentrations and the interannual variability in background $O_3$ levels with a focus on the highest 10 events in each of the 10 U.S. EPA regions during each summer (JJA) or year. We aim to answer the following questions: (1) Which sources exert the strongest influence on $O_3$ on the ten days with the highest model biases against observations? (2) Which background sources influence total $O_3$ the most on average versus the 10 highest $O_3$ days? (3) Which sources influence the interannual variability of $O_3$ in each region on average versus the 10 highest $O_3$ days?

## 2    Observations and model simulations

### 2.1  Observations

We use observed 2004-2012 MDA8 $O_3$ data from the EPA Air Quality System (AQS) network of urban, suburban, and rural monitoring sites, the Clean Air Status and Trends Network (CASTNet), and the Mount Bachelor

90 Observatory

(https://digital.lib.washington.edu/researchworks/browse?type=subject&value=Mt.+Bachelor+Observatory) in

Oregon. MDA8 $O_3$ values for the AQS sites were downloaded from

http://aqsdr1.epa.gov/aqsweb/aqstmp/airdata/download_files.html#Daily (2004-2012 data last updated June 28,

2013). This dataset includes 1644 total sites from the contiguous U.S.A. from 2004-2012 with 1207 to 1333 sites

95 collecting data each year (U.S. Environmental Protection Agency, 2014) (Supplemental Table 1).

The CASTNet (ftp://ftp.epa.gov/castnet/data) $O_3$ monitoring sites are located in rural areas away from

emission sources and densely populated regions. CASTNet sites are designed to capture background $O_3$ levels and

characterize broad spatial and temporal $O_3$ trends. We calculate the MDA8 $O_3$ concentration from hourly values at

108 CASTNet sites with data between 2004-2012, requiring at least 18 hours of data per day for each MDA8 $O_3$

calculation.

The Mount Bachelor Observatory, established in 2004 by the University of Washington Jaffe Research

Group, is located 2.7 km above sea level on the summit of Mount Bachelor, an extinct volcano in the Cascade

Mountains of central Oregon. It provides an estimate of baseline $O_3$ levels over the West Coast of the United States.

Baseline $O_3$ is defined as the $O_3$ concentration at sites with negligible influence from local emissions (National

Research Council, 2010). Baseline $O_3$ is a measurable quantity and differs from background $O_3$ in that it contains

some influence from U.S. anthropogenic emissions that were not recently emitted but contributed to the global

background. This station represents variations in baseline $O_3$ concentrations (Baylon et al., 2016) and is analyzed as

a standalone site in Section 3.2 given the relevance of high-altitude measurements for downwind surface $O_3$

(Stauffer et al., 2017). We take all hourly $O_3$ concentrations from Mount Bachelor and calculate the MDA8 $O_3$

concentrations for 2004-2012. Daily averages are included only if at least 18 hours of data are available and monthly

averages require at least 20 days with valid 24-hour mean or MDA8 data. For our comparison to monthly average

$O_3$ at Mount Bachelor Observatory, we sample the model both at the level closest to 2.7 km and at the surface.

We use temperature data from a 0.5º x 0.5º resolution gridded dataset developed by Fan and van den Dool

(2008) using data from the Global Historical Climatology Network (GHCN) and the Climate Anomaly Monitoring

System (CAMS). GHCN Gridded V2 data was provided by the NOAA/OAR/ESRL PSD, Boulder, Colorado, USA

(https://www.esrl.noaa.gov/psd/). Each observational site is matched to the temperature model grid cell it falls in and

the average monthly temperature is computed by averaging across all the sites in each region.

In order to evaluate the GEOS-Chem model $O_3$ simulation (described below in Sect. 0) at a spatial scale comparable to the coarse horizontal resolution global grid (2° x 2.5°), we use an available 1° x 1° grid of surface

MDA8 $O_3$ measurements, interpolated from the AQS, CASTNet, and Canadian NAPS networks (Schnell and Prather, 2017). We degrade this 1° x 1° dataset to 2° x 2.5° to match the horizontal resolution of the GEOS-Chem simulations. As we did not archive three-dimensional high frequency data, all MDA8 $O_3$ values from the model are sampled at the lowest surface layer for comparison to observational sites.

**2.2   Analysis regions**

Each observational site in the EPA AQS and CASTNet datasets is linked to one of the 10 U.S. EPA air quality regions (Supplemental Figure 1) based on which state the site is in. The Mount Bachelor data were included with Region 10 (Pacific Northwest) sites even though it is not a regulatory monitor. Following Reidmiller et al. (2009), we select two regions, the Southeast (Region 4) and Mountains and Plains (Region 8), as representative regions for the EUS and WUS for illustration purposes in the main text. Figures for the other eight regions are

included in the supplement.

To find the daily mean $O_3$ concentration within each region, we first match each observational site to the model grid within which it falls. We then average across all sites in each region to obtain a regional mean MDA8 $O_3$ value in the observations and in the model. From the regionally averaged observed MDA8 $O_3$, we find: (1) the ten days with the highest observed $O_3$ during each year (hereafter, $O_3$_top10obs days), similar to the definition for extreme events used in Schnell et al. (2014), (2) the ten days with the highest $O_3$ observations during each season

(hereafter, $O_3$_top10obs_MAM, $O_3$_top10obs_JJA, and $O_3$_top10obs_SON), and (3) the 4th highest MDA8 $O_3$ within each year. In addition, we sample the model to find the ten days each year with the highest positive biases. There is at most a 2-6 day overlap between the top 10 $O_3$_Base days and the top 10 most biased days in 2004-2012 across all regions, but during most years, the overlap is around 0-2 days. We restrict our analysis to examining the

top 10 observed $O_3$ days as these days are most relevant from a policy perspective. We use $O_3$_top10obs as our primary metric, however, instead of the policy-relevant 4th highest $O_3$ because the model bias is typically lower on $O_3$_top10obs days (Supplemental Figure 2 versus Supplemental Figure 3). On the days when the 4th highest values occur, the model bias is generally more strongly negative in the west and South Central regions and more strongly positive in the Midwest than on $O_3$_top10obs days (Supplemental Figure 2, Supplemental Figure 3). In addition,

while the model rarely captures the exact day of the 4th highest MDA8 $O_3$ event, there is a 3-4 day overlap on average between the O$_3$_top10obs days and the highest 10 MDA8 $O_3$ days in the model. This overlap is similar to the 3 and 6 day overlap Jaffe et al. (2018) found in their regional models for May 1st to September 29th, 2011.

### 2.3 GEOS-Chem model simulations

We use the GEOS-Chem v9_02 global 3D chemical transport model (CTM) (http://www.geos-chem.org)
simulations driven by Modern-Era Retrospective analysis for Research and Applications (MERRA) reanalysis meteorology from the NASA Global Modeling and Assimilation Office for 2004-2012 (Rienecker et al., 2011). The MERRA reanalysis is available at 1/2° by 2/3° horizontal resolution, which we degrade here to 2° by 2.5° horizontal resolution. MERRA meteorology captures summer mean surface temperatures to within 1-2 K across U.S. regions and precipitation to within 0.5 mm d$^{-1}$ except for over the Northern Great Plains where a positive bias exceeds 1 mm
d$^{-1}$, but the variance in summer mean precipitation is lower than observed in some regions (Bosilovich, 2013). While interannual variability in cloudiness observed at weather stations is largely captured by MERRA, the reanalysis generally underestimates cloud cover and thus overestimates observed downward surface shortwave fluxes (Free et al., 2016). Methane surface concentrations are prescribed each month using spatially interpolated surface distributions from NOAA Global Monitoring Division flash data. We use the standard v9_02 chemical mechanism
which includes recycling of isoprene nitrates (Mao et al., 2013) in contrast to the mechanisms used in earlier versions of GEOS-Chem (e.g., Zhang et al., 2014 as discussed in Fiore et al., 2014). Anthropogenic base emissions are from the Emission Database for Global Atmospheric Research (EDGAR) version 3.2-FT2000 inventory (Olivier et al., 2005) for inorganic compounds and the REanalysis of the TROpospheric chemical composition (RETRO) inventory (Hu et al., 2015; Schultz, 2007) for organic compounds. Inorganic emissions are overwritten by regional inventories
for the U.S. (EPA National Emissions Inventory 2005), Canada (Criteria Air Contaminants), Mexico (Big Bend Regional Aerosol and Visibility Observational study; Kuhns and Green, 2003), Europe (European Monitoring and Evaluation Programme; Auvray and Bey, 2005), and South and East Asia (Streets et al., 2006). Separate global inventories are used for ammonia (Bouwman et al., 1997), black carbon (Bond et al., 2007; Leibensperger et al., 2012), and ethane (Xiao et al., 2008). Anthropogenic surface emissions have diurnal and monthly variability, some
with additional weekly cycles, and are scaled each year on the basis of economic data and estimates provided by individual countries, where available (van Donkelaar et al., 2008). The model does not include daily variations in U.S. anthropogenic emissions associated with higher electricity demand on hotter days (e.g., Abel et al., 2017).

Aircraft emissions are from the Aviation Emissions Inventory Code (AEIC) inventory (Stettler et al., 2011) and shipping emissions are from International Comprehensive Ocean-Atmosphere Data Set (ICOADS; Lee et al., 2011;

Wang et al., 2008). Biomass burning emissions follow the interannually-varying monthly Global Fire Emissions Database version 3 (GFED3) inventory driven by satellite observations of fire activity (Giglio et al., 2010; Van Der Werf et al., 2010). Biofuel emissions are constant (Yevich and Logan, 2003). Biogenic VOC emissions from terrestrial plants follow the Model of Emissions of Gases and Aerosols from Nature (MEGAN) scheme version 2.1 (Guenther et al., 2012) and vary with meteorology (Barkley et al., 2011). Global and U.S. emissions are 29.5 Tg N

$yr^{-1}$ and 5.2 Tg N $yr^{-1}$ respectively, for anthropogenic $NO_x$ emissions (including biofuels); 4.2 Tg N $yr^{-1}$ and 0.1 Tg N $yr^{-1}$ for biomass burning; 8.7 Tg N $yr^{-1}$ and 0.9 Tg N $yr^{-1}$ for soil NOx; 6.7 Tg N $yr^{-1}$ and 1.0 Tg N $yr^{-1}$ for lightning NOx; 466.1 Tg C $yr^{-1}$ and 20.6 Tg C $yr^{-1}$ for isoprene emissions. Emissions for $NO_x$ sources and isoprene are provided globally and within the U.S.A. for each year in Supplemental Table 3.

We first perform a base simulation ($O_3\_Base$) with all emissions turned on for 2003-2012. We conduct a

parallel suite of sensitivity simulations, in which selected sources are removed. In all simulations, we discard 2003 from our analysis as initialization. Our first set of sensitivity simulations estimates three different "background" definitions: (1) "North American Background" (denoted $O_3\_NAB$) in which anthropogenic emissions within Canada, Mexico, and the U.S.A. are set to zero, but methane surface abundances are kept at present-day values; (2) "U.S. background" ($O_3\_USB$), which is similar to $O_3\_NAB$ except only U.S. anthropogenic emissions are set to zero; (3)

"Natural background" ($O_3\_NAT$), in which all anthropogenic emissions have been removed globally and methane is prescribed at preindustrial levels. We estimate Canadian and Mexican influence ($O_3\_CA+MX$) on U.S. $O_3$ by subtracting $O_3\_NAB$ from $O_3\_USB$; the influence from intercontinental pollution transport plus global methane ($O_3\_ICT+CH_4$) is estimated by subtracting $O_3\_NAT$ from $O_3\_NAB$. A second set of sensitivity simulations enable us to estimate the contribution of individual "background" sources to total simulated surface $O_3$ by subtracting a

simulation with that source shut off from the $O_3\_Base$ simulation: (1) $O_3\_NALNO_x$ by turning off North American lightning NOx; (2) $O_3\_SNO_x$ by zeroing out global soil NOx; (3) $O_3\_BVOC$ by zeroing out terrestrial biogenic VOC emissions (we also examine this "$O_3\_noBVOC$" simulation in Section 3.3); (4) $O_3\_BB$ by zeroing-out biomass burning emissions, as summarized in Table 1. Due to non-linearities in atmospheric photochemistry, these "zero-out" estimates of source contributions depend on the presence of all other precursor emissions at present-day levels

(e.g., the impact of BVOCs emissions is sensitive to the amount of anthropogenic $NO_x$ emissions in the Base

simulation). This set of model simulations does not directly isolate stratospheric $O_3$ or Asian influences. Previous work has shown that stratospheric $O_3$ can increase springtime $O_3$ levels by 17-40 ppb in the WUS when MDA8 $O_3$ levels are 70-85 ppb, and Asian emissions can contribute 8-15 ppb to MDA8 $O_3$ on days above 60 ppb (Lin et al., 2012, 2015a). Stratospheric and Asian influences are included in $O_3\_USB$; Asian influences are included in $O_3\_ICT+CH_4$; and $O_3\_NAT$ includes stratospheric $O_3$, biogenic emissions of $O_3$ precursors, wildfires, and lightning $NO_x$. As $O_3\_BVOC$ includes $O_3$ produced from biogenic VOC reacting with both natural and anthropogenic $NO_x$, $O_3\_USA$ and $O_3\_BVOC$ are not additive. $O_3\_BVOC$ thus contributes to both $O_3\_USA$ and $O_3\_USB$.

## 3   Model evaluation

### 3.1   MDA8 $O_3$ distributions

To evaluate the ability of our coarse resolution model to capture observed high-$O_3$ events, we compare the MDA8 $O_3$ averaged over each of the 10 EPA regions simulated by GEOS-Chem to the observations in two ways. In the first method, we use the (Schnell and Prather, 2017) gridded dataset degraded to the model resolution and sample the model directly at each of the degraded Schnell grid cells prior to calculating the regional average. In the second method, we sample the model grid cell containing each individual observational site (EPA AQS, CASTNet, and Mount Bachelor Observatory) prior to calculating the regional average. The model is biased positively with either method (Figure 1a, b), but the shape of the model distribution constructed with the latter approach (Figure 1b) better matches the observed distribution than that of the former (Figure 1a). Matching individual sites to the nearest model grid (Figure 1b) yields a better estimate of high-$O_3$ days; the model overestimates the percentage of days above 70 ppb by about three times when we match to individual measurement sites (3.14% of days are above 70 ppb in the observations versus 9.92% in model) but by about ten times in comparison to the re-gridded Schnell (2014) dataset (0.37% of days are above 70 ppb in the observations versus 3.91% in the re-gridded dataset).

Simulated seasonal mean MDA8 averaged over the full 2004-2012 period is higher than observed by 5-30 ppb (Figure 2a, b, c), with the largest biases typically occurring in the Northeast and Midwest. The model bias is highest in summer (JJA) (15-30 ppb at most sites), followed by fall (SON) (10-20 ppb) (Figure 2a, b, c). Recent work in a newer version of GEOS-Chem attributes some of the positive model bias in the EUS to excessive $NO_x$ emissions in the 2011 National Emission Inventory (NEI) (Travis et al., 2016), an inability of the model to resolve vertical mixing in the boundary layer, and a weak response to cloud cover (Travis et al., 2017). Travis et al. (2016)

find that the 3.5 Tg N $y^{-1}$ NEI 2011 estimate for U.S. fuel NOx emissions is too high and contributes to excessive

surface $O_3$. Our simulations include even higher U.S. fuel NOx emissions of 4.4 Tg N $y^{-1}$ during 2010-2012

(Supplemental Table 3), implying that some portion of the model $O_3$bias reflects excessively high anthropogenic

NOx emissions (Travis et al., 2016). The low bias in cloud cover in the MERRA meteorology and associated

overestimate in downward shortwave surface radiation (Free et al., 2016) may also contribute to excessive $O_3$

production in the model. The model is closest to the observations in spring, with a positive bias usually <10 ppb

over the eastern states and generally within ±5 ppb over most western sites (Figure 2a, b, c).

**3.2 Baseline $O_3$ at Mount Bachelor**

Mount Bachelor Observatory (MBO) regularly samples free tropospheric $O_3$ and is rarely influenced by

local anthropogenic emissions (Reidmiller et al., 2009). It is therefore, a valuable site for examining baseline $O_3$. In

Supplemental Figure 4, we compare the observed 24-hour and MDA8 $O_3$ concentrations at MBO for 2004-2012.

The observed $O_3$ concentrations vary from year to year, and by definition, MDA8 $O_3$ is a few ppb higher than the

24-hour mean mixing ratios. However, the seasonal pattern is similar across both metrics, with a springtime peak,

maximum in April, and a secondary summertime peak in July.

Figure 3 compares modeled and observed monthly mean 24-hour $O_3$ concentrations at the grid box that

contains Mount Bachelor. For the model, we examine $O_3\_Base$ and $O_3\_USB$ 24-hour average concentrations at 2.7

km, the height of the Mount Bachelor Observatory, as well as at the surface. It is important to note that the diurnal

variations on the mountain may not be well captured by the CTM, due to upslope (daytime)/downslope (nighttime)

flow. We focus on the 24-hour average because we only archived hourly $O_3$ fields from the model at the surface and

thus, do not have the MDA8 $O_3$ metric available at 2.7 km. The year-to-year variability is smaller in the model than

observed (narrower shaded range). In all months, the $O_3\_Base$ and $O_3\_USB$ values are higher by 9-14 ppb and 11-21

ppb, respectively, at 2.7 km than at the surface. The model captures the magnitude of the observed springtime peak

at 2.7 km, but summertime values are too high, with an overall peak in August. $O_3\_USB$ contributes a greater

fraction to $O_3\_Base$ at 2.7 km (92-94%) than at the surface (72-94%). The simulated seasonal cycle differs at the

surface, peaking in spring (March-April) and in September. In 2012, the observations show equivalent springtime

and summertime peaks, more similar to the modeled seasonal cycle. While the observations generally decline from

spring into summer, the model indicates an increase, leading to a substantial model overestimate during summer in

most years. This model bias occurs across much of the U.S.A. as we show below.

Our sensitivity simulations enable us to interpret the sources contributing to the simulated seasonal

distribution. The model indicates that at MBO, $O_3$_USB is the major component of $O_3$_Base, including during the

summertime overestimate. In turn, the model indicates that the seasonality of $O_3$_USB is largely driven by $O_3$_NAT,

which includes the influence from biogenic VOC and NOx, lightning $NO_x$, as well as stratospheric $O_3$. $O_3$_ICT+$CH_4$

contributes around 15 ppb at 2.7km and 5-10 ppb at the surface (Figure 3). The model does suggest a springtime

peak influence from $O_3$_ICT+$CH_4$ in the WUS, consistent with earlier work (e.g., Task Force on Hemispheric

Transport of Air Pollution, 2010). Even at this baseline site, the model indicates that $O_3$_USA enhances monthly

mean $O_3$ by at least a few ppb at 2.7 km; at the surface, the model simulates a seasonal cycle for $O_3$_USA that is

typical of photochemical production from regional precursor emissions. $O_3$_CA+MX is less than a few ppb at MBO

whether the model is sampled at 2.7 km or the surface (not shown).

**3.3 Magnitude and timing of high-$O_3$ events**

On $O_3$_top10obs days, the model biases are typically lower than on average days (Figure 2, Table 2; see

also year-by-year maps in Supplemental Figure 2). At some WUS sites, the model underestimates $O_3$ levels during

the highest events by 10-20 ppb. The model systematically underestimates $O_3$ in the Central Valley of California in

all three seasons, which we attribute to the inability of the coarse model resolution to resolve topographical gradients

and valley circulations (or stagnation) in this region which experiences some of the highest observed $O_3$ in the

nation.

We compare the MDA8 $O_3$ distributions in the observations versus the model ($O_3$_Base) during the 10

most biased days in each of the ten regions across the nine years (900 total events). These "most-biased" days in

the model tend to fall around the observed median (Figure 1c) during the warm season (June - October), with

almost 40% of the days falling in August alone (Supplemental Figure 5), and are 9-45 ppb higher than the

observations (circles in Supplemental Figure 6). We analyze the perturbation simulations (Table 1) to identify

which sources influence simulated $O_3$ most strongly on the "most-biased" days versus on average (i.e., all 365 or

366 days), which we assume are also likely the main drivers of the bias. In all regions, the largest sources on the

280    "most-biased" model days are $O_3$_USA (3-30 ppb higher MDA8 $O_3$ than on average with the exception of the

Pacific SW where $O_3$_USA is smaller than on average days), $O_3$_BVOC (by 1-15 ppb), and $O_3$_SNO$_x$ (by 1-10 ppb; Figure 4, Supplemental Figure 6). By contrast, $O_3$_ICT+CH$_4$ is up to a few ppb higher on average days than on the most-biased model days.

To explore possible drivers of model biases across the different seasons, we evaluate the timing of the highest ten events across each year in the $O_3$_Base, $O_3$_USB, and $O_3$_noBVOC (BVOCs shut off) simulations for each region (900 events). We bin these 900 events by month and calculate the percentage of the total events that fall within each month. Note that all the top ten days fall between March and October. The standard model ($O_3$_Base) underestimates the occurrence of high events early in the $O_3$ season (March-June) and overestimates them later in the season (July-September) (Supplemental Figure 7). While the model indicates that most top ten $O_3$ days fall between July-August (35% each), the observations show that May through August each contain around 15-25% with the maximum in June at 25%. Both $O_3$_noBVOC and $O_3$_USB shift the relative timing of the 10 highest $O_3$ events towards April and May compared to $O_3$_Base, but the shortage of high springtime $O_3$ events remains (Supplemental Figure 7). The lack of high events in spring may reflect in part poor representation of stratospheric $O_3$ intrusions at the coarse resolution of the CTM (Lin et al., 2012; Zhang et al., 2014), in addition to the role of U.S. anthropogenic and BVOC emissions in the temporal mis-match as indicated by the improvements to the timing that occur in the $O_3$_USB (U.S. anthropogenic emissions shut off) and $O_3$_noBVOC simulations. In addition to contributions from these sources, poor representation of $O_3$ sinks may contribute to the model biases. For example, Makar et al. (2017) suggest that failing to represent canopy turbulence and shading effects on photolysis can lead to high-$O_3$ biases in models.

**3.4 Interannual variability**

Supplemental Figure 8 shows the Pearson correlations coefficients (*r*) between monthly average observed and $O_3$_Base values from 2004-2012. In May, correlations are generally strong ($r \geq 0.9$) in the Mid-Atlantic and Southeast regions, but much lower ($r = 0.2$) in the New England region. This pattern may reflect shortcomings in representing the onset of BVOC emissions. In July, the regions flip, with lower correlations in the Southeast and higher correlations in New England. At some sites in the WUS, lower correlations occur during summer months, which may be tied to excessive influence from lightning NO$_x$ advected from Mexico (see also Zhang et al., 2011; 2014) or anomalous events such as wildfires that are not well captured by the model.

In general, correlations only average about $r = 0.2$ in the winter and early spring over much of the United States (Supplemental Figure 8); the drivers for these weak correlations may be connected to the model tendency to underestimate the occurrence of springtime high-$O_3$ events. From May to September, however, the months during which high-$O_3$ events are most likely to occur, the correlation between 2004-2012 observed and simulated $O_3$ monthly averages over much of the contiguous United States exceed $r = 0.7$ (Supplemental Figure 8). We conclude that the model broadly captures monthly variations from year-to-year during the warm season and can thus be applied to interpret the role of background sources in contributing to interannual variations during most of the high-$O_3$ season. We note that Clifton et al. (2017) found that the GEOS-Chem model does not capture interannual variability in deposition velocities observed at Harvard Forest, MA, but it is unclear to what extent this process would amplify or dampen interannual variability associated with changes in emissions.

**4    Influence of individual sources on average versus high-$O_3$ days**

In Table 2 and Table 3, we report the influence of the $O_3$ sources defined in Table 1 on average versus $O_3$_top10obs days separately for spring (MAM), summer (JJA), and fall (SON) (ten days from each of the nine simulation years for 900 events for each region and season). We also report the difference in source influences between average and $O_3$_top10obs days, which we interpret as the enhancement from that source relative to average conditions.

We first consider the average ranges in MDA8 $O_3$ contributed by the various sources. Both $O_3$_USA and $O_3$_USB tend to follow the seasonal cycle of $O_3$_Base, with highest abundances in summer. The model indicates that $O_3$_USB is 30-50 ppb (range over regions) during summer and highest over the WUS. $O_3$_USA is generally 20-30 ppb over the EUS in summer, but only 10-20 ppb over the WUS (Table 2). $O_3$_ICT+CH$_4$ averages 2-13 ppb over all regions and is highest in spring (8-13 ppb compared to 2-11 ppb in summer and 6-12 ppb in fall) (Table 3, Figure 5, Supplemental Figure 9). $O_3$_NALNO$_x$ has a relatively minor influence (at most 1.5 ppb) in all regions and seasons. The influence from $O_3$_CA+MX is generally less than a couple of ppb except in NY+NJ and New England where it can be as much as 4-7 ppb (Table 3, Supplemental Figure 9).

We interpret the "difference" lines in Table 2 and Table 3 as the enhancements from each source on high days in each season ($O_3$_top10obs_MAM, $O_3$_top10obs_JJA, $O_3$_top10obs_SON) relative to average conditions. Over all regions, $O_3$_BVOC and $O_3$_SNO$_x$ influence $O_3$_Base more on $O_3$_top10obs days (for all seasons) than on

average whereas $O_3\_ICT+CH_4$ is typically lower by up to 3 ppb on $O_3\_top10obs$ days (for all seasons) than on

average days (Table 2, Table 3, Figure 5, Supplemental Figure 9). $O_3\_USA$ is 8-11 ppb higher on $O_3\_top10obs\_JJA$

days versus average over the New England, NY+NJ, Mid-Atlantic, Midwest, and South Central regions, but only up

to 5 ppb higher over other regions (Table 2, Figure 5, Supplemental Figure 9). The model indicates an even stronger

anthropogenic enhancement (up to 19 ppb) on $O_3\_top10obs\_SON$ days in some EUS regions (Table 2). $O_3\_USB$ is

enhanced on $O_3\_top10obs\_JJA$ days by 2-12 ppb relative to average, with the smallest enhancements occurring in

the Mid-Atlantic, Southeast, and Midwest regions, and the largest enhancements occurring in the Pacific NW. In

contrast to all the other regions, $O_3\_USB$ is the dominant source enhancing $O_3\_top10days\_JJA$ over the Mountains

and Plains, Pacific NW, and Pacific SW regions (4-12 ppb for $O_3\_USB$ but < 5 ppb from either $O_3\_USA$ or

$O_3\_BVOC$). In line with earlier work reviewed by Jaffe et al. (2017), enhanced $O_3\_USA$ dominates

$O_3\_top10obs\_JJA$ days over much of the U.S.A., whereas in the WUS, $O_3\_USB$ enhancements exceed $O_3\_USA$

enhancements on $O_3\_top10days\_JJA$. $O_3\_BVOC$ enhances $O_3\_top10obs$ days (for all seasons) by up to 9 ppb, with

the influence often largest in fall (when $O_3$ formation is more sensitive to VOC; e.g., Jacob et al., 1995). We re-

emphasize that BVOCs contribute both to $O_3\_USA$ when reacting with anthropogenic $NO_x$ and to $O_3\_USB$ when

reacting with all other $NO_x$ sources. In contrast to the sources discussed above, $O_3\_ICT+CH_4$ influences average

days by up to a few ppb more than on $O_3\_top10obs$ days (for all seasons), with the largest differences between

average and high days occurring in EUS regions (1-3 ppb lower on $O_3\_top10obs$ days (for all seasons) in New

England, NY+NJ, Mid-Atlantic; Table 3, Figure 5, Supplemental Figure 9). $O_3\_NALNO_x$ is at most 2 ppb higher

than average on $O_3\_top10obs$ days. The $O_3\_CA+MX$ influence is roughly equivalent (generally to within a ppb) on

average versus $O_3\_top10obs$ days during all seasons.

**5   Interannual variability in the sources influencing high vs. average ground-level $O_3$**

      Despite its high mean bias and seasonal phase shift, the model does capture some of the observed

interannual variability in observed $O_3\_top10obs\_JJA$ MDA8 $O_3$ concentrations (Figure 6, Supplemental Figure 10; *r*

= 0.5 to ≥ 0.9). Comparing the 2004-2006 period with 2010-2012, both observed and simulated MDA8 $O_3$

concentrations on $O_3\_top10obs\_JJA$ days hold steady or decrease across all regions. This change reflects opposing

influences in the model: rising $O_3\_USB$ (by 2 ppb averaged over all regions) and declining $O_3\_USA$ concentrations

(by 6 ppb averaged over all regions) (Figure 6, Table 4, Supplemental Figure 10). We note that over the Pacific NW

there is a 4 ppb decrease in $O_3$_USB from 2004-2006 to 2010-2012. Over this period, temperatures generally warm over the EUS, but slightly cool in the WUS. Within the ten regions, the model captures the sign of the changes in MDA8 $O_3$ over this period but not the magnitude (Table 4). The model monthly mean temperatures in the model (from the MERRA reanalysis) closely match the observed GHCN+CAMS dataset (Supplemental Table 4). Table 4 shows that regions with $O_3$_USB increases generally experienced rising temperatures over this period, as the 2010-2012 period includes two of the warmest years on record. Figure 6 shows that $O_3$_NAT tracks with $O_3$_USB and temperature (dips in MDA8 $O_3$ occur during years with cooler temperatures (2008-2009) and increases in years with warmer temperatures (2011-2012), indicating that year-to-year variability in $O_3$_USB on $O_3$_top10obs_JJA days is primarily driven in the model by natural sources sensitive to meteorology rather than international $O_3$ transport (Figure 6, Supplemental Figure 10). Although 2012 was the hottest year on average between 2004-2012 (except in the Pacific NW where 2004 was warmer by about a degree), it was not the hottest summer in all regions.

We find that $O_3$_USB drives the interannual variability on $O_3$_top10obs_JJA days in the WUS ($r = 0.72$-$0.85$ for $O_3$_USB versus $O_3$_Base, whereas $r = 0.05$-$0.64$ for $O_3$_USA versus $O_3$_Base; Supplemental Table 5). In NY+NJ, the Southeast, Midwest, South Central, and Plains regions, $O_3$_USB and $O_3$_USA both contribute to the interannual variability on $O_3$_top10obs_JJA days ($r = 0.5$-$0.8$ for both $O_3$_USB and $O_3$_USA versus $O_3$_Base) while in New England and the Mid-Atlantic regions, $O_3$_USA drives the interannual variability more than $O_3$_USB ($r = 0.64$ and $0.72$ for $O_3$_USA versus $O_3$_Base but only $0.28$ and $0.54$, respectively, for $O_3$_USB versus $O_3$_Base; Supplemental Table 5).

Year-to-year variations in monthly average $O_3$_USB are relatively large, with 10-15 ppb differences between the highest and lowest $O_3$_USB years during the warmest months (Figure 7, Supplemental Figure 11). Seasonal variations also differ by region, especially during summer. For example, the western U.S. regions have a smooth seasonal cycle with $O_3$_USB concentrations rising from January to a peak in July and August, and then declining again. Interannual and seasonal variability in $O_3$_USB are generally greater in the Southeast than in the Mountains and Plains, and Plains regions (Figure 7, Supplemental Figure 11). Year-to-year variability in $O_3$_BVOC is smaller than $O_3$_USB, with a maximum range of about 10 ppb between the highest and lowest years during August (Figure 7, Supplemental Figure 12). $O_3$_SNOx ranges by a few ppb throughout the summer in the Southeast, and by up to 6 ppb over the Mountains and Plains in August (Figure 7, Supplemental Figure 13).

O$_3$_USA anomalies relative to the 2004-2012 average illustrate declining influence in all regions, with

negative anomalies after 2007 on both O$_3$_top10obs and average days (Figure 8, Supplemental Figure 14). This

finding is well established by earlier work demonstrating decreases in high-O$_3$ concentrations as a result of regional

NO$_x$ emissions reductions over the past few decades (Cooper et al., 2012, 2014a; Jaffe et al., 2018; Young et al.,

2017). O$_3$_BVOC is the main driver of the high and low O$_3$ anomalies (up to ±5 ppb on O$_3$_top10obs_JJA days)

from year-to-year (Figure 8, Supplemental Figure 15).

Specific events can affect O$_3$ in any given year. For example, in 2008, there were extensive fires across

much of California in May, June, and July. In 2008, the Pacific SW region that includes California, Nevada, and

Arizona, shows a positive anomaly in O$_3$_BB (> 1 ppb) on the O$_3$_top10obs days, stronger than during any other

year in that region (Supplemental Figure 15). If we restrict our analysis solely to Reno, NV, the anomaly for O$_3$_BB

was 7 ppb in July 2008 relative to the 2004-2012 July average (not shown). We emphasize that a single location can

be more strongly influenced by a specific source than the regional averages on which we have focused.

Currently, the U.S. EPA uses a 3-year averaging period of the 4[th]-highest MDA8 O$_3$ to assess compliance

with the O$_3$ NAAQS. We evaluate the extent to which this 3-year averaging period removes interannual variability

in meteorology (the grounds for the averaging) (Figure 9, Supplemental Figure 16). The observed range is

generally much smaller than the model estimate. We find that the 3-year average of the 4[th] highest day decreases

the range by 2-6 ppb and 5-18 ppb in the observations and O$_3$_Base respectively when compared to taking the 4[th]

highest day in any given year when we look across all regions (Table 5). However, the 3-year average of the 4[th]

highest day still ranges from 3-9 ppb and 2-11 ppb in the observations and O$_3$_Base, respectively, across all

regions (compared to 5-15 ppb and 10-36 ppb in the observations and O$_3$_Base on the 4th highest day in each

individual year). Thus, while averaging across the years decreases the spread, variability remains. In keeping with

our previous analysis of the O$_3$_top10obs days, we compare the spread of the 4[th] highest O$_3$ day in each of the three

years to the range of the O$_3$_top10obs days across each three year span; the 4[th] highest days can range almost as

widely as the O$_3$_top10obs days in some years, but in other years, are clustered closer together (Figure 9). Figure 9

shows that the range in O$_3$_top10obs days for O$_3$_Base generally correlates with O$_3$_UBS in the WUS, suggesting

that O$_3$_USB is the dominant influence on the high days there, but there is little correlation in the EUS. We

conclude that a three-year smoothing period is not long enough to eliminate entirely the interannual variability in

MDA8 $O_3$ levels, and in the WUS, this interannual variability tends to reflect variations in O3_USB.

## 6    Discussion and Conclusions

As air quality controls decrease U.S. anthropogenic precursor emissions to $O_3$, the relative importance of the

background influence on total surface $O_3$ increases. We use $O_3$ MDA8 concentrations spanning 2004-2012 from the

EPA AQS, CASTNet, and Mount Bachelor Observatory sites, and sensitivity simulations from the global GEOS-

Chem 3D chemistry transport model to estimate the influence from various individual background sources on $O_3$ in

each of the ten EPA regions in the continental U.S.A. The global scale of the GEOS-Chem model allows us to quantify

intercontinental transport (including global methane) in addition to regional natural and anthropogenic sources of $O_3$.

The sensitivity simulations span nine years, allowing us to examine the role of these sources in contributing to

interannual variability. Our analysis contrasts average- and high-$O_3$ days.

Correlations between monthly averages across 2004-2012 show that the model captures monthly variations

from year-to-year, especially during summer (JJA). The model shows substantial variability in simulated U.S.

background $O_3$ concentrations from year-to-year, on the order of 10-20 ppb between 2004-2012 in summer (Figure

7). We find that the extent to which the current three-year averaging period for assessing compliance with the National

Ambient Air Quality Standard for $O_3$ succeeds in smoothing out interannual variability depends on the range in

consecutive years, and thus varies by region and time period, but is generally not long enough to completely eliminate

the interannual variability in background $O_3$ (Figure 9).

We find substantial biases in the severity (+0-19 ppb in maximum daily 8-hour average (MDA8) $O_3$) and

timing of high-$O_3$ events in the model. The model underestimates the frequency of high events in spring, possibly

associated with stratospheric intrusions (Fiore et al., 2014; Zhang et al., 2011; 2014). Future efforts would benefit

from quantifying the stratospheric (as well as Asian) influence alongside the other background sources we consider.

We find a stronger influence of U.S. anthropogenic emissions on regionally averaged MDA8 $O_3$ (up to 30 ppb) from

BVOCs (up to 15 ppb) and soil $NO_x$ (up to 10 ppb) on the ten most biased days as compared to average days. We

conclude that regional production of $O_3$ is driving the pervasive high positive model bias in summer, as opposed to

transported background, although our sensitivity simulations do not allow us to rule out the possibility of a

coincident missing sink.

Our finding that BVOC emissions contribute to the summertime surface $O_3$ biases could reflect poor representation of the emissions (and subsequent oxidation chemistry). Earlier work has noted that MEGAN BVOC emissions are too high over California (Bash et al., 2016), Southeast Texas (Kota et al., 2015), the Ozarks in southern Missouri (Carlton and Baker, 2011), and across much of the U.S.A. (Wang et al., 2017). One recent model study uniformly reduced MEGAN isoprene emissions by 20% (Li et al., ACP 2018), but we did not apply any such scaling here. In regions that are highly NOx-sensitive, additional isoprene should not strongly influence $O_3$, as found over southeast Texas (Kota et al., 2015). While not eliminated entirely, the summertime model bias does lessen in the simulation with BVOC emissions set to zero, suggesting that the $O_3$ bias is indeed exacerbated if BVOC emissions are overestimated in the model.

On the ten days with the highest observed MDA8 $O_3$ values ($O_3$_top10obs) in each season, the model indicates that U.S. anthropogenic and biogenic VOC emissions are the most important drivers relative to average days, over most regions (Table 2, Table 3). $O_3$_top10obs_MAM and $O_3$_top10obs_SON days (i.e., the ten highest spring and fall MDA8 $O_3$ days) are up to 9°C warmer than average, but $O_3$_top10obs_JJA days (i.e., the ten highest summer MDA8 $O_3$ days) are only 1-2 °C warmer than average (Table 2). U.S. anthropogenic emissions enhance $O_3$_top10obs_JJA days by 5-11 ppb above average in the eastern U.S. regions, but by less than 2 ppb over the three western regions. Over these westernmost regions, U.S. background $O_3$ is 4-12 ppb higher on $O_3$_top10obs_JJA days than on average (Table 2). Across the continental U.S.A., biogenic VOC emissions enhance $O_3$ by 1-7 ppb above average on $O_3$_top10obs_JJA days, while intercontinental pollution is either similar or up to 2 ppb higher on average days (Table 3). Analysis of our simulations thus indicates that the highest $O_3$ events are associated with regional $O_3$ production rather than transported background.

From 2004-2006 to 2010-2012, MDA8 $O_3$ concentrations on $O_3$_top10obs_JJA days vary from year-to-year, but show little overall trend, decreasing by 3 ppb in both the observations and the model averaged over all regions (Figure 6, Table 4). With our sensitivity simulations, we interpret this lack of an overall trend as a balance between rising U.S. background $O_3$ (by 2 ppb for $O_3$_USB from 2004-2006 to 2010-2012 averaged over all regions) and declining U.S. anthropogenic emissions (by 6 ppb for $O_3$_USA from 2004-2006 to 2010-2012 averaged over all regions). The declining influence of U.S. anthropogenic emissions on $O_3$_top10obs_JJA days is consistent with earlier

work showing high-$O_3$ concentrations decreasing in response to regional precursor emissions controls since the late 1990s (e.g. Cooper et al., 2012, 2014b; Frost et al., 2006; Simon et al., 2016).

In contrast to previous work, including with the GEOS-Chem model (e.g. Fiore et al., 2014 and references therein), we find that U.S. background $O_3$ tends to be higher in summer than in spring in most regions. This likely reflects differences in the isoprene chemistry, specifically the isoprene nitrates, between our version of GEOS-Chem (Mao et al., 2013) and older versions that treat isoprene nitrates as greater sinks for $NO_x$ and thereby, suppress $O_3$ production. The coarse resolution of our model will excessively mix isoprene and soil $NO_x$ sources (e.g., Yu et al.,

2016), and thus may exaggerate the relative importance of enhanced background $O_3$ resulting from soil $NO_x$ and isoprene. Nevertheless, the model skill at capturing the observed year-to-year variability in the regionally averaged ten highest days lends some confidence to its attribution of this variability to natural sources (e.g. Figure 6). Future work with high-resolution models (e.g., at the regional scale, ideally with boundary conditions that include source attributions from a global model) is needed, along with observational evidence, to quantify the extent to which

biogenic VOC and $NO_x$ contribute to the highest observed $O_3$ levels in the warm season. The importance of temperature sensitive sources like biogenic VOC and $NO_x$ emissions to background $O_3$ imply that in a warmer climate, these background influences on $O_3$ will play an even more important role in driving up $O_3$ levels.

**Acknowledgments**

We acknowledge insightful discussions with Gail Tonnesen and Pat Dolwick (U.S. EPA). We gratefully

acknowledge support from NASA AQAST (NNX12AF15G) and NASA HAQAST (NNX16AQ20G). This project has been funded in part by the United States Environmental Protection Agency under assistance agreement RD83587801 to AMF. The contents of this document do not necessarily reflect the views and policies of the Environmental Protection Agency, nor does the EPA endorse trade names or recommend the use of commercial products mentioned in this document. Data behind the figures is provided online at Columbia University Academic

Commons.

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

 **Figures**

**Table 1: Sensitivity simulations with the GEOS-Chem model and their application to estimate sources of ground-level $O_3$.**

| *Ozone Source* | *Definition* | *Notation* |
|---|---|---|
| Base | Standard simulation | $O_3$_Base |
| Natural Background | Simulation with no global anthropogenic emissions + preindustrial $CH_4$ levels | $O_3$_NAT |
| North American Background | Simulation with no North American anthropogenic emissions | $O_3$_NAB |
| U.S. Background | Simulation with no U.S. anthropogenic emissions | $O_3$_USB |
| U.S. Anthropogenic Emissions | $O_3$_Base – $O_3$_USB | $O_3$_USA |
| Anthropogenic Emissions from Canada and Mexico | $O_3$_USB – $O_3$_NAB | $O_3$_CA+MX |
| Intercontinental Transport + Preindustrial $CH_4$ Levels | $O_3$_NAB – $O_3$_NAT | $O_3$_ICT+CH4 |
| North American Lightning $NO_x$ | $O_3$_Base – simulation with the lightning $NO_x$ source shut off | $O_3$_NALNO$_x$ |
| Soil $NO_x$ Emissions | $O_3$_Base – simulation with the soil $NO_x$ emissions shut off | $O_3$_SNO$_x$ |
| Terrestrial Biogenic VOC Emissions | $O_3$_Base – simulation with the terrestrial biogenic emissions shut off | $O_3$_BVOC |
| All Emissions except Terrestrial Biogenic VOCs | Simulation with terrestrial biogenic VOC emissions shut off | $O_3$_noBVOC |
| Biomass Burning Emissions | $O_3$_Base – simulation with biomass burning emissions ($NO_x$, CO, VOCs, aerosols, and precursors from fires) shut off | $O_3$_BB |

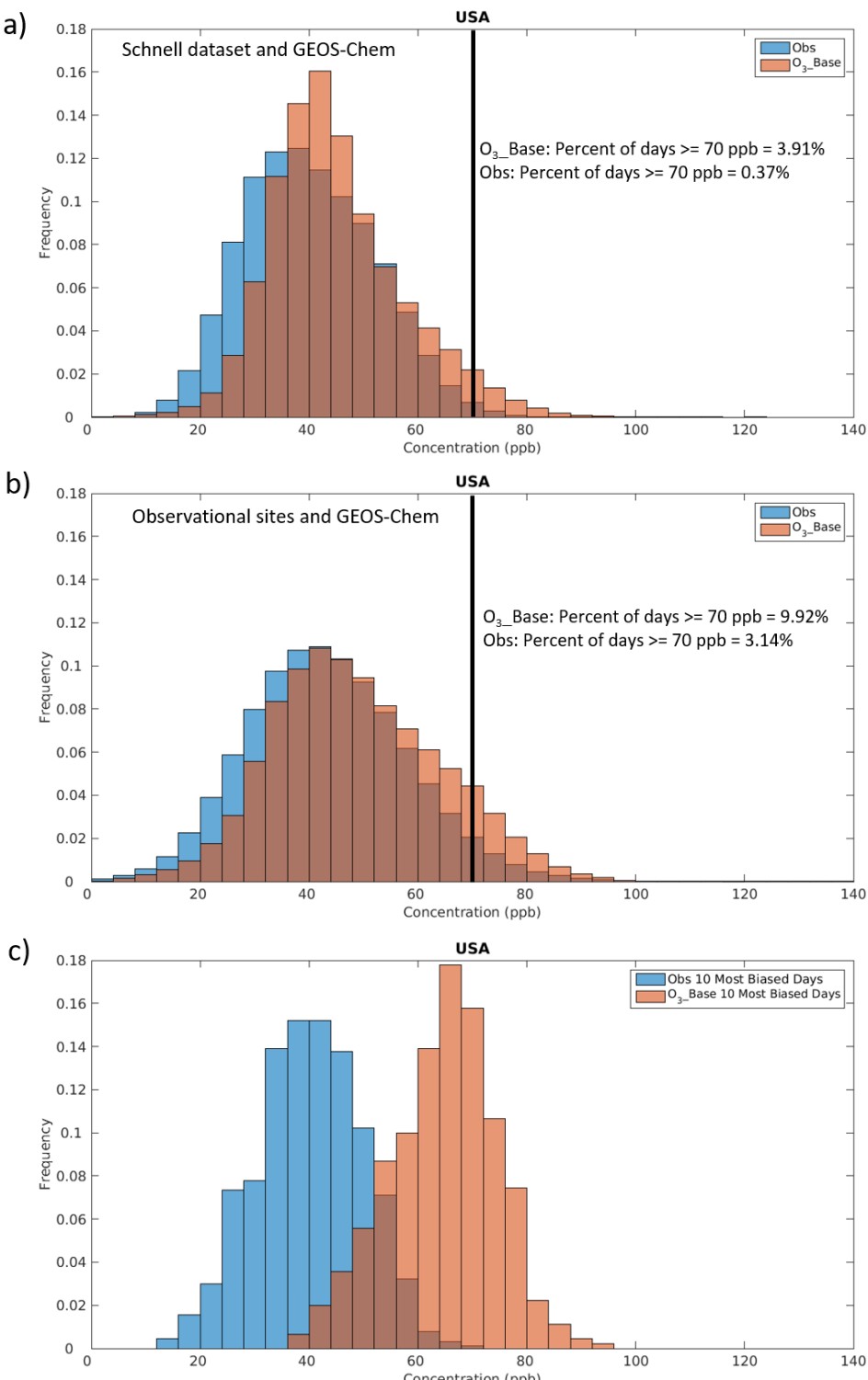

**Figure 1: Frequency distribution of regionally averaged U.S. MDA8 O₃ values from 2004-2012 in the (a) Schnell and Prather (2017) dataset interpolated to 2° by 2.5° and (b) at individual observational sites prior to averaging over each of the 10 EPA regions (total number of points is 9 years x 365 or 366 days x 10 regions) in the observations (blue) and the GEOS-Chem model (orange). c) As in panel (b) but selecting for the 10 most biased days in each region (total number of points is 9 years x 10 days x 10 regions). The line drawn at 70 ppb in panels (a) and (b) is the current O₃ NAAQS level.**


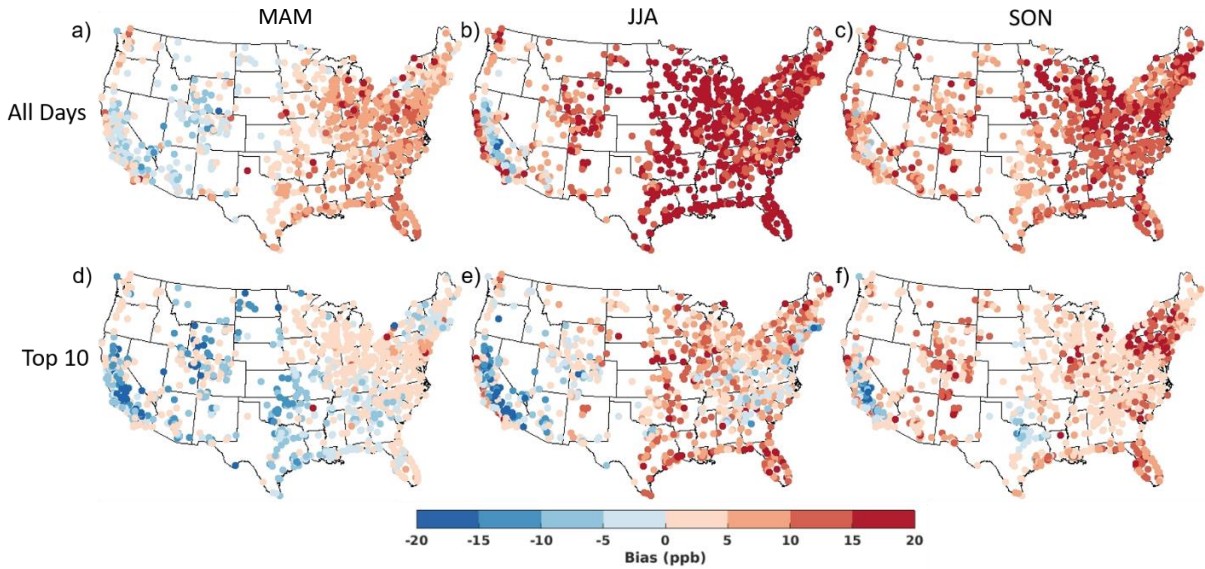

**Figure 2: Average MDA8 O₃ model bias (O₃_Base – observed) on all days in (a)MAM, (b) JJA, and (c) SON versus on the (d) O₃_top10obs_MAM, (e) O₃_top10obs_JJA, and (f) O₃_top10obs_SON days at each observational site averaged across 2004-2012.**

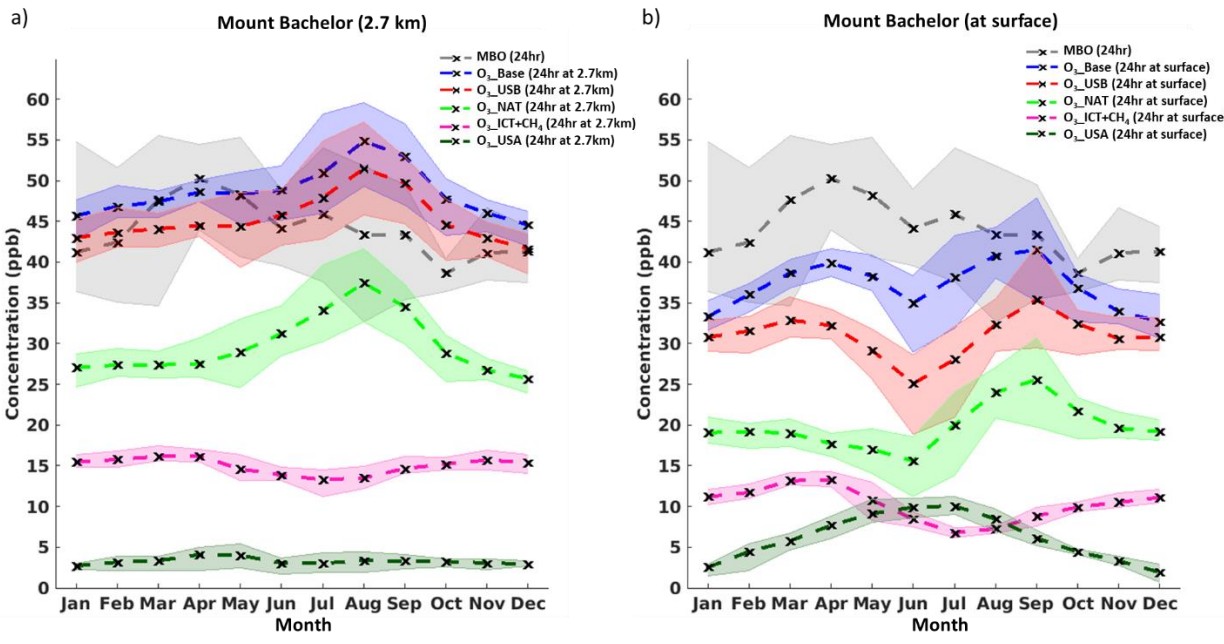


**Figure 3: Monthly 2004-2012 average 24-hour O₃ concentrations at Mount Bachelor Observatory. Observations (grey) are the same in both panels. Simulations from the GEOS-Chem model are sampled in the grid cell containing Mount Bachelor at (a) 2.7 km (the height of the Mount Bachelor Observatory) and at (b) the surface: O₃_Base (blue), O₃_USB (red), O₃_NAT (light green), O₃_ICT+CH₄ (pink), and O₃_USA (dark green). The shaded range spans the highest and**

**lowest years.**

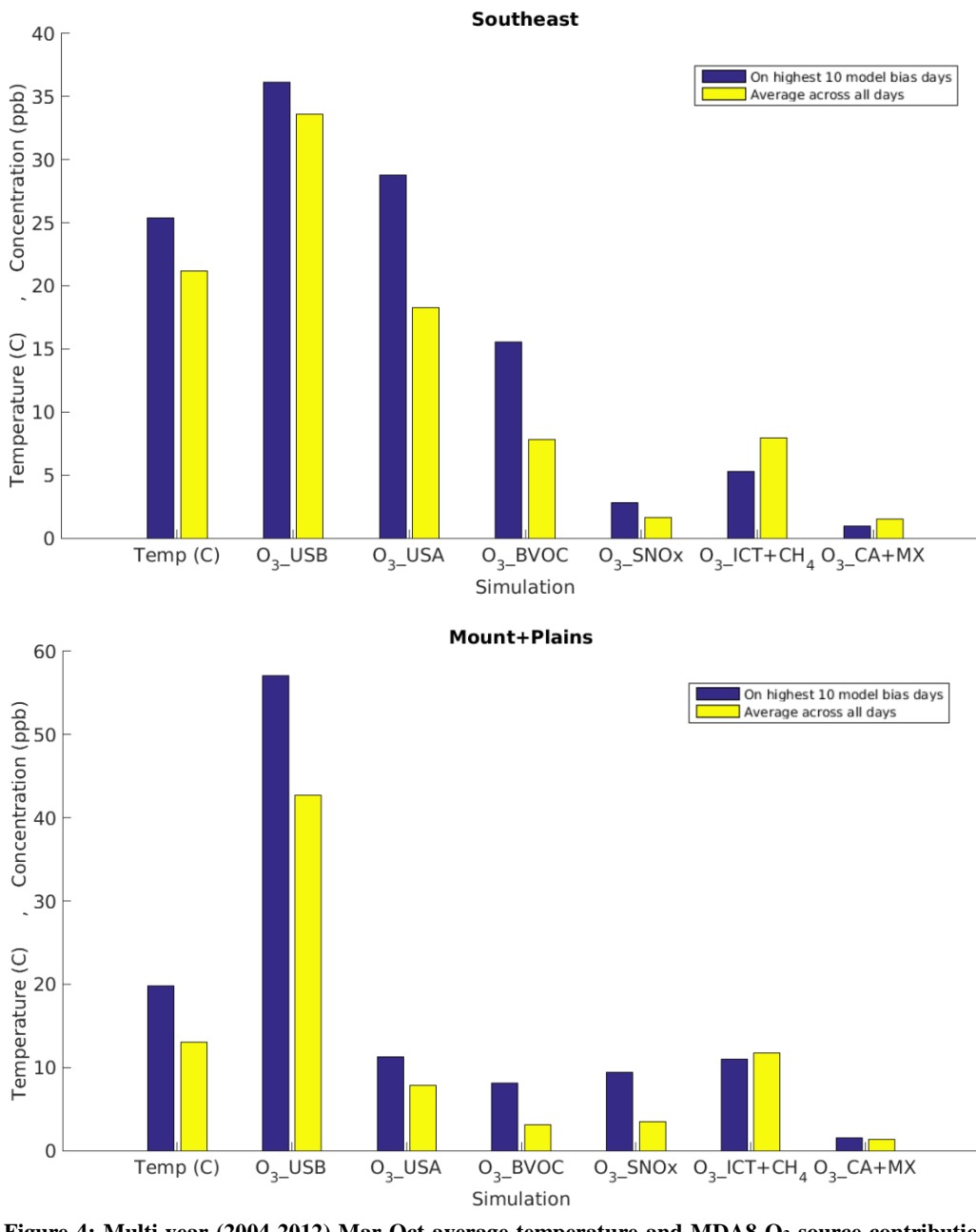

**Figure 4: Multi-year (2004-2012) Mar-Oct average temperature and MDA8 O₃ source contributions estimated with the GEOS-Chem model in the (a) Southeast and (b) Mountain and Plains regions on the 10 most biased days (blue) versus averaged across all days (yellow). Note that the two regions are on different scales.**


**Table 2: Summary information for each region. The "Model Bias" column shows the model bias in each region on the (1) $O_3$_top10obs days in each season (average of 2004-2012), 2) across all days in each season (average of 2004-2012), and (3) the difference between these values, rounded to the nearest whole number. The other columns show the concentration for the observations, $O_3$_Base, and $O_3$_USA, and daily average temperature (in degrees C) on the (1) $O_3$_top10obs days in each season (average of 2004-2012), (2) across all days in each season (average of 2004-2012), and (3) the difference between these values.**

| Region | Metric / Season | Model Bias MAM | JJA | SON | O₃_Base MAM | JJA | SON | Obs MAM | JJA | SON | O₃_USB MAM | JJA | SON | O₃_USA MAM | JJA | SON | Temperature (C) MAM | JJA | SON |
|---|---|---|---|---|---|---|---|---|---|---|---|---|---|---|---|---|---|---|---|
| New England | Top 10 Days | -1 | 9 | 12 | 57 | 72 | 59 | 58 | 62 | 59 | 35 | 39 | 36 | 22 | 33 | 23 | 12 | 22 | 18 |
| | Avg all days | 2 | 14 | 10 | 46 | 56 | 41 | 44 | 43 | 31 | 35 | 35 | 32 | 12 | 21 | 9 | 7 | 19 | 11 |
| | Difference | -3 | -4 | 2 | 11 | 16 | 18 | 13 | 20 | 28 | 0 | 4 | 3 | 10 | 11 | 14 | 5 | 2 | 6 |
| NY+NJ | Top 10 Days | 4 | 14 | 15 | 63 | 80 | 65 | 59 | 67 | 65 | 36 | 42 | 38 | 27 | 38 | 27 | 16 | 24 | 20 |
| | Avg all days | 5 | 18 | 11 | 49 | 65 | 42 | 44 | 47 | 31 | 35 | 39 | 33 | 14 | 27 | 10 | 9 | 21 | 12 |
| | Difference | -1 | -5 | 4 | 14 | 15 | 23 | 15 | 19 | 34 | 1 | 3 | 6 | 13 | 11 | 18 | 6 | 2 | 7 |
| Mid-Atlantic | Top 10 Days | 3 | 13 | 15 | 65 | 81 | 69 | 63 | 68 | 69 | 34 | 40 | 37 | 31 | 40 | 32 | 18 | 25 | 21 |
| | Avg all days | 6 | 18 | 12 | 51 | 70 | 45 | 46 | 52 | 33 | 34 | 38 | 32 | 17 | 32 | 13 | 12 | 23 | 14 |
| | Difference | -3 | -5 | 3 | 14 | 11 | 24 | 17 | 16 | 36 | 0 | 2 | 5 | 14 | 9 | 19 | 6 | 2 | 7 |
| Southeast | Top 10 Days | 0 | 13 | 10 | 62 | 72 | 63 | 61 | 59 | 63 | 34 | 39 | 34 | 27 | 34 | 28 | 19 | 26 | 21 |
| | Avg all days | 6 | 19 | 12 | 55 | 65 | 51 | 48 | 46 | 39 | 34 | 37 | 32 | 21 | 28 | 19 | 17 | 26 | 18 |
| | Difference | -6 | -6 | -1 | 7 | 7 | 12 | 13 | 13 | 24 | 1 | 2 | 3 | 7 | 5 | 10 | 2 | 1 | 3 |
| Midwest | Top 10 Days | 4 | 14 | 17 | 63 | 77 | 70 | 59 | 63 | 70 | 36 | 44 | 42 | 27 | 33 | 27 | 17 | 24 | 21 |
| | Avg all days | 6 | 19 | 11 | 49 | 68 | 43 | 44 | 48 | 32 | 34 | 42 | 33 | 15 | 26 | 10 | 10 | 22 | 12 |
| | Difference | -1 | -5 | 6 | 14 | 10 | 26 | 15 | 15 | 37 | 1 | 2 | 9 | 12 | 8 | 17 | 7 | 1 | 9 |
| South Central | Top 10 Days | 0 | 13 | 9 | 60 | 75 | 67 | 60 | 62 | 67 | 39 | 45 | 40 | 21 | 30 | 26 | 20 | 27 | 23 |
| | Avg all days | 5 | 17 | 10 | 52 | 62 | 51 | 47 | 46 | 41 | 36 | 41 | 35 | 16 | 21 | 16 | 18 | 27 | 19 |
| | Difference | -5 | -4 | -2 | 8 | 12 | 15 | 14 | 16 | 26 | 3 | 4 | 5 | 6 | 9 | 10 | 2 | 1 | 4 |
| Plains | Top 10 Days | 0 | 13 | 13 | 58 | 74 | 67 | 58 | 61 | 67 | 37 | 47 | 42 | 21 | 28 | 25 | 17 | 26 | 22 |
| | Avg all days | 5 | 18 | 10 | 50 | 67 | 45 | 44 | 49 | 35 | 34 | 44 | 34 | 15 | 23 | 11 | 13 | 25 | 13 |
| | Difference | -6 | -5 | 3 | 8 | 8 | 23 | 14 | 13 | 33 | 2 | 3 | 9 | 6 | 5 | 14 | 4 | 1 | 9 |
| Mountains + Plains | Top 10 Days | -1 | 8 | 13 | 56 | 69 | 64 | 57 | 60 | 64 | 45 | 57 | 54 | 11 | 12 | 10 | 12 | 22 | 18 |
| | Avg all days | 0 | 11 | 9 | 50 | 64 | 48 | 50 | 53 | 39 | 41 | 53 | 41 | 10 | 11 | 7 | 7 | 20 | 9 |
| | Difference | -1 | -2 | 4 | 6 | 5 | 16 | 7 | 7 | 25 | 5 | 4 | 12 | 1 | 0 | 3 | 5 | 2 | 9 |
| Pacific SW | Top 10 Days | -3 | 3 | 6 | 57 | 64 | 63 | 60 | 62 | 63 | 41 | 47 | 48 | 16 | 18 | 15 | 18 | 25 | 24 |
| | Avg all days | 0 | 4 | 8 | 49 | 57 | 49 | 49 | 53 | 42 | 37 | 41 | 39 | 12 | 16 | 10 | 14 | 23 | 17 |
| | Difference | -3 | -1 | -2 | 8 | 7 | 14 | 10 | 9 | 21 | 4 | 6 | 8 | 4 | 2 | 5 | 5 | 2 | 7 |
| Pacific NW | Top 10 Days | -1 | 6 | 11 | 48 | 59 | 51 | 49 | 52 | 51 | 39 | 49 | 44 | 9 | 10 | 7 | 12 | 22 | 17 |
| | Avg all days | 2 | 8 | 10 | 43 | 46 | 40 | 41 | 38 | 30 | 35 | 36 | 36 | 8 | 10 | 4 | 8 | 17 | 10 |
| | Difference | -3 | -2 | 0 | 5 | 13 | 11 | 9 | 14 | 21 | 4 | 12 | 9 | 1 | 0 | 3 | 4 | 4 | 7 |

Table 3: Summary information for each region. Each column shows the concentration for each background O₃ source influence on the (1) O₃_top10obs days in each season (average of 2004-2012), (2) across all days in each season (average of 2004-2012), and (3) the difference between these values, rounded to the nearest whole number.

| Region | Metric / Season | O₃_USB MAM | JJA | SON | O₃_BVOC MAM | JJA | SON | O₃_SNOₓ MAM | JJA | SON | O₃_NALNOₓ MAM | JJA | SON | O₃_ICT+CH₄ MAM | JJA | SON | O₃_CA+MX MAM | JJA | SON |
|---|---|---|---|---|---|---|---|---|---|---|---|---|---|---|---|---|---|---|---|
| New England | Top 10 Days | 35 | 39 | 36 | 6 | 17 | 13 | 1 | 3 | 2 | 1 | 2 | 1 | 8 | 3 | 5 | 7 | 7 | 5 |
| | Avg all days | 35 | 35 | 32 | 2 | 10 | 6 | 1 | 3 | 2 | 1 | 2 | 2 | 10 | 4 | 7 | 6 | 6 | 4 |
| | Difference | 0 | 4 | 3 | 4 | 7 | 8 | 0 | 0 | 1 | 0 | 0 | 0 | -2 | -1 | -3 | 1 | 1 | 2 |
| NY+NJ | Top 10 Days | 36 | 42 | 38 | 9 | 20 | 17 | 1 | 4 | 3 | 1 | 2 | 2 | 7 | 2 | 4 | 6 | 6 | 5 |
| | Avg all days | 35 | 39 | 33 | 3 | 14 | 7 | 1 | 3 | 2 | 1 | 2 | 2 | 10 | 4 | 7 | 5 | 6 | 4 |
| | Difference | 1 | 3 | 6 | 6 | 6 | 9 | 0 | 0 | 1 | 0 | 0 | 0 | -2 | -1 | -3 | 1 | 0 | 2 |
| Mid-Atlantic | Top 10 Days | 34 | 40 | 37 | 10 | 20 | 18 | 1 | 4 | 3 | 1 | 3 | 2 | 7 | 3 | 5 | 4 | 3 | 4 |
| | Avg all days | 34 | 38 | 32 | 5 | 16 | 9 | 1 | 3 | 2 | 1 | 3 | 2 | 9 | 4 | 7 | 4 | 4 | 3 |
| | Difference | 0 | 2 | 5 | 5 | 4 | 9 | 0 | 1 | 1 | 0 | 0 | 0 | -2 | -1 | -2 | 0 | 0 | 1 |
| Southeast | Top 10 Days | 34 | 39 | 34 | 7 | 16 | 14 | 2 | 4 | 2 | 1 | 3 | 2 | 8 | 4 | 6 | 2 | 2 | 2 |
| | Avg all days | 34 | 37 | 32 | 5 | 14 | 9 | 1 | 3 | 2 | 2 | 4 | 2 | 9 | 5 | 7 | 2 | 1 | 2 |
| | Difference | 1 | 2 | 3 | 2 | 2 | 4 | 0 | 1 | 1 | -1 | -1 | 0 | 0 | -1 | -1 | 0 | 1 | 0 |
| Midwest | Top 10 Days | 36 | 44 | 42 | 8 | 16 | 16 | 2 | 6 | 5 | 1 | 2 | 2 | 6 | 1 | 4 | 3 | 3 | 3 |
| | Avg all days | 34 | 42 | 33 | 3 | 13 | 8 | 1 | 6 | 2 | 1 | 2 | 2 | 9 | 2 | 6 | 4 | 3 | 2 |
| | Difference | 1 | 2 | 9 | 4 | 3 | 8 | 1 | 0 | 2 | 0 | 0 | 0 | -3 | -1 | -3 | 0 | 0 | 1 |
| South Central | Top 10 Days | 39 | 45 | 40 | 6 | 17 | 14 | 3 | 5 | 4 | 2 | 4 | 2 | 8 | 5 | 6 | 3 | 2 | 2 |
| | Avg all days | 36 | 41 | 35 | 4 | 12 | 8 | 2 | 4 | 2 | 2 | 6 | 2 | 9 | 7 | 8 | 3 | 2 | 2 |
| | Difference | 3 | 4 | 5 | 2 | 5 | 6 | 1 | 1 | 2 | 0 | -2 | 0 | -1 | -2 | -2 | 0 | 0 | 0 |
| Plains | Top 10 Days | 37 | 47 | 42 | 5 | 16 | 14 | 3 | 8 | 6 | 1 | 3 | 2 | 7 | 2 | 4 | 3 | 1 | 1 |
| | Avg all days | 34 | 44 | 34 | 3 | 13 | 7 | 2 | 8 | 3 | 1 | 3 | 2 | 8 | 3 | 7 | 3 | 2 | 2 |
| | Difference | 2 | 3 | 9 | 2 | 3 | 7 | 1 | 0 | 3 | 0 | 0 | 0 | -1 | -1 | -2 | 0 | 0 | 0 |
| Mountains + Plains | Top 10 Days | 45 | 57 | 54 | 1 | 8 | 7 | 3 | 9 | 7 | 3 | 5 | 5 | 12 | 11 | 12 | 2 | 1 | 1 |
| | Avg all days | 41 | 53 | 41 | 1 | 7 | 4 | 2 | 8 | 3 | 2 | 5 | 4 | 12 | 11 | 12 | 2 | 2 | 1 |
| | Difference | 5 | 4 | 12 | 1 | 1 | 3 | 1 | 1 | 4 | 1 | 0 | 1 | 0 | 1 | 1 | 0 | 0 | 0 |
| Pacific SW | Top 10 Days | 41 | 47 | 48 | 3 | 9 | 9 | 3 | 5 | 5 | 2 | 4 | 4 | 11 | 8 | 10 | 2 | 2 | 2 |
| | Avg all days | 37 | 41 | 39 | 1 | 7 | 5 | 1 | 4 | 3 | 2 | 4 | 3 | 12 | 8 | 11 | 1 | 2 | 2 |
| | Difference | 4 | 6 | 8 | 2 | 2 | 4 | 1 | 1 | 2 | 0 | 0 | 1 | -1 | 0 | -1 | 0 | 0 | 0 |
| Pacific NW | Top 10 Days | 39 | 49 | 44 | 0 | 9 | 7 | 2 | 7 | 5 | 1 | 3 | 3 | 12 | 9 | 10 | 3 | 4 | 2 |
| | Avg all days | 35 | 36 | 36 | -1 | 4 | 3 | 1 | 4 | 2 | 1 | 2 | 3 | 13 | 9 | 10 | 2 | 3 | 1 |
| | Difference | 4 | 12 | 9 | 1 | 5 | 4 | 1 | 3 | 3 | 0 | 1 | 0 | 0 | 0 | 0 | 1 | 1 | 1 |

a)                     b)

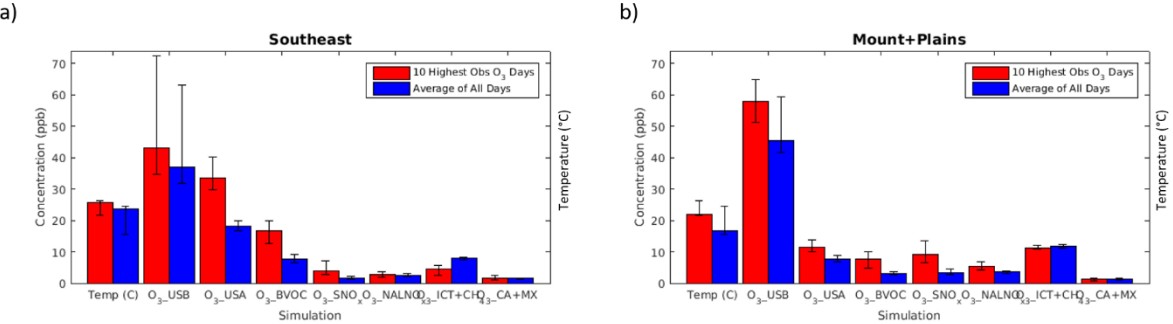

Figure 5: Average 2004-2012 influence of each sensitivity simulation to O₃_Base in the (a) Southeast and (b) Mountains and Plains regions on MDA8 O₃_top10obs_JJA days (red) versus averaged across all days (blue). Error bars show the concentration on the lowest versus highest year for each sensitivity simulation in each region.

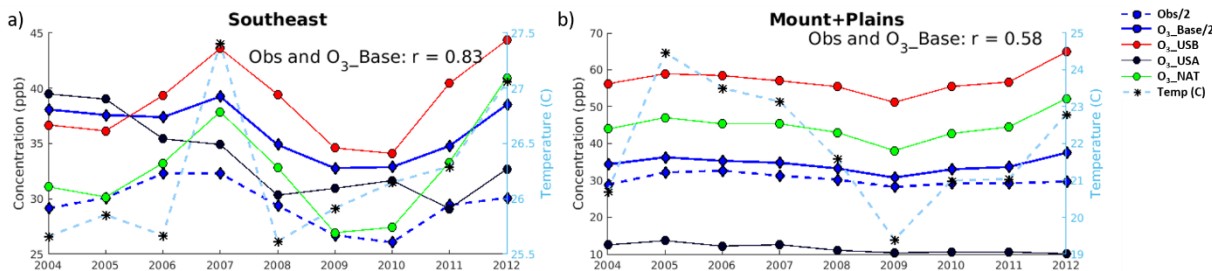

**Figure 6: Average yearly MDA8 O₃_top10obs_JJA concentrations for observations (divided by 2 to fit on the same axes; blue dashed line), O₃_Base (divided by 2; blue solid line), O₃_USB (red), O₃_USA (black), O₃_NAT (green) MDA8, and daily average temperature (in degrees C; light blue) in the (a) Southeast and (b) Mountains and Plains regions.**


**Table 4: Change in MDA8 O₃ concentrations from 2004-2006 to 2010-2012 on O₃_top10obs_JJA days in the observations, O₃_Base, O₃_USB, O₃_USA, and temperature.**

|  | Obs | O₃_Base | O₃_USB | O₃_USA | Temperature (C) |
|---|---|---|---|---|---|
| **New England** | -6 | -4 | 6 | -10 | 2 |
| **NY+NJ** | -2 | -4 | 3 | -7 | 1 |
| **Mid-Atlantic** | 0 | -3 | 4 | -7 | 1 |
| **Southeast** | -4 | -5 | 2 | -7 | 1 |
| **Midwest** | -2 | -4 | 2 | -6 | 0 |
| **South Central** | -6 | -2 | 5 | -7 | 1 |
| **Plains** | -1 | -2 | 4 | -5 | 1 |
| **Mountains + Plains** | -4 | -1 | 1 | -2 | -1 |
| **Pacific SW** | -3 | -4 | 0 | -4 | -1 |
| **Pacific NW** | -7 | -5 | -4 | -1 | -1 |
| ***Average*** | ***-3*** | ***-3*** | ***2*** | ***-6*** | ***0*** |

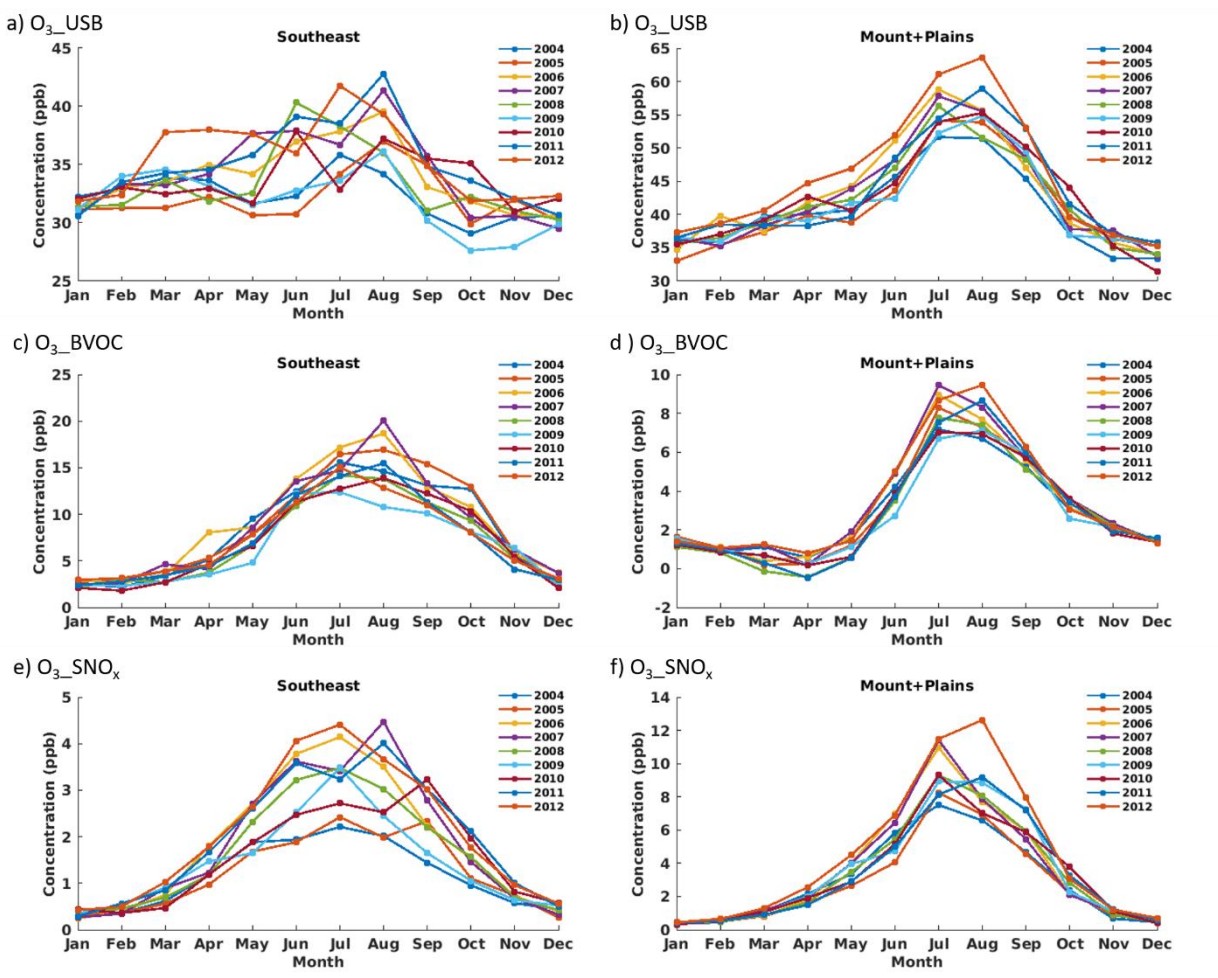

**Figure 7: Monthly average MDA8 O$_3$_USB (a, b), O$_3$_BVOC (c, d), and O$_3$_SNO$_x$ (e, f) concentrations in the Southeast (a, c, e) and Mountains and Plains (b, d, f) regions.**


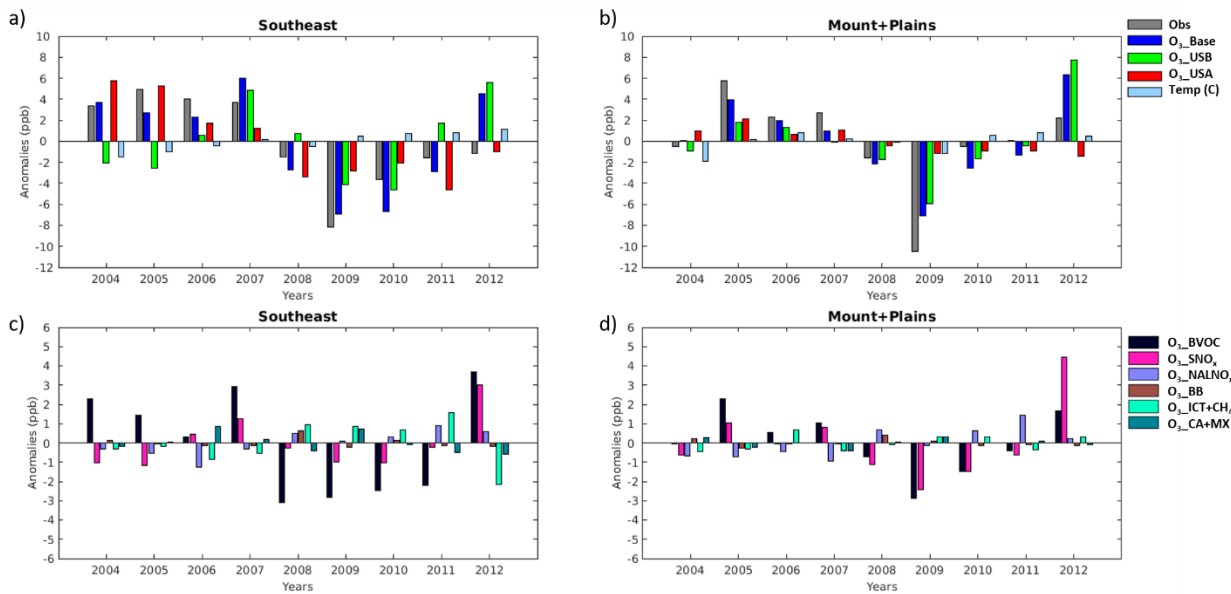

**Figure 8: Anomaly on the MDA8 O₃_top10obs_JJA days relative to the 2004-2012 average in the Southeast (a, c) and in the Mountains and Plains (b, d) regions. Panels (a) and (b) show the observations, O₃_Base, O₃_USB, O₃_USA, and temperature (in degrees C). Panels (c) and (d) show O₃_BVOC, O₃_SNOₓ, O₃_NALNOₓ, O₃_BB, O₃_ICT+CH₄, and O₃_CA+MX.**


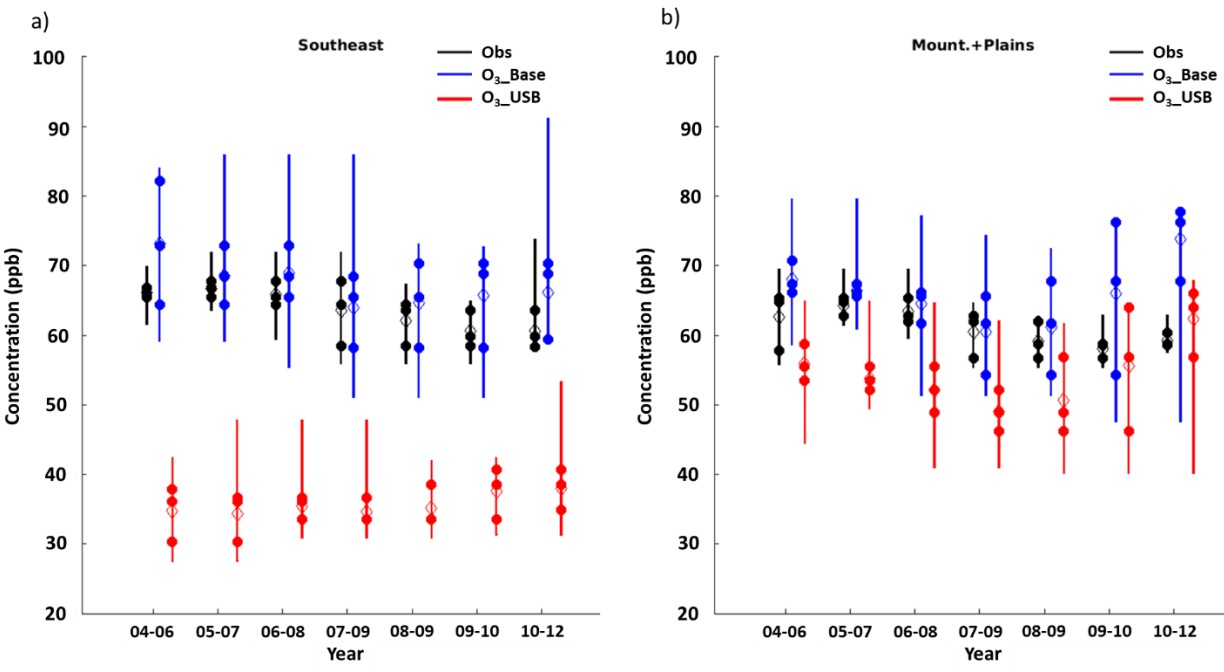

**Figure 9: The three 4th highest days in each year (solid dots) that went into the calculation of the three-year average of the 4th highest MDA8 O₃ day (hollow diamond). Error bars show the range between the highest and lowest O₃_top10obs days across each 3-year span (i.e, across 30 total points) occurring between March and October in the (a) Southeast and (b) Mountains and Plains regions in the observations (black), and the O₃_Base (blue) and O₃_USB (red) simulations sampled**

**on the same days as the top 10 observed values.**

**Table 5: Summary information for each region. The first row next to each region reports the range across 2004-2012 of the 4th highest values from each of the 9 individual years for the observations, O₃_Base, and O₃_USB. The second row reports the range across 2004-2012 of each of the 3-year averages of the 4th highest values (7 values) in each region for the observations, O₃_Base, and O₃_USB.**


| Region | Range | Obs | O₃_Base | O₃_USB |
|---|---|---|---|---|
| **New England** | 4th highest day | 15 | 16 | 10 |
| | 3-year average 4th highest day | 9 | 10 | 3 |
| | *Difference* | **-6** | **-6** | **-7** |
| **NY+NJ** | 4th highest day | 11 | 10 | 12 |
| | 3-year average 4th highest day | 6 | 2 | 6 |
| | *Difference* | **-5** | **-8** | **-6** |
| **Mid-Atlantic** | 4th highest day | 13 | 36 | 25 |
| | 3-year average 4th highest day | 7 | 21 | 10 |
| | *Difference* | **-6** | **-15** | **-15** |
| **Southeast** | 4th highest day | 9 | 24 | 10 |
| | 3-year average 4th highest day | 6 | 9 | 4 |
| | *Difference* | **-3** | **-15** | **-7** |
| **Midwest** | 4th highest day | 13 | 22 | 24 |
| | 3-year average 4th highest day | 8 | 11 | 10 |
| | *Difference* | **-6** | **-11** | **-14** |
| **South Central** | 4th highest day | 11 | 26 | 22 |
| | 3-year average 4th highest day | 8 | 13 | 13 |
| | *Difference* | **-3** | **-13** | **-9** |
| **Plains** | 4th highest day | 14 | 32 | 24 |
| | 3-year average 4th highest day | 9 | 18 | 11 |
| | *Difference* | **-5** | **-15** | **-13** |
| **Mountains + Plains** | 4th highest day | 9 | 23 | 20 |
| | 3-year average 4th highest day | 6 | 13 | 13 |
| | *Difference* | **-2** | **-10** | **-7** |
| **Pacific SW** | 4th highest day | 5 | 23 | 20 |
| | 3-year average 4th highest day | 3 | 5 | 5 |
| | *Difference* | **-2** | **-18** | **-15** |
| **Pacific NW** | 4th highest day | 11 | 14 | 15 |
| | 3-year average 4th highest day | 5 | 9 | 12 |
| | *Difference* | **-5** | **-5** | **-3** |