# Peer review of "Average versus high surface ozone levels over the continental U.S.A.: Model bias, background influences, and interannual variability"

_Atmospheric Chemistry and Physics, 2018_

## Referee Comment (RC1) · Anonymous Referee #1 · 27 Mar 2018

This manuscript presents an attempt to derive information about mean maximum daily 8-hour average (MDA8) O3 in the United States, based on ambient measurements and using the global model GEOS-CHEM. Sensitivity simulations examine different sources that affect the 10 highest O3 events and that affect the 10 days with highest model bias against observations for 2004 to 2012 for each 10 EPA regions.

General comments: The analysis is a valuable contribution to the current understanding of ground level O3 and air quality standard settings. The topic itself is highly relevant and thus will be of interest to the readers of ACP. Discussion of the results and their implications is also scientifically sound and the paper includes comprehensive analy-

ses. However, I feel that the paper tried to cover lots of information, which makes it a bit hard for the reader to follow key conclusions from this study. Thus I recommend that the paper should be published after addressing the following comments.

General Some comment about day of week effects and model biases in temperature as they relate to the questions raised in the paper seem warranted. There should be dramatic changes in the temperature dependence of ozone over this period coincident with the NOx changes. Those changes should have a day of week variation that might appear in the top 10 days.

Specific comments: The authors use terms "Baseline O3" and "U.S. background O3". U.S. background O3 is defined as "the O3 levels that would exist in the absence of U.S. anthropogenic emissions of precursors" and Baseline O3 is defined as "tropospheric O3 concentrations that have a negligible influence from local anthropogenic emissions". They sound the same, don't they? If yes, please be consistent in the text.

Page 4, lines 106-109, Please clarify if the authors apply Schnell et al. (2014)'s interpolation procedure or they use their dataset. Schnell et al. (2014) use surface MDA8 O3 measurements from air quality networks for 2000–2009, while this paper analyzes the data from 2004-2012.

A valuable addition would be a statement about the chemistry scheme applied in the version GEOS-Chem at the 2.3 section (GEOS-Chem model simulation). The authors mention issues of isoprene chemistry in last paragraph of Conclusions but a brief description or reference to the specific version of the chemistry should be presented before the last paragraph of the paper.

The last paragraph of page 6 needs elaboration where the authors state the sensitivity simulations. The notations for all model simulations should be mentioned and the description of Table 1 should be modified so that the Table is read from top to bottom.

Figure 3 : Observed O3 concentrations should be represented in a different color to be

more visible (maybe black instead of grey) and I would also suggest to plot the curves as an average for 2004-2012 period with associated error bars.

Minor comments: The tables start from Table 2 at the manuscript and Table 1 is referenced at Page 8 for the first time. Please fix ordering of table numbers as they appear in the text.

Page 7, line 195 : "a maximum in and" should read "a maximum in summertime and"

---

## Referee Comment (RC2) · Anonymous Referee #2 · 29 Mar 2018

General:

The paper is very well-written and concerns a topic of considerable interest to air quality planners. However, there are some concerns about the suitability of this particular model configuration to address some of the stated objectives of the paper (lines 76-79), as discussed below. In general, the paper would be improved if there was greater clarity about the potential connections between the findings and possible configuration concerns. The value of the paper would be enhanced if the conclusions section was bolstered with a "next steps" or "considerations" sentence or two that described how such a global model-based sensitivity study could be improved in the future.

[Figure]

In particular, there is concern about the use of a coarse resolution model (2 x 2.5 deg) to investigate contributions of U.S. anthropogenic emissions (O3_USA) given that those contributions originate at scales much smaller than the resolution of the model (i.e., point source emissions, urban area emissions). The paper acknowledges the limitations associated with the coarse modeled resolution in several places (lines 212, 242, 399). The paper may want to revisit these caveats in the conclusion and perhaps provide some thoughts on what alternate global model configurations would be better suited for an analysis of source contributions.

Kudos to the authors for providing sufficient detail regarding the performance evaluation to allow readers to interpret the contribution findings in light of the model bias/error. However, the ozone overestimations (3-14 ppb in JJA MDA8 top 10 days by region, even worse for JJA all-day averages) suggest caution should be exercised in over-interpreting the contributions. Based on Figure 5 and the associated analyses, it appears that the model vastly overestimates ozone on hot days in the late summer, especially in the eastern U.S. (even without consideration of potential additional emissions due to increased power demand on those days). Section 3.3 briefly summarizes potential causes for this overestimation based on similar studies, but it would be valuable if the paper provided more application-specific hypotheses for the underlying cause. FYI, along w/ the possible causes from the Travis research, others have raised concerns about MEGAN biogenic VOC estimates (e.g., Bash et al., 2016; Carlton and Baker, 2011; Kota et al., 2015; Wang et al., 2017).

One of the more noteworthy findings concerns the modeled trends over the 10-year period (e.g., lines 386-389) where the analysis appears to confirm previous findings that improving trends in U.S. air quality from emissions controls have been tempered by increases in background contributions (and increases in temperature). However, one interesting finding here that could use additional explanation is the regional breakout of this "USB vs. USA" tradeoff. Table 5 suggests that the largest increases in high JJA-day O3 USB concentrations between 2004-2006 and 2010-2012 have occurred in the

[Figure]

New England and Mid-Atlantic regions, not the western regions where USB concerns are typically greatest. More explanation of the regional differences in modeled USB trends would be beneficial (e.g., is this just an artifact of the meteorology of the two 3-year periods in these regions).

Given model performance findings, would the authors see value in revising the "2-step" contribution analysis (assessing contributions on high-bias days, then assessing contributions on high/all observed days) to a "3-step" contribution where as an intermediate step you also investigated contributions on top-10 modeled days? This could be valuable presuming that the subset of days would differ from top 10 highest bias days.

Rather than lumping the Mount Bachelor observations (and subsequent pairs) with surface sites in Region 10, it would be interesting to see how model contributions varied as functions of model performance and observation concentration as a standalone site.

Specific:

Line 86: "download" should be "downloaded".

Lines 124-127: Would be easier to read, if a new sentence was started w/ "On the days with . . .".

Line 146: "Avian" should be "Aviation".

Line 195: Is the word "summer" missing from this sentence . . . "The model, however, has a maximum in [summer] and underestimates springtime baseline O3"?

Line 205: Are Travis et al. (2016) conclusions regarding 2011 NEI relevant to a model configuration based on 2005 NEI w/ annual scalars?

Line 248: Clarify that these monthly averages are MDA8 O3 (not hourly)?

Line 367: Move mention of lack of daily variation in emissions to early section?

Line 396: Same as above, maybe mention this earlier in modeling methodology section?

---

## Referee Comment (RC3) · Anonymous Referee #3 · 11 Apr 2018

This paper presents a comprehensive modeling analysis of surface ozone and the various factors that contribute to its variability over the United States. By conducting multiple sensitivity simulation removing various sources for the 2004-2012 period, the authors estimate the influence of different background sources and of U.S. anthropogenic sources on mean surface O3 and high O3 events as a function of region, season, and year.

Two aspects of the paper that I'd like to see more discussion on are listed below:

1) The paper is very detailed with many figures and tables and is one more study on top of a rich set of published work, including by some of the co-authors. The authors

[Figure]

often cite previous work, saying it is consistent with their results, but it would be useful to highlight what are the new key contributions from their specific analysis. What new information did the detailed modeling analysis bring to this prolific field?

2) There isn't much discussion on the causes of the large summer bias over the Eastern US and how this bias affects the interpretation of the results. Discussing this in more detail would strengthen the paper. The authors have one sentence addressing this by referring to the work of Travis et al. (2016) using a more recent version of the GEOS-Chem model. They mention potential errors in anthropogenic NOx emission in the NEI inventory, but Travis et al. use the NEI 2011 inventory while the authors use the NEI 2005 inventory. How different are they? If the NEI NOx inventory is indeed too high, how would that affect the calculation of O3_USA? They mention meteorological factors associated with boundary layer mixing and cloud cover which would affect the vertical distribution of O3, but Travis et al. used different meteorological fields (GEOS-FP) compared to the MERRA fields used by the authors. It is unclear whether these potential explanations apply in this case. If MERRA meteorology is indeed biased, then that would certainly affect the validity of the relative influence of various sources on the "most-biased" days analysis and on the average MDA8 O3 levels. A discussion of this would be valuable.

Minor comments:

Line 154. "Anthropogenic emissions. . . are scaled each year on the basis of economic data". It would be useful to have a bit more discussion on how anthropogenic emissions are scaled over the continental U.S. which uses 2005 as the baseline. By how much do NOx emissions change over the time period of the simulation 2004-2012. Are these scaling factors taken from the NEI trends report (https://www.epa.gov/air-emissions-inventories/air-pollutant-emissions-trends-data) itself or was independent estimate done?

Line 195. "a maximum in and underestimate springtime. . ." is "summer" missing after

maximum?

Line 196. While the authors talk about potential causes for the springtime underestimate (stratospheric intrusions), they do not talk about the summertime overestimate, which is quite large.

---

## Author Comment (AC1) · 12 Jun 2018

Average versus high surface ozone levels over the continental U.S.A.: Model bias, background influences, and interannual variability, Jean Guo et al., ACP, (2018) Please see supplemental document for formatted version of the response

Author response to Reviewer #1 This manuscript presents an attempt to derive information about mean maximum daily 8-hour average (MDA8) O3 in the United States, based on ambient measurements and using the global model GEOS-CHEM. Sensitivity simulations examine different sources that affect the 10 highest O3 events and that affect the 10 days with highest model bias against observations for 2004 to 2012 for

each 10 EPA regions. General comments: The analysis is a valuable contribution to the current understanding of ground level O3 and air quality standard settings. The topic itself is highly relevant and thus will be of interest to the readers of ACP. Discussion of the results and their implications is also scientifically sound and the paper includes comprehensive analyses. However, I feel that the paper tried to cover lots of information, which makes it a bit hard for the reader to follow key conclusions from this study. Thus, I recommend that the paper should be published after addressing the following comments.

General

Some comment about day of week effects and model biases in temperature as they relate to the questions raised in the paper seem warranted. There should be comment dramatic changes in the temperature dependence of ozone over this period coincident with the NOx changes. Those changes should have a day of week variation that might appear in the top 10 days.

We have addressed the biases in temperature by adding a comparison to the Global Historical Climatology Network Global Historical Climatology Network (GHCN) and the Climate Anomaly Monitoring System (CAMS). See Supplemental Table 4 and the associated discussion in the text (lines 363-364): "The model monthly mean temperatures in the model (from the MERRA reanalysis) closely match the observed GHCN+CAMS dataset (Supplemental Table 4)."

The request to investigate day of week effects substantially widens the scope of the paper. The general comments suggest that the manuscript is already covering too much information. We feel that tackling day of week effects and its changes over time is a study unto itself and thus outside the scope of this particular paper.

Specific comments: The authors use terms "Baseline O3" and "U.S. background O3". U.S. background O3 is defined as "the O3 levels that would exist in the absence of U.S. anthropogenic emissions of precursors" and Baseline O3 is defined as "tropospheric O3 concentrations that have a negligible influence from local anthropogenic emissions". They sound the same, don't they? If yes, please be consistent in the text.

These definitions are not the same. Clarification has been added in lines 105-107. "Baseline O3 is a measurable quantity and differs from background O3 in that it contains some influence from U.S. anthropogenic emissions that were not recently emitted but contributed to the global background." We follow here the definitions of Jaffe et al., 2018 which builds on the 2009 National Academies report "Global Sources of Local Pollution", and the HTAP 2010 report (available at www.htap.org).

Page 4, lines 106-109, Please clarify if the authors apply Schnell et al. (2014)'s interpolation procedure or they use their dataset. Schnell et al. (2014) use surface MDA8 O3 measurements from air quality networks for 2000–2009, while this paper analyzes the data from 2004-2012.

Jordan Schnell is a co-author and provided the dataset that we used here. The interpolation procedure for his dataset is described in his 2014 paper; he provided us with the data for the years since 2009. We have edited the citation to reflect a newer paper in which this extended dataset has been used. See lines 118-120: "we use an available $1°$ x $1°$ grid of surface MDA8 O3 measurements that were interpolated from the AQS, CASTNet, and Canadian NAPS networks (Schnell and Prather, 2017)."

A valuable addition would be a statement about the chemistry scheme applied in the version GEOS-Chem at the 2.3 section (GEOS-Chem model simulation). The authors mention issues of isoprene chemistry in last paragraph of Conclusions but a brief description or reference to the specific version of the chemistry should be presented before the last paragraph of the paper.

On lines 158-160 we now state: "We use the standard v9_02 chemical mechanism which includes recycling of isoprene nitrates (Mao et al., 2013) in contrast to the mechanisms used in earlier versions of GEOS-Chem (e.g., Zhang et al., 2014 as discussed in Fiore et al., 2014)."

The last paragraph of page 6 needs elaboration where the authors state the sensitivity simulations. The notations for all model simulations should be mentioned and the description of Table 1 should be modïfied so that the Table is read from top to bottom. We have completely rewritten this section with the intent of improving clarity. See paragraph starting from line 183 ("We first perform a base simulation. . .").

Figure 3: Observed O3 concentrations should be represented in a different color to be more visible (maybe black instead of grey) and I would also suggest to plot the curves as an average for 2004-2012 period with associated error bars. Thanks for this suggestion. We have edited the figure (now Figure 2) to show the curves as an average for 2004-2012 period with associated error bars.

Minor comments: The tables start from Table 2 at the manuscript and Table 1 is referenced at Page 8 for the first time. Please fix ordering of table numbers as they appear in the text. Fixed Page 7, line 195: "a maximum in and" should read "a maximum in summertime and" Thank you. Section has been edited and this sentence was removed.

Author response to Reviewer #2

General:

The paper is very well-written and concerns a topic of considerable interest to air quality planners. However, there are some concerns about the suitability of this particular model configuration to address some of the stated objectives of the paper (lines 76-79), as discussed below. In general, the paper would be improved if there was greater clarity about the potential connections between the findings and possible configuration concerns. The value of the paper would be enhanced if the conclusions section was bolstered with a "next steps" or "considerations" sentence or two that described how such a global model-based sensitivity study could be improved in the future.

We have attempted to strengthen the paper throughout as suggested by the reviewer.

In particular, we added a sentence in the introduction, (lines 77-80) to highlight the key benefit and drawback of using a coarse resolution model: "Though coarse resolution global models such as GEOS-Chem will mix emissions into the same grid cell that may remain separate in the real atmosphere, a global model is necessary to quantify background O3 transported intercontinentally, including that produced via oxidation of methane." We added a sentence in the conclusion (lines 476-481) to emphasize a need to confirm our findings with finer scale models: "Future work with high-resolution models (e.g., at the regional scale, ideally with boundary conditions that include source attributions from a global model) is needed, along with observational evidence, to quantify the extent to which biogenic VOC and NOx contribute to the highest observed O3 levels in the warm season. The importance of temperature sensitive sources like biogenic VOC and NOx emissions to background O3 imply that in a warmer climate, these background influences on O3 will play an even more important role in driving up O3 levels."

In particular, there is concern about the use of a coarse resolution model (2 x 2.5 deg) to investigate contributions of U.S. anthropogenic emissions (O3_USA) given that those contributions originate at scales much smaller than the resolution of the model (i.e., point source emissions, urban area emissions). The paper acknowledges the limitations associated with the coarse modeled resolution in several places (lines 212, 242, 399). The paper may want to revisit these caveats in the conclusion and perhaps provide some thoughts on what alternate global model configurations would be better suited for an analysis of source contributions.

Agreed. Please see above.

Kudos to the authors for providing sufficient detail regarding the performance evaluation to allow readers to interpret the contribution findings in light of the model bias/error. However, the ozone overestimations (3-14 ppb in JJA MDA8 top 10 days by region, even worse for JJA all-day averages) suggest caution should be exercised in overinterpreting the contributions. Based on Figure 5 and the associated analyses,

it appears that the model vastly overestimates ozone on hot days in the late summer, especially in the eastern U.S. (even without consideration of potential additional emissions due to increased power demand on those days). Section 3.3 briefly summarizes potential causes for this overestimation based on similar studies, but it would be valuable if the paper provided more application-specific hypotheses for the underlying cause.

We have added more discussion of the potential for biases in the meteorology (see responses to reviewer #3), as well as in anthropogenic NOx emissions to contribute to the summertime overestimate in the model compared to observations. As our set of sensitivity simulations identifies a potential role for biogenic VOC and soil NOx in contributing to the bias, we have added to the text some discussion calling out the need for better constraints on these biogenic emissions, though we do note that the model nevertheless shows some skill at capturing the observed year-to-year variability, which includes a correlation with O$_3$ produced from natural sources (BVOC and soil NOx), which, like total O$_3$, correlate with temperature (Figure 6). We now state, in lines 441-449 "Our finding that BVOC emissions contribute to the summertime surface O3 biases could reflect poor representation of the emissions (and subsequent oxidation chemistry). Earlier work has noted that MEGAN BVOC emissions are too high over California (Bash et al., 2016), Southeast Texas (Kota et al., 2015), the Ozarks in southern Missouri (Carlton and Baker, 2011), and across much of the U.S.A. (Wang et al., 2017). One recent model study uniformly reduced MEGAN isoprene emissions by 20% (Li et al., ACP 2018), but we did not apply any such scaling here. In regions that are highly NOx-sensitive, additional isoprene should not strongly influence O3, as found over southeast Texas (Kota et al., 2015). While not eliminated entirely, the summertime model bias does lessen in the simulation with BVOC emissions set to zero, suggesting that the O3 bias is indeed exacerbated if BVOC emissions are overestimated in the model."

FYI, along w/ the possible causes from the Travis research, others have raised concerns about MEGAN biogenic VOC estimates (e.g., Bash et al., 2016; Carlton and Baker, 2011; Kota et al., 2015; Wang et al., 2017). Thank you for pointing these out. We added these references (see previous response). One of the more noteworthy findings concerns the modeled trends over the 10-year period (e.g., lines 386-389) where the analysis appears to confirm previous findings that improving trends in U.S. air quality from emissions controls have been tempered by increases in background contributions (and increases in temperature). However, one interesting finding here that could use additional explanation is the regional breakout of this "USB vs. USA" tradeoff. Table 5 suggests that the largest increases in high JJA-day O3_USB concentrations between 2004-2006 and 2010-2012 have occurred in the New England and Mid-Atlantic regions, not the western regions where USB concerns are typically greatest. More explanation of the regional differences in modeled USB trends would be beneficial (e.g., is this just an artifact of the meteorology of the two 3-year periods in these regions).

We agree. The "trends" over such a short period are strongly influenced by fluctuations in temperature. While it may indeed be an 'artifact' of looking at such a short period, it nevertheless suggests that regionally produced background O3 from temperature-sensitive emissions (BVOC and NOx) may grow in importance in the coming decades in light of a warming climate. We have attempted to make this clearer by adding a column to Table 4 that shows the change in temperature between these two 3-year periods in each region. We have edited the accompanying discussion to the main text:

Starting from line 364: "Table 4 shows that regions with O3_USB increases generally experienced rising temperatures over this period, as the 2010-2012 period includes two of the warmest years on record. Figure 6 shows that O3_NAT tracks with... "

In response to a comment from Reviewer 1, we have also added Supplementary Table 4 that evaluates monthly mean model temperatures with the Global Historical Climatology Network.

Given model performance findings, would the authors see value in revising the "2-step" contribution analysis (assessing contributions on high-bias days, then assessing contributions on high/all observed days) to a "3-step" contribution where as an intermediate step you also investigated contributions on top-10 modeled days? This could be valuable presuming that the subset of days would differ from top 10 highest bias days.

Though we did conduct some exploratory analysis using this 3-step method early on, we did not end up pursuing this method in the paper because the highest model days are less relevant to the "real world" and if this method were used throughout our paper, the number of figures would have doubled. As the paper is already lengthy, we choose to focus on the days in the observations when the O3 NAAQS is most likely to be exceeded.

We have, however added text that clarifies the extent to which there is overlap between the highest 10 days in the model and the 10 days with the highest biases: "There is at most a 2-6 day overlap between the top 10 O3_Base days and the top 10 most biased days in 2004-2012 across all regions, but during most years, the overlap is around 0-2 days. We restrict our analysis to examining the top 10 observed O3 days as these days are most relevant from a policy perspective." (lines 137-139).

Rather than lumping the Mount Bachelor observations (and subsequent pairs) with surface sites in Region 10, it would be interesting to see how model contributions varied as functions of model performance and observation concentration as a standalone site.

Thanks for this suggestion. We now include a more detailed analysis of Mount Bachelor as a separate standalone section (Section 3.2) that includes new figures (Figure 2 and supplemental figure 4).

Specific:

Line 86: "download" should be "downloaded". FIXED

Lines 124-127: Would be easier to read, if a new sentence was started w/ "On the days

with . . .". FIXED Line 146: "Avian" should be "Aviation". FIXED

Line 195: Is the word "summer" missing from this sentence . . . "The model, however, has a maximum in [summer] and underestimates springtime baseline O3"? Thank you. Section has been edited and this sentence was removed.

Line 205: Are Travis et al. (2016) conclusions regarding 2011 NEI relevant to a model configuration based on 2005 NEI w/ annual scalars?

We add a comparison with Travis et al. (2016) in lines 226-230: "Travis et al. (2016) find that the 3.5 Tg N y-1 NEI 2011 estimate for U.S. fuel NOx emissions is too high and contributes to excessive surface O3. Our simulations include even higher U.S. fuel NOx emissions of 4.4 Tg N y-1 during 2010-2012 (Supplemental Table 3), implying that some portion of the model O3 bias reflects excessively high anthropogenic NOx emissions (Travis et al., 2016)."

Line 248: Clarify that these monthly averages are MDA8 O3 (not hourly)? FIXED

Line 367: Move mention of lack of daily variation in emissions to early section? Done. Now at lines 179-183)

Line 396: Same as above, maybe mention this earlier in modeling methodology section? See lines 179-183)

Author response to Reviewer #3

This paper presents a comprehensive modeling analysis of surface ozone and the various factors that contribute to its variability over the United States. By conducting multiple sensitivity simulation removing various sources for the 2004-2012 period, the authors estimate the influence of different background sources and of U.S. anthropogenic sources on mean surface O3 and high O3 events as a function of region, season, and year.

Two aspects of the paper that I'd like to see more discussion on are listed below:

[Figure]

1) The paper is very detailed with many figures and tables and is one more study on top of a rich set of published work, including by some of the co-authors. The authors often cite previous work, saying it is consistent with their results, but it would be useful to highlight what are the new key contributions from their specific analysis. What new information did the detailed modeling analysis bring to this prolific field? We added text with the intent of providing stronger motivation to the introduction in which we highlight the use of sensitivity simulations to help us identify which sources contribute most to the summertime bias and to the highest O3 days (lines 70-87). To our knowledge, the finding that increasing O3 production from temperature-sensitive biogenic emissions might be offsetting some of the gains achieved by reducing anthropogenic ozone precursor emissions is new, and potentially of growing importance as record-setting warm years have been increasing. We believe that our finding that the summertime bias is associated with regionally produced ozone – including both U.S. anthropogenic and components of U.S. background – rather than transported background (either internationally or intercontinentally) is also new. We have also rewritten the conclusions to emphasize these points.

2) There isn't much discussion on the causes of the large summer bias over the Eastern US and how this bias affects the interpretation of the results.

To our knowledge, prior studies have not used such a broad set of sensitivity simulations to interpret which sources are contributing most in places and times when the model is most biased against observations. Section 3.3 in the submitted paper is entirely devoted to addressing this point. We thus assume that the reviewer is instead driving at the deeper question of the specific causes of the bias, beyond what we can identify cleanly with the sensitivity simulations. We have added additional discussion in response to reviewer 2 that attempts to address both the causes and how it affects the interpretation of the results.

Specifically, we added

1) a sentence in the introduction, (lines 77-80) to highlight the key benefit and drawback of using a coarse resolution model: "Though coarse resolution global models such as GEOS-Chem will mix emissions into the same grid cell that may remain separate in the real atmosphere, a global model is necessary to quantify background O3 transported intercontinentally, including that produced via oxidation of methane."

2) We also added a sentence in the conclusion (lines 476-481) to emphasize a need to confirm our findings with finer scale models: "Future work with high-resolution models (e.g., at the regional scale, ideally with boundary conditions that include source attributions from a global model) is needed, along with observational evidence, to quantify the extent to which biogenic VOC and NOx contribute to the highest observed O3 levels in the warm season. The importance of temperature sensitive sources like biogenic VOC and NOx emissions to background O3 imply that in a warmer climate, these background influences on O3 will play an even more important role in driving up OÂň3 levels."

3) We now state, in lines 441-449 "Our finding that BVOC emissions contribute to the summertime surface O3 biases could reflect poor representation of the emissions (and subsequent oxidation chemistry). Earlier work has noted that MEGAN BVOC emissions are too high over California (Bash et al., 2016), Southeast Texas (Kota et al., 2015), the Ozarks in southern Missouri (Carlton and Baker, 2011), and across much of the U.S.A. (Wang et al., 2017). One recent model study uniformly reduced MEGAN isoprene emissions by 20% (Li et al., ACP 2018), but we did not apply any such scaling here. In regions that are highly NOx-sensitive, additional isoprene should not strongly influence O3, as found over southeast Texas (Kota et al., 2015). While not eliminated entirely, the summertime model bias does lessen in the simulation with BVOC emissions set to zero, suggesting that the O3 bias is indeed exacerbated if BVOC emissions are overestimated in the model."

Discussing this in more detail would strengthen the paper. The authors have one sentence addressing this by referring to the work of Travis et al. (2016) using a more

recent version of the GEOS-Chem model. They mention potential errors in anthropogenic NOx emission in the NEI inventory, but Travis et al. use the NEI 2011 inventory while the authors use the NEI 2005 inventory. How different are they? If the NEI NOx inventory is indeed too high, how would that affect the calculation of O3_USA?

We now directly compare our NOx emissions to those used in Travis et al., 2016 and include a supplementary table providing the NOx emissions applied in each year within the U.S.A. and globally.

Lines 226-230: "Travis et al. (2016) find that the 3.5 Tg N y-1 NEI 2011 estimate for U.S. fuel NOx emissions is too high and contributes to excessive surface O3. Our simulations include even higher U.S. fuel NOx emissions of 4.4 Tg N y-1 during 2010-2012 (Supplemental Table 3), implying that some portion of the model O3 bias reflects excessively high anthropogenic NOx emissions (Travis et al., 2016)." They mention meteorological factors associated with boundary layer mixing and cloud cover which would affect the vertical distribution of O3, but Travis et al. used different meteorological fields (GEOS-FP) compared to the MERRA fields used by the authors. It is unclear whether these potential explanations apply in this case. If MERRA meteorology is indeed biased, then that would certainly affect the validity of the relative influence of various sources on the "most-biased" days analysis and on the average MDA8 O3 levels. A discussion of this would be valuable.

Thanks for this suggestion. We have attempted to address this point by including more discussion of published evaluations of MERRA meteorology:

1) Lines 152-157: "MERRA meteorology captures summer mean surface temperatures to within 1-2 K across U.S. regions and precipitation to within 0.5 mm d-1 except for over the Northern Great Plains where a positive bias exceeds 1 mm d-1, but the variance in summer mean precipitation is lower than observed in some regions (Bosilovich, 2013). While interannual variability in cloudiness observed at weather stations is largely captured by MERRA, the reanalysis generally underestimates cloud cover and thus overestimates observed downward surface shortwave fluxes (Free et al., 2016)."

2) Lines 230-232: "The low bias in cloud cover in the MERRA meteorology and associated overestimate in downward shortwave surface radiation (Free et al., 2016) may also contribute to excessive O3 production in the model."

3) We also added our own evaluation of surface temperature over the U.S.A. in the MERRA fields (Supplemental Table 4).

Minor comments:

Line 154. "Anthropogenic emissions. . . are scaled each year on the basis of economic data". It would be useful to have a bit more discussion on how anthropogenic emissions are scaled over the continental U.S. which uses 2005 as the baseline. By how much do NOx emissions change over the time period of the simulation 2004-2012.

Supplemental Table 3 was added to provide the NOx emissions within each year, both globally and within the U.S.A. (Lines 178-183)

Are these scaling factors taken from the NEI trends report (https://www.epa.gov/air-emissions-inventories/air-pollutant-emissions-trends-data) itself or was independent estimate done?

The van Donkelaar et al., 2008 describes the standard GEOS-Chem emissions scaling reference. The scale factors use government statistics where available.

Edited this sentence (lines 169-170) to include "provided by individual countries, where available" Line 195. "a maximum in and underestimate springtime. . ." is "summer" missing after maximum?

Yes. Thank you

Line 196. While the authors talk about potential causes for the springtime underestimate (stratospheric intrusions), they do not talk about the summertime overestimate, which is quite large.

[Figure]

See response to general comments above and our additions above regarding anthropogenic NOx emissions and citations of prior work evaluating MERRA meteorology (temperature, precipitation and cloud cover).

Please also note the supplement to this comment:
https://www.atmos-chem-phys-discuss.net/acp-2018-115/acp-2018-115-AC1-supplement.pdf

[Figure]

a)

**Mount Bachelor (2.7 km)**

b)

**Mount Bachelor (at surface)**

Legend (panel a):
- MBO (24hr)
- O$_3$_Base (24hr at 2.7km)
- O$_3$_USB (24hr at 2.7km)
- O$_3$_NAT (24hr at 2.7km)
- O$_3$_ICT+CH$_4$ (24hr at 2.7km)
- O$_3$_USA (24hr at 2.7km)

Legend (panel b):
- MBO (24hr)
- O$_3$_Base (24hr at surface)
- O$_3$_USB (24hr at surface)
- O$_3$_NAT (24hr at surface)
- O$_3$_ICT+CH$_4$ (24hr at surface)
- O$_3$_USA (24hr at surface)

Axis labels: Concentration (ppb) vs Month (Jan Feb Mar Apr May Jun Jul Aug Sep Oct Nov Dec)

**Fig. 1.** Monthly 2004-2012 average 24-hour O3 concentrations at Mount Bachelor Observatory. Observations (grey) are the same in both panels. Simulations from the GEOS-Chem model are sampled in the grid cell co

[Figure]

**Fig. 2.** The three 4th highest days in each year (solid dots) that went into the calculation of the three-year average of the 4th highest MDA8 O3 day (hollow diamond). Error bars show the range between the hig

[Figure]

**Fig. 3.** Monthly average of observed (a) daily 24-hour and (b) MDA8 O3 concentrations averaged across 2004-2012 at Mount Bachelor Observatory. Black line shows the average of each month from 2004-2012. Error b

[Figure]

Summary information for each region.

**Fig. 4.** Summary information for each region showing the three 4th highest days in each year (solid dots) that went into the calculation of the three-year average of the 4th highest MDA8 O3 day (hollow diamond

| | Emissions | 2004 | 2005 | 2006 | 2007 | 2008 | 2009 | 2010 | 2011 | 2012 |
|---|---|---|---|---|---|---|---|---|---|---|
| Global | Anthropogenic NO with biofuels (Tg N) | 30.3 | 30.2 | 30.1 | 29.9 | 29.5 | 29.0 | 28.8 | 28.8 | 28.8 |
| | Biomass burning (Tg N) | 4.5 | 4.7 | 4.6 | 4.6 | 3.9 | 3.5 | 5.0 | 3.7 | 3.7 |
| | Soil (Tg N) | 9.0 | 9.1 | 8.8 | 8.5 | 8.4 | 8.6 | 8.4 | 8.6 | 9.2 |
| | Lightning (Tg N) | 5.5 | 6.1 | 6.2 | 6.4 | 6.9 | 7.3 | 7.2 | 7.1 | 7.2 |
| | Isoprene (Tg C) | 493.0 | 499.3 | 471.5 | 453.6 | 435.3 | 455.4 | 466.0 | 453.3 | 467.3 |
| US | Anthropogenic NO with biofuels (Tg N) | 6.32 | 6.04 | 5.75 | 5.44 | 5.13 | 4.63 | 4.36 | 4.36 | 4.36 |
| | Biomass burning (Tg N) | 0.02 | 0.06 | 0.06 | 0.07 | 0.04 | 0.04 | 0.05 | 0.12 | 0.12 |
| | Soil (Tg N) | 0.78 | 0.86 | 1.02 | 0.92 | 0.82 | 0.79 | 0.77 | 0.95 | 1.10 |
| | Lightning (Tg N) | 0.86 | 0.86 | 0.77 | 0.75 | 1.10 | 1.13 | 1.13 | 1.28 | 1.31 |
| | Isoprene (Tg C) | 18.1 | 21.5 | 21.9 | 22.0 | 19.3 | 18.3 | 20.2 | 22.0 | 22.4 |

**Fig. 5.** Global and US emissions totals for 2004-2012.

| | Model Temperature (C) | | | GHCN Temperature (C) | | | Model Temp. Bias | | |
|---|---|---|---|---|---|---|---|---|---|
| **Region** | **MAM** | **JJA** | **SON** | **MAM** | **JJA** | **SON** | **MAM** | **JJA** | **SON** |
| New England | 7 | 19 | 11 | 8 | 20 | 11 | 0 | -1 | 1 |
| NY+NJ | 9 | 21 | 12 | 9 | 21 | 12 | 0 | 0 | 1 |
| Mid-Atlantic | 12 | 23 | 14 | 12 | 23 | 14 | 0 | 0 | 0 |
| Southeast | 17 | 26 | 18 | 17 | 26 | 18 | 0 | 0 | 0 |
| Midwest | 10 | 22 | 12 | 10 | 22 | 12 | 0 | 0 | 0 |
| South Central | 18 | 27 | 19 | 19 | 28 | 20 | -1 | -1 | -1 |
| Plains | 13 | 25 | 13 | 13 | 25 | 13 | 0 | 0 | 0 |
| Mountains + Plains | 7 | 20 | 9 | 7 | 19 | 8 | 0 | 1 | 1 |
| Pacific SW | 14 | 23 | 17 | 14 | 22 | 17 | -1 | 0 | 0 |
| Pacific NW | 8 | 17 | 10 | 8 | 17 | 9 | 0 | 0 | 1 |

**Fig. 6.** Monthly average temperature across all days in each season (average of 2004-2012) in (1) GEOS-Chem, in (2) the Global Historical Climatology Network (GHCN) and the Climate Anomaly Monitoring System (C

| Region | Correlation | |
| --- | --- | --- |
| | $O_3$ Base and $O_3$ USB | $O_3$ Base and $O_3$ USA |
| New England | 0.28 | 0.64 |
| NY+NJ | 0.50 | 0.58 |
| Mid-Atlantic | 0.54 | 0.70 |
| Southeast | 0.66 | 0.59 |
| Midwest | 0.75 | 0.76 |
| South Central | 0.71 | 0.72 |
| Plains | 0.80 | 0.75 |
| Mountains + Plains | 0.95 | 0.64 |
| Pacific SW | 0.72 | 0.28 |
| Pacific NW | 0.98 | 0.05 |

**Fig. 7.** Correlation between (1) O3_Base and O3_USB and (2) O3_Base and O3_USA on the average of O3_top10obs_JJA days from 2004-2012 in each region.

|  | Obs | O$_3$_Base | O$_3$_USB | O$_3$_USA | Temperature (C) |
|---|---|---|---|---|---|
| New England | -6 | -4 | 6 | -10 | 2 |
| NY+NJ | -2 | -4 | 3 | -7 | 1 |
| Mid-Atlantic | 0 | -3 | 4 | -7 | 1 |
| Southeast | -4 | -5 | 2 | -7 | 1 |
| Midwest | -2 | -4 | 2 | -6 | 0 |
| South Central | -6 | -2 | 5 | -7 | 1 |
| Plains | -1 | -2 | 4 | -5 | 1 |
| Mountains + Plains | -4 | -1 | 1 | -2 | -1 |
| Pacific SW | -3 | -4 | 0 | -4 | -1 |
| Pacific NW | -7 | -5 | -4 | -1 | -1 |
| *Average* | *-3* | *-3* | *2* | *-6* | *0* |

**Fig. 8.** Change in MDA8 O3 concentrations from 2004-2006 to 2010-2012 on O3_top10obs_JJA days in the observations, O3_Base, O3_USB, O3_USA, and temperature.

| Region | Range | Obs | O$_3$_Base | O$_3$_USB |
|---|---|---|---|---|
| New England | 4th highest day | 15 | 16 | 10 |
| | 3-year average 4th highest day | 9 | 10 | 3 |
| | *Difference* | -6 | -6 | -7 |
| NY+NJ | 4th highest day | 11 | 10 | 12 |
| | 3-year average 4th highest day | 6 | 2 | 6 |
| | *Difference* | -5 | -8 | -6 |
| Mid-Atlantic | 4th highest day | 13 | 36 | 25 |
| | 3-year average 4th highest day | 7 | 21 | 10 |
| | *Difference* | -6 | -15 | -15 |
| Southeast | 4th highest day | 9 | 24 | 10 |
| | 3-year average 4th highest day | 6 | 9 | 4 |
| | *Difference* | -3 | -15 | -7 |
| Midwest | 4th highest day | 13 | 22 | 24 |
| | 3-year average 4th highest day | 8 | 11 | 10 |
| | *Difference* | -6 | -11 | -14 |
| South Central | 4th highest day | 11 | 26 | 22 |
| | 3-year average 4th highest day | 8 | 13 | 13 |
| | *Difference* | -3 | -13 | -9 |
| Plains | 4th highest day | 14 | 32 | 24 |
| | 3-year average 4th highest day | 9 | 18 | 11 |
| | *Difference* | -5 | -15 | -13 |
| Mountains + Plains | 4th highest day | 9 | 23 | 20 |
| | 3-year average 4th highest day | 6 | 13 | 13 |
| | *Difference* | -2 | -10 | -7 |
| Pacific SW | 4th highest day | 5 | 23 | 20 |
| | 3-year average 4th highest day | 3 | 5 | 5 |
| | *Difference* | -2 | -18 | -15 |
| Pacific NW | 4th highest day | 11 | 14 | 15 |
| | 3-year average 4th highest day | 5 | 9 | 12 |
| | *Difference* | -5 | -5 | -3 |

**Fig. 9.** Summary information for each region. The first row next to each region reports the range across 2004-2012 of the 4th highest values from each of the 9 individual years for the observations, O3_Base, a

**Supplement:**

*Average versus high surface ozone levels over the continental U.S.A.: Model bias, background influences, and interannual variability*, **Jean Guo et al., ACP, (2018)**

*Authors' response is written in bold type; Reviewer comment in normal type. Figures, including new ones added to the paper that were not in direct response to reviewer's comments are at the end.*

**Author response to Reviewer #1**

This manuscript presents an attempt to derive information about mean maximum daily 8-hour average (MDA8) $O_3$ in the United States, based on ambient measurements and using the global model GEOS-CHEM. Sensitivity simulations examine different sources that affect the 10 highest $O_3$ events and that affect the 10 days with highest model bias against observations for 2004 to 2012 for each 10 EPA regions.

General comments: The analysis is a valuable contribution to the current understanding of ground level $O_3$ and air quality standard settings. The topic itself is highly relevant and thus will be of interest to the readers of ACP. Discussion of the results and their implications is also scientifically sound and the paper includes comprehensive analyses. However, I feel that the paper tried to cover lots of information, which makes it a bit hard for the reader to follow key conclusions from this study. Thus, I recommend that the paper should be published after addressing the following comments.

**General**

Some comment about day of week effects and model biases in temperature as they relate to the questions raised in the paper seem warranted. There should be comment dramatic changes in the temperature dependence of ozone over this period coincident with the NOx changes. Those changes should have a day of week variation that might appear in the top 10 days.

**We have addressed the biases in temperature by adding a comparison to the Global Historical Climatology Network Global Historical Climatology Network (GHCN) and the Climate Anomaly Monitoring System (CAMS). See Supplemental Table 4 and the associated discussion in the text (lines 363-364):** *"The model monthly mean temperatures in the model (from the MERRA reanalysis) closely match the observed GHCN+CAMS dataset (Supplemental Table 1)."*

**The request to investigate day of week effects substantially widens the scope of the paper. The general comments suggest that the manuscript is already covering too much information. We feel that tackling day of week effects and its changes over time is a study unto itself and thus outside the scope of this particular paper.**

Specific comments: The authors use terms "Baseline $O_3$" and "U.S. background $O_3$". U.S. background $O_3$ is defined as "the $O_3$ levels that would exist in the absence of U.S. anthropogenic emissions of precursors" and Baseline $O_3$ is defined as "tropospheric $O_3$ concentrations that have a negligible influence from local anthropogenic emissions". They sound the same, don't they? If yes, please be consistent in the text.

**These definitions are not the same. Clarification has been added in lines 105-107.** *"Baseline $O_3$ is a measurable quantity and differs from background $O_3$ in that it contains some influence from U.S. anthropogenic emissions that were not recently emitted but contributed to the global background."* **We follow here the definitions of Jaffe et al., 2018 which builds on the 2009 National Academies report "Global Sources of Local Pollution", and the HTAP 2010 report (available at www.htap.org).**

Page 4, lines 106-109, Please clarify if the authors apply Schnell et al. (2014)'s interpolation procedure or they use their dataset. Schnell et al. (2014) use surface MDA8 $O_3$ measurements from air quality networks for 2000–2009, while this paper analyzes the data from 2004-2012.

**Jordan Schnell is a co-author and provided the dataset that we used here. The interpolation procedure for his dataset is described in his 2014 paper; he provided us with the data for the years since 2009. We have edited the citation to reflect a newer paper in which this extended dataset has been used. See lines 118-120:** *"we use an available $1^\bullet$ x $1^\bullet$ grid of surface MDA8 $O_3$ measurements that were interpolated from the AQS, CASTNet, and Canadian NAPS networks (Schnell and Prather, 2017)."*

A valuable addition would be a statement about the chemistry scheme applied in the version GEOS-Chem at the 2.3 section (GEOS-Chem model simulation). The authors mention issues of isoprene chemistry in last paragraph of Conclusions but a brief description or reference to the specific version of the chemistry should be presented before the last paragraph of the paper.

**On lines 158-160 we now state:** *"We use the standard v9_02 chemical mechanism which includes recycling of isoprene nitrates (Mao et al., 2013) in contrast to the mechanisms used in earlier versions of GEOS-Chem (e.g., Zhang et al., 2014 as discussed in Fiore et al., 2014)."*

The last paragraph of page 6 needs elaboration where the authors state the sensitivity simulations. The notations for all model simulations should be mentioned and the description of Table 1 should be modified so that the Table is read from top to bottom. **We have completely rewritten this section with the intent of improving clarity. See paragraph starting from line 183** *("We first perform a base simulation…").*

Figure 3: Observed $O_3$ concentrations should be represented in a different color to be more visible (maybe black instead of grey) and I would also suggest to plot the curves as an average for 2004-2012 period with associated error bars. **Thanks for this suggestion. We have edited the figure to show the curves as an average for 2004-2012 period with associated error bars.**

Minor comments: The tables start from Table 2 at the manuscript and Table 1 is referenced at Page 8 for the first time. Please fix ordering of table numbers as they appear in the text. **Fixed**

Page 7, line 195: "a maximum in and" should read "a maximum in summertime and" **Thank you. Section has been edited and this sentence was removed.**

Author response to Reviewer #2

General:

The paper is very well-written and concerns a topic of considerable interest to air quality planners. However, there are some concerns about the suitability of this particular model configuration to address some of the stated objectives of the paper (lines 76-79), as discussed below. In general, the paper would be improved if there was greater clarity about the potential connections between the findings and possible configuration concerns. The value of the paper would be enhanced if the conclusions section was bolstered with a "next steps" or "considerations" sentence or two that described how such a global model-based sensitivity study could be improved in the future.

**We have attempted to strengthen the paper throughout as suggested by the reviewer. In particular, we added a sentence in the introduction, (lines 77-80) to highlight the key benefit and drawback of using a coarse resolution model:** *"Though coarse resolution global models such as GEOS-Chem will mix emissions into the same grid cell that may remain separate in the real atmosphere, a global model is necessary to quantify background $O_3$ transported intercontinentally, including that produced via oxidation of methane."*

**We added a sentence in the conclusion (lines 476-481) to emphasize a need to confirm our findings with finer scale models:** *"Future work with high-resolution models (e.g., at the regional scale, ideally with boundary conditions that include source attributions from a global model) is needed, along with observational evidence, to quantify the extent to which biogenic VOC and NOx contribute to the highest observed $O_3$ levels in the warm season. The importance of temperature sensitive sources like biogenic VOC and NOx emissions to background $O_3$ imply that in a warmer climate, these background influences on $O_3$ will play an even more important role in driving up $O_3$ levels."*

In particular, there is concern about the use of a coarse resolution model (2 x 2.5 deg) to investigate contributions of U.S. anthropogenic emissions ($O_3$_USA) given that those contributions originate at scales much smaller than the resolution of the model (i.e., point source emissions, urban area emissions). The paper acknowledges the limitations associated with the coarse modeled resolution in several places (lines 212, 242, 399). The paper may want to revisit these caveats in the conclusion and perhaps provide some thoughts on what alternate global model configurations would be better suited for an analysis of source contributions.

**Agreed. Please see above.**

Kudos to the authors for providing sufficient detail regarding the performance evaluation to allow readers to interpret the contribution findings in light of the model bias/error. However, the ozone overestimations (3-14 ppb in JJA MDA8 top 10 days by region, even worse for JJA all-day averages) suggest caution should be exercised in overinterpreting the contributions. Based on Figure 5 and the associated analyses, it appears that the model vastly overestimates ozone on hot days in the late summer, especially in the eastern U.S. (even without consideration of potential additional emissions due to increased power demand on those days). Section 3.3 briefly

summarizes potential causes for this overestimation based on similar studies, but it would be valuable if the paper provided more application-specific hypotheses for the underlying cause.

**We have added more discussion of the potential for biases in the meteorology (see responses to reviewer #3), as well as in anthropogenic $NO_x$ emissions to contribute to the summertime overestimate in the model compared to observations. As our set of sensitivity simulations identifies a potential role for biogenic VOC and soil $NO_x$ in contributing to the bias, we have added to the text some discussion calling out the need for better constraints on these biogenic emissions, though we do note that the model nevertheless shows some skill at capturing the observed year-to-year variability, which includes a correlation with $O_3$ produced from natural sources (BVOC and soil $NO_x$), which, like total $O_3$, correlate with temperature (Figure 6).**

**We now state, in lines 441-449** *"Our finding that BVOC emissions contribute to the summertime surface $O_3$ biases could reflect poor representation of the emissions (and subsequent oxidation chemistry). Earlier work has noted that MEGAN BVOC emissions are too high over California (Bash et al., 2016), Southeast Texas (Kota et al., 2015), the Ozarks in southern Missouri (Carlton and Baker, 2011), and across much of the U.S.A. (Wang et al., 2017). One recent model study uniformly reduced MEGAN isoprene emissions by 20% (Li et al., ACP 2018), but we did not apply any such scaling here. In regions that are highly NOx-sensitive, additional isoprene should not strongly influence $O_3$, as found over southeast Texas (Kota et al., 2015). While not eliminated entirely, the summertime model bias does lessen in the simulation with BVOC emissions set to zero, suggesting that the $O_3$ bias is indeed exacerbated if BVOC emissions are overestimated in the model."*

FYI, along w/ the possible causes from the Travis research, others have raised concerns about MEGAN biogenic VOC estimates (e.g., Bash et al., 2016; Carlton and Baker, 2011; Kota et al., 2015; Wang et al., 2017). **Thank you for pointing these out. We added these references (see previous response).**

One of the more noteworthy findings concerns the modeled trends over the 10-year period (e.g., lines 386-389) where the analysis appears to confirm previous findings that improving trends in U.S. air quality from emissions controls have been tempered by increases in background contributions (and increases in temperature). However, one interesting finding here that could use additional explanation is the regional breakout of this "USB vs. USA" tradeoff. Table 5 suggests that the largest increases in high JJA-day $O_3\_USB$ concentrations between 2004-2006 and 2010-2012 have occurred in the New England and Mid-Atlantic regions, not the western regions where USB concerns are typically greatest. More explanation of the regional differences in modeled USB trends would be beneficial (e.g., is this just an artifact of the meteorology of the two 3-year periods in these regions).

**We agree. The "trends" over such a short period are strongly influenced by fluctuations in temperature. While it may indeed be an 'artifact' of looking at such a short period, it nevertheless suggests that regionally produced background $O_3$ from temperature-sensitive emissions (BVOC and $NO_x$) may grow in importance in the coming decades in light of a**

**warming climate. We have attempted to make this clearer by adding a column to Table 4 that shows the change in temperature between these two 3-year periods in each region. We have edited the accompanying discussion to the main text:**

**Starting from line 364:** *"Table 4 shows that regions with O$_3$_USB increases generally experienced rising temperatures over this period, as the 2010-2012 period includes two of the warmest years on record. Figure 6 shows that O$_3$_NAT tracks with…"*

**In response to a comment from Reviewer 1, we have also added Supplementary Table 4 that evaluates monthly mean model temperatures with the Global Historical Climatology Network.**

Given model performance findings, would the authors see value in revising the "2-step" contribution analysis (assessing contributions on high-bias days, then assessing contributions on high/all observed days) to a "3-step" contribution where as an intermediate step you also investigated contributions on top-10 modeled days? This could be valuable presuming that the subset of days would differ from top 10 highest bias days.

**Though we did conduct some exploratory analysis using this 3-step method early on, we did not end up pursuing this method in the paper because the highest model days are less relevant to the "real world" and if this method were used throughout our paper, the number of figures would have doubled. As the paper is already lengthy, we choose to focus on the days in the observations when the O$_3$ NAAQS is most likely to be exceeded.**

**We have, however added text that clarifies the extent to which there is overlap between the highest 10 days in the model and the 10 days with the highest biases:** *"There is at most a 2-6 day overlap between the top 10 O$_3$_Base days and the top 10 most biased days in 2004-2012 across all regions, but during most years, the overlap is around 0-2 days. We restrict our analysis to examining the top 10 observed O$_3$ days as these days are most relevant from a policy perspective."* **(lines 137-139).**

Rather than lumping the Mount Bachelor observations (and subsequent pairs) with surface sites in Region 10, it would be interesting to see how model contributions varied as functions of model performance and observation concentration as a standalone site.

**Thanks for this suggestion. We now include a more detailed analysis of Mount Bachelor as a separate standalone section (Section 3.2) that includes new figures (Figure 3 and Supplemental Figure 4).**

**Specific:**

Line 86: "download" should be "downloaded". **FIXED**

Lines 124-127: Would be easier to read, if a new sentence was started w/ "On the days with . . .". **FIXED**

Line 146: "Avian" should be "Aviation". **FIXED**

Line 195: Is the word "summer" missing from this sentence . . . "The model, however, has a maximum in [summer] and underestimates springtime baseline $O_3$"? **Thank you. Section has been edited and this sentence was removed**

Line 205: Are Travis et al. (2016) conclusions regarding 2011 NEI relevant to a model configuration based on 2005 NEI w/ annual scalars?

**We add a comparison with Travis et al. (2016) in lines 226-230:** *"Travis et al. (2016) find that the 3.5 Tg N y-1 NEI 2011 estimate for U.S. fuel NOx emissions is too high and contributes to excessive surface $O_3$. Our simulations include even higher U.S. fuel NOx emissions of 4.4 Tg N y-1 during 2010-2012 (Supplemental Table 3), implying that some portion of the model $O_3$ bias reflects excessively high anthropogenic NOx emissions (Travis et al., 2016)."*

Line 248: Clarify that these monthly averages are MDA8 $O_3$ (not hourly)? **FIXED**

Line 367: Move mention of lack of daily variation in emissions to early section? **Done. Now at lines 179-183)**

Line 396: Same as above, maybe mention this earlier in modeling methodology section? **See lines 179-183)**

**Author response to Reviewer #3**

This paper presents a comprehensive modeling analysis of surface ozone and the various factors that contribute to its variability over the United States. By conducting multiple sensitivity simulation removing various sources for the 2004-2012 period, the authors estimate the influence of different background sources and of U.S. anthropogenic sources on mean surface $O_3$ and high $O_3$ events as a function of region, season, and year.

Two aspects of the paper that I'd like to see more discussion on are listed below:

1) The paper is very detailed with many figures and tables and is one more study on top of a rich set of published work, including by some of the co-authors. The authors often cite previous work, saying it is consistent with their results, but it would be useful to highlight what are the new key contributions from their specific analysis. What new information did the detailed modeling analysis bring to this prolific field?

**We added text with the intent of providing stronger motivation to the introduction in which we highlight the use of sensitivity simulations to help us identify which sources contribute most to the summertime bias and to the highest $O_3$ days (lines 70-87). To our knowledge, the finding that increasing $O_3$ production from temperature-sensitive biogenic emissions might be offsetting some of the gains achieved by reducing anthropogenic ozone precursor emissions is new, and potentially of growing importance as record-setting warm years have been increasing. We believe that our finding that the summertime bias is associated with regionally produced ozone – including both U.S. anthropogenic and components of U.S.**

**background – rather than transported background (either internationally or intercontinentally) is also new. We have also rewritten the conclusions to emphasize these points.**

2) There isn't much discussion on the causes of the large summer bias over the Eastern US and how this bias affects the interpretation of the results.

**To our knowledge, prior studies have not used such a broad set of sensitivity simulations to interpret which sources are contributing most in places and times when the model is most biased against observations. Section 3.3 in the submitted paper is entirely devoted to addressing this point. We thus assume that the reviewer is instead driving at the deeper question of the specific causes of the bias, beyond what we can identify cleanly with the sensitivity simulations. We have added additional discussion in response to reviewer 2 that attempts to address both the causes and how it affects the interpretation of the results.**

**Specifically, we added**

1) **A sentence in the introduction, (lines 77-80) to highlight the key benefit and drawback of using a coarse resolution model:** *"Though coarse resolution global models such as GEOS-Chem will mix emissions into the same grid cell that may remain separate in the real atmosphere, a global model is necessary to quantify background $O_3$ transported intercontinentally, including that produced via oxidation of methane."*

2) **We also added a sentence in the conclusion (lines 476-481) to emphasize a need to confirm our findings with finer scale models:** *"Future work with high-resolution models (e.g., at the regional scale, ideally with boundary conditions that include source attributions from a global model) is needed, along with observational evidence, to quantify the extent to which biogenic VOC and NOx contribute to the highest observed $O_3$ levels in the warm season. The importance of temperature sensitive sources like biogenic VOC and NOx emissions to background $O_3$ imply that in a warmer climate, these background influences on $O_3$ will play an even more important role in driving up $O_3$ levels."*

3) **We now state, in lines 441-449** *"Our finding that BVOC emissions contribute to the summertime surface $O_3$ biases could reflect poor representation of the emissions (and subsequent oxidation chemistry). Earlier work has noted that MEGAN BVOC emissions are too high over California (Bash et al., 2016), Southeast Texas (Kota et al., 2015), the Ozarks in southern Missouri (Carlton and Baker, 2011), and across much of the U.S.A. (Wang et al., 2017). One recent model study uniformly reduced MEGAN isoprene emissions by 20% (Li et al., ACP 2018), but we did not apply any such scaling here. In regions that are highly NOx-sensitive, additional isoprene should not strongly influence $O_3$, as found over southeast Texas (Kota et al., 2015). While not eliminated entirely, the summertime model bias does lessen in the simulation with BVOC emissions set to zero, suggesting that the $O_3$ bias is indeed exacerbated if BVOC emissions are overestimated in the model."*

Discussing this in more detail would strengthen the paper. The authors have one sentence addressing this by referring to the work of Travis et al. (2016) using a more recent version of the GEOS-Chem model. They mention potential errors in anthropogenic NOx emission in the NEI inventory, but Travis et al. use the NEI 2011 inventory while the authors use the NEI 2005 inventory. How different are they? If the NEI NOx inventory is indeed too high, how would that affect the calculation of $O_3$_USA?

**We now directly compare our NOx emissions to those used in Travis et al., 2016 and include a supplementary table (see above – Supplemental Table 3) providing the NOx emissions applied in each year within the U.S.A. and globally.**

**Lines 226-230: "*Travis et al. (2016) find that the 3.5 Tg N y-1 NEI 2011 estimate for U.S. fuel NOx emissions is too high and contributes to excessive surface $O_3$. Our simulations include even higher U.S. fuel NOx emissions of 4.4 Tg N y-1 during 2010-2012 (Supplemental Table 3), implying that some portion of the model $O_3$ bias reflects excessively high anthropogenic NOx emissions (Travis et al., 2016)."***

They mention meteorological factors associated with boundary layer mixing and cloud cover which would affect the vertical distribution of $O_3$, but Travis et al. used different meteorological fields (GEOS-FP) compared to the MERRA fields used by the authors. It is unclear whether these potential explanations apply in this case. If MERRA meteorology is indeed biased, then that would certainly affect the validity of the relative influence of various sources on the "most-biased" days analysis and on the average MDA8 $O_3$ levels. A discussion of this would be valuable.

**Thanks for this suggestion. We have attempted to address this point by including more discussion of published evaluations of MERRA meteorology:**

1) **Lines 152-157: "*MERRA meteorology captures summer mean surface temperatures to within 1-2 K across U.S. regions and precipitation to within 0.5 mm d-1 except for over the Northern Great Plains where a positive bias exceeds 1 mm d-1, but the variance in summer mean precipitation is lower than observed in some regions (Bosilovich, 2013). While interannual variability in cloudiness observed at weather stations is largely captured by MERRA, the reanalysis generally underestimates cloud cover and thus overestimates observed downward surface shortwave fluxes (Free et al., 2016)."***
2) **Lines 230-232: "*The low bias in cloud cover in the MERRA meteorology and associated overestimate in downward shortwave surface radiation (Free et al., 2016) may also contribute to excessive $O_3$ production in the model."***
3) **We also added our own evaluation of surface temperature over the U.S.A. in the MERRA fields (Supplemental Table 4).**

Minor comments:

Line 154. "Anthropogenic emissions. . . are scaled each year on the basis of economic data". It would be useful to have a bit more discussion on how anthropogenic emissions are scaled over

the continental U.S. which uses 2005 as the baseline. By how much do NOx emissions change over the time period of the simulation 2004-2012.

**Supplemental Table 3 was added to provide the NOx emissions within each year, both globally and within the U.S.A. (Lines 178-183)**

Are these scaling factors taken from the NEI trends report (https://www.epa.gov/air-emissions-inventories/air-pollutant-emissions-trends-data) itself or was independent estimate done?

**The van Donkelaar et al., 2008 describes the standard GEOS-Chem emissions scaling reference. The scale factors use government statistics where available.**

**Edited this sentence (lines 169-170) to include "provided by individual countries, where available"**

Line 195. "a maximum in and underestimate springtime. . ." is "summer" missing after maximum? **Yes. Thank you**

Line 196. While the authors talk about potential causes for the springtime underestimate (stratospheric intrusions), they do not talk about the summertime overestimate, which is quite large.

**See response to general comments above and our additions above regarding anthropogenic NO$_x$ emissions and citations of prior work evaluating MERRA meteorology (temperature, precipitation and cloud cover).**

**Figures and tables edited/added in response to reviewer's comments:**

[Figure]

**Figure 1: Monthly 2004-2012 average 24-hour O₃ concentrations at Mount Bachelor Observatory. Observations (grey) are the same in both panels. Simulations from the GEOS-Chem model are sampled in the grid cell containing Mount Bachelor at (a) 2.7 km (the height of the Mount Bachelor Observatory) and at (b) the surface: O₃_Base (blue), O₃_USB (red), O₃_NAT (light green), O₃_ICT+CH₄ (pink), and O₃_USA (dark green). The shaded range spans the highest and lowest years.**

[Figure]

**Supplemental Figure 1: Monthly average of observed (a) daily 24-hour and (b) MDA8 O₃ concentrations averaged across 2004-2012 at Mount Bachelor Observatory. Black line shows the average of each month from 2004-2012. Error bars show the standard deviation in the interannual variability in each month. Dashed lines show the concentrations for each individual year.**

**Table 1: Change in MDA8 O$_3$ concentrations from 2004-2006 to 2010-2012 on O$_3$_top10obs_JJA days in the observations, O$_3$_Base, O$_3$_USB, O$_3$_USA, and temperature.**

|  | Obs | O$_3$_Base | O$_3$_USB | O$_3$_USA | Temperature (C) |
|---|---|---|---|---|---|
| New England | -6 | -4 | 6 | -10 | 2 |
| NY+NJ | -2 | -4 | 3 | -7 | 1 |
| Mid-Atlantic | 0 | -3 | 4 | -7 | 1 |
| Southeast | -4 | -5 | 2 | -7 | 1 |
| Midwest | -2 | -4 | 2 | -6 | 0 |
| South Central | -6 | -2 | 5 | -7 | 1 |
| Plains | -1 | -2 | 4 | -5 | 1 |
| Mountains + Plains | -4 | -1 | 1 | -2 | -1 |
| Pacific SW | -3 | -4 | 0 | -4 | -1 |
| Pacific NW | -7 | -5 | -4 | -1 | -1 |
| *Average* | *-3* | *-3* | *2* | *-6* | *0* |

**Supplemental Table 1: Monthly average temperature across all days in each season (average of 2004-2012) in (1) GEOS-Chem, in (2) the Global Historical Climatology Network (GHCN) and the Climate Anomaly Monitoring System (CAMS) (in degrees C), and (3) the difference between these values.**

|  | Model Temperature (C) | | | GHCN+CAMS Temperature (C) | | | Model Temp. Bias | | |
|---|---|---|---|---|---|---|---|---|---|
| *Region* | *MAM* | *JJA* | *SON* | *MAM* | *JJA* | *SON* | *MAM* | *JJA* | *SON* |
| New England | 7 | 19 | 11 | 8 | 20 | 11 | 0 | -1 | 1 |
| NY+NJ | 9 | 21 | 12 | 9 | 21 | 12 | 0 | 0 | 1 |
| Mid-Atlantic | 12 | 23 | 14 | 12 | 23 | 14 | 0 | 0 | 0 |
| Southeast | 17 | 26 | 18 | 17 | 26 | 18 | 0 | 0 | 0 |
| Midwest | 10 | 22 | 12 | 10 | 22 | 12 | 0 | 0 | 0 |
| South Central | 18 | 27 | 19 | 19 | 28 | 20 | -1 | -1 | -1 |
| Plains | 13 | 25 | 13 | 13 | 25 | 13 | 0 | 0 | 0 |
| Mountains + Plains | 7 | 20 | 9 | 7 | 19 | 8 | 0 | 1 | 1 |
| Pacific SW | 14 | 23 | 17 | 14 | 22 | 17 | -1 | 0 | 0 |
| Pacific NW | 8 | 17 | 10 | 8 | 17 | 9 | 0 | 0 | 1 |

**Supplemental Table 2: Global and US emissions totals for 2004-2012.**

| | Emissions | 2004 | 2005 | 2006 | 2007 | 2008 | 2009 | 2010 | 2011 | 2012 |
|---|---|---|---|---|---|---|---|---|---|---|
| Global | Anthropogenic NO with biofuels (Tg N) | 30.3 | 30.2 | 30.1 | 29.9 | 29.5 | 29.0 | 28.8 | 28.8 | 28.8 |
| | Biomass burning (Tg N) | 4.5 | 4.7 | 4.6 | 4.6 | 3.9 | 3.5 | 5.0 | 3.7 | 3.7 |
| | Soil (Tg N) | 9.0 | 9.1 | 8.8 | 8.5 | 8.4 | 8.6 | 8.4 | 8.6 | 9.2 |
| | Lightning (Tg N) | 5.5 | 6.1 | 6.2 | 6.4 | 6.9 | 7.3 | 7.2 | 7.1 | 7.2 |
| | Isoprene (Tg C) | 493.0 | 499.3 | 471.5 | 453.6 | 435.3 | 455.4 | 466.0 | 453.3 | 467.3 |
| US | Anthropogenic NO with biofuels (Tg N) | 6.32 | 6.04 | 5.75 | 5.44 | 5.13 | 4.63 | 4.36 | 4.36 | 4.36 |
| | Biomass burning (Tg N) | 0.02 | 0.06 | 0.06 | 0.07 | 0.04 | 0.04 | 0.05 | 0.12 | 0.12 |
| | Soil (Tg N) | 0.78 | 0.86 | 1.02 | 0.92 | 0.82 | 0.79 | 0.77 | 0.95 | 1.10 |
| | Lightning (Tg N) | 0.86 | 0.86 | 0.77 | 0.75 | 1.10 | 1.13 | 1.13 | 1.28 | 1.31 |
| | Isoprene (Tg C) | 18.1 | 21.5 | 21.9 | 22.0 | 19.3 | 18.3 | 20.2 | 22.0 | 22.4 |

**Additional figures/tables added to paper:**

**Supplemental Table 3: Correlation between (1) $O_3$_Base and $O_3$_USB and (2) $O_3$_Base and $O_3$_USA on the average of $O_3$_top10obs_JJA days from 2004-2012 in each region.**

| Region | Correlation | |
|---|---|---|
| | $O_3$ Base and $O_3$ USB | $O_3$ Base and $O_3$ USA |
| New England | 0.28 | 0.64 |
| NY+NJ | 0.50 | 0.58 |
| Mid-Atlantic | 0.54 | 0.70 |
| Southeast | 0.66 | 0.59 |
| Midwest | 0.75 | 0.76 |
| South Central | 0.71 | 0.72 |
| Plains | 0.80 | 0.75 |
| Mountains + Plains | 0.95 | 0.64 |
| Pacific SW | 0.72 | 0.28 |
| Pacific NW | 0.98 | 0.05 |

[Figure]

**Figure 2: The three 4th highest days in each year (solid dots) that went into the calculation of the three-year average of the 4th highest MDA8 O3 day (hollow diamond). Error bars show the range between the highest and lowest O3_top10obs days across each 3-year span (i.e, across 30 total points) occurring between March and October in the (a) Southeast and (b) Mountains and Plains regions in the observations (black), and the O3_Base (blue) and O3_USB (red) simulations sampled on the same days as the top 10 observed values.**

[Figure]

**Supplemental Figure 2: Summary information for each region showing the three 4th highest days in each year (solid dots) that went into the calculation of the three-year average of the 4th highest MDA8 O₃ day (hollow diamond). Error bars show the range between the highest and lowest O₃_top10obs days across each 3-year span (i.e, across 30 total points) occurring between March and October in (a) New England, (b) NY+NJ, (c) Mid-Atlantic, (d) Midwest, (e) South Central, (f) Plains, (g) Pacific SW, and (h) Pacific NW. Observations are shown in black, O₃_Base is in blue, and O₃_USB is in red.**

**Table 2: Summary information for each region. The first row next to each region reports the range across 2004-2012 of the 4th highest values from each of the 9 individual years for the observations, O₃_Base, and O₃_USB. The second row reports the range across 2004-2012 of each of the 3-year averages of the 4th highest values (7 values) in each region for the observations, O₃_Base, and O₃_USB.**

| Region | Range | Obs | O$_3$_Base | O$_3$_USB |
|---|---|---|---|---|
| New England | 4th highest day | 15 | 16 | 10 |
| New England | 3-year average 4th highest day | 9 | 10 | 3 |
| New England | *Difference* | **-6** | **-6** | **-7** |
| NY+NJ | 4th highest day | 11 | 10 | 12 |
| NY+NJ | 3-year average 4th highest day | 6 | 2 | 6 |
| NY+NJ | *Difference* | **-5** | **-8** | **-6** |
| Mid-Atlantic | 4th highest day | 13 | 36 | 25 |
| Mid-Atlantic | 3-year average 4th highest day | 7 | 21 | 10 |
| Mid-Atlantic | *Difference* | **-6** | **-15** | **-15** |
| Southeast | 4th highest day | 9 | 24 | 10 |
| Southeast | 3-year average 4th highest day | 6 | 9 | 4 |
| Southeast | *Difference* | **-3** | **-15** | **-7** |
| Midwest | 4th highest day | 13 | 22 | 24 |
| Midwest | 3-year average 4th highest day | 8 | 11 | 10 |
| Midwest | *Difference* | **-6** | **-11** | **-14** |
| South Central | 4th highest day | 11 | 26 | 22 |
| South Central | 3-year average 4th highest day | 8 | 13 | 13 |
| South Central | *Difference* | **-3** | **-13** | **-9** |
| Plains | 4th highest day | 14 | 32 | 24 |
| Plains | 3-year average 4th highest day | 9 | 18 | 11 |
| Plains | *Difference* | **-5** | **-15** | **-13** |
| Mountains + Plains | 4th highest day | 9 | 23 | 20 |
| Mountains + Plains | 3-year average 4th highest day | 6 | 13 | 13 |
| Mountains + Plains | *Difference* | **-2** | **-10** | **-7** |
| Pacific SW | 4th highest day | 5 | 23 | 20 |
| Pacific SW | 3-year average 4th highest day | 3 | 5 | 5 |
| Pacific SW | *Difference* | **-2** | **-18** | **-15** |
| Pacific NW | 4th highest day | 11 | 14 | 15 |
| Pacific NW | 3-year average 4th highest day | 5 | 9 | 12 |
| Pacific NW | *Difference* | **-5** | **-5** | **-3** |

---

## Author Response (AR1)

*Average versus high surface ozone levels over the continental U.S.A.: Model bias, background influences, and interannual variability*, **Jean Guo et al., ACP, (2018)**

*Authors' response is written in bold type; Reviewer comment in normal type. Figures, including new ones added to the paper that were not in direct response to reviewer's comments are at the end.*

**Author response to Reviewer #1**

This manuscript presents an attempt to derive information about mean maximum daily 8-hour average (MDA8) $O_3$ in the United States, based on ambient measurements and using the global model GEOS-CHEM. Sensitivity simulations examine different sources that affect the 10 highest $O_3$ events and that affect the 10 days with highest model bias against observations for 2004 to 2012 for each 10 EPA regions.

General comments: The analysis is a valuable contribution to the current understanding of ground level $O_3$ and air quality standard settings. The topic itself is highly relevant and thus will be of interest to the readers of ACP. Discussion of the results and their implications is also scientifically sound and the paper includes comprehensive analyses. However, I feel that the paper tried to cover lots of information, which makes it a bit hard for the reader to follow key conclusions from this study. Thus, I recommend that the paper should be published after addressing the following comments.

**General**

Some comment about day of week effects and model biases in temperature as they relate to the questions raised in the paper seem warranted. There should be comment dramatic changes in the temperature dependence of ozone over this period coincident with the NOx changes. Those changes should have a day of week variation that might appear in the top 10 days.

**We have addressed the biases in temperature by adding a comparison to the Global Historical Climatology Network Global Historical Climatology Network (GHCN) and the Climate Anomaly Monitoring System (CAMS). See Supplemental Table 4 and the associated discussion in the text (lines 363-364):** *"The model monthly mean temperatures in the model (from the MERRA reanalysis) closely match the observed GHCN+CAMS dataset (Supplemental Table 1)."*

**The request to investigate day of week effects substantially widens the scope of the paper. The general comments suggest that the manuscript is already covering too much information. We feel that tackling day of week effects and its changes over time is a study unto itself and thus outside the scope of this particular paper.**

Specific comments: The authors use terms "Baseline $O_3$" and "U.S. background $O_3$". U.S. background $O_3$ is defined as "the $O_3$ levels that would exist in the absence of U.S. anthropogenic emissions of precursors" and Baseline $O_3$ is defined as "tropospheric $O_3$ concentrations that have a negligible influence from local anthropogenic emissions". They sound the same, don't they? If yes, please be consistent in the text.

**These definitions are not the same. Clarification has been added in lines 105-107.** *"Baseline O₃ is a measurable quantity and differs from background O₃ in that it contains some influence from U.S. anthropogenic emissions that were not recently emitted but contributed to the global background."* **We follow here the definitions of Jaffe et al., 2018 which builds on the 2009 National Academies report "Global Sources of Local Pollution", and the HTAP 2010 report (available at www.htap.org).**

Page 4, lines 106-109, Please clarify if the authors apply Schnell et al. (2014)'s interpolation procedure or they use their dataset. Schnell et al. (2014) use surface MDA8 O₃ measurements from air quality networks for 2000–2009, while this paper analyzes the data from 2004-2012.

**Jordan Schnell is a co-author and provided the dataset that we used here. The interpolation procedure for his dataset is described in his 2014 paper; he provided us with the data for the years since 2009. We have edited the citation to reflect a newer paper in which this extended dataset has been used. See lines 118-120:** *"we use an available 1° x 1° grid of surface MDA8 O₃ measurements that were interpolated from the AQS, CASTNet, and Canadian NAPS networks (Schnell and Prather, 2017)."*

A valuable addition would be a statement about the chemistry scheme applied in the version GEOS-Chem at the 2.3 section (GEOS-Chem model simulation). The authors mention issues of isoprene chemistry in last paragraph of Conclusions but a brief description or reference to the specific version of the chemistry should be presented before the last paragraph of the paper.

**On lines 158-160 we now state:** *"We use the standard v9_02 chemical mechanism which includes recycling of isoprene nitrates (Mao et al., 2013) in contrast to the mechanisms used in earlier versions of GEOS-Chem (e.g., Zhang et al., 2014 as discussed in Fiore et al., 2014)."*

The last paragraph of page 6 needs elaboration where the authors state the sensitivity simulations. The notations for all model simulations should be mentioned and the description of Table 1 should be modified so that the Table is read from top to bottom. **We have completely rewritten this section with the intent of improving clarity. See paragraph starting from line 183** *("We first perform a base simulation…")*.

Figure 3: Observed O₃ concentrations should be represented in a different color to be more visible (maybe black instead of grey) and I would also suggest to plot the curves as an average for 2004-2012 period with associated error bars. **Thanks for this suggestion. We have edited the figure to show the curves as an average for 2004-2012 period with associated error bars.**

Minor comments: The tables start from Table 2 at the manuscript and Table 1 is referenced at Page 8 for the first time. Please fix ordering of table numbers as they appear in the text. **Fixed**

Page 7, line 195: "a maximum in and" should read "a maximum in summertime and" **Thank you. Section has been edited and this sentence was removed.**

Author response to Reviewer #2

General:

The paper is very well-written and concerns a topic of considerable interest to air quality planners. However, there are some concerns about the suitability of this particular model configuration to address some of the stated objectives of the paper (lines 76-79), as discussed below. In general, the paper would be improved if there was greater clarity about the potential connections between the findings and possible configuration concerns. The value of the paper would be enhanced if the conclusions section was bolstered with a "next steps" or "considerations" sentence or two that described how such a global model-based sensitivity study could be improved in the future.

**We have attempted to strengthen the paper throughout as suggested by the reviewer. In particular, we added a sentence in the introduction, (lines 77-80) to highlight the key benefit and drawback of using a coarse resolution model:** *"Though coarse resolution global models such as GEOS-Chem will mix emissions into the same grid cell that may remain separate in the real atmosphere, a global model is necessary to quantify background $O_3$ transported intercontinentally, including that produced via oxidation of methane."*

**We added a sentence in the conclusion (lines 476-481) to emphasize a need to confirm our findings with finer scale models:** *"Future work with high-resolution models (e.g., at the regional scale, ideally with boundary conditions that include source attributions from a global model) is needed, along with observational evidence, to quantify the extent to which biogenic VOC and NOx contribute to the highest observed $O_3$ levels in the warm season. The importance of temperature sensitive sources like biogenic VOC and NOx emissions to background $O_3$ imply that in a warmer climate, these background influences on $O_3$ will play an even more important role in driving up $O_3$ levels."*

In particular, there is concern about the use of a coarse resolution model (2 x 2.5 deg) to investigate contributions of U.S. anthropogenic emissions ($O_3$_USA) given that those contributions originate at scales much smaller than the resolution of the model (i.e., point source emissions, urban area emissions). The paper acknowledges the limitations associated with the coarse modeled resolution in several places (lines 212, 242, 399). The paper may want to revisit these caveats in the conclusion and perhaps provide some thoughts on what alternate global model configurations would be better suited for an analysis of source contributions.

**Agreed. Please see above.**

Kudos to the authors for providing sufficient detail regarding the performance evaluation to allow readers to interpret the contribution findings in light of the model bias/error. However, the ozone overestimations (3-14 ppb in JJA MDA8 top 10 days by region, even worse for JJA all-day averages) suggest caution should be exercised in overinterpreting the contributions. Based on Figure 5 and the associated analyses, it appears that the model vastly overestimates ozone on hot days in the late summer, especially in the eastern U.S. (even without consideration of potential additional emissions due to increased power demand on those days). Section 3.3 briefly

summarizes potential causes for this overestimation based on similar studies, but it would be valuable if the paper provided more application-specific hypotheses for the underlying cause.

**We have added more discussion of the potential for biases in the meteorology (see responses to reviewer #3), as well as in anthropogenic NO$_x$ emissions to contribute to the summertime overestimate in the model compared to observations. As our set of sensitivity simulations identifies a potential role for biogenic VOC and soil NO$_x$ in contributing to the bias, we have added to the text some discussion calling out the need for better constraints on these biogenic emissions, though we do note that the model nevertheless shows some skill at capturing the observed year-to-year variability, which includes a correlation with O$_3$ produced from natural sources (BVOC and soil NO$_x$), which, like total O$_3$, correlate with temperature (Figure 6).**

**We now state, in lines 441-449** *"Our finding that BVOC emissions contribute to the summertime surface O$_3$ biases could reflect poor representation of the emissions (and subsequent oxidation chemistry). Earlier work has noted that MEGAN BVOC emissions are too high over California (Bash et al., 2016), Southeast Texas (Kota et al., 2015), the Ozarks in southern Missouri (Carlton and Baker, 2011), and across much of the U.S.A. (Wang et al., 2017). One recent model study uniformly reduced MEGAN isoprene emissions by 20% (Li et al., ACP 2018), but we did not apply any such scaling here. In regions that are highly NOx-sensitive, additional isoprene should not strongly influence O$_3$, as found over southeast Texas (Kota et al., 2015). While not eliminated entirely, the summertime model bias does lessen in the simulation with BVOC emissions set to zero, suggesting that the O$_3$ bias is indeed exacerbated if BVOC emissions are overestimated in the model."*

FYI, along w/ the possible causes from the Travis research, others have raised concerns about MEGAN biogenic VOC estimates (e.g., Bash et al., 2016; Carlton and Baker, 2011; Kota et al., 2015; Wang et al., 2017). **Thank you for pointing these out. We added these references (see previous response).**

One of the more noteworthy findings concerns the modeled trends over the 10-year period (e.g., lines 386-389) where the analysis appears to confirm previous findings that improving trends in U.S. air quality from emissions controls have been tempered by increases in background contributions (and increases in temperature). However, one interesting finding here that could use additional explanation is the regional breakout of this "USB vs. USA" tradeoff. Table 5 suggests that the largest increases in high JJA-day O$_3$_USB concentrations between 2004-2006 and 2010-2012 have occurred in the New England and Mid-Atlantic regions, not the western regions where USB concerns are typically greatest. More explanation of the regional differences in modeled USB trends would be beneficial (e.g., is this just an artifact of the meteorology of the two 3-year periods in these regions).

**We agree. The "trends" over such a short period are strongly influenced by fluctuations in temperature. While it may indeed be an 'artifact' of looking at such a short period, it nevertheless suggests that regionally produced background O$_3$ from temperature-sensitive emissions (BVOC and NO$_x$) may grow in importance in the coming decades in light of a**

**warming climate. We have attempted to make this clearer by adding a column to Table 4 that shows the change in temperature between these two 3-year periods in each region. We have edited the accompanying discussion to the main text:**

**Starting from line 364:** *"Table 4 shows that regions with O₃_USB increases generally experienced rising temperatures over this period, as the 2010-2012 period includes two of the warmest years on record. Figure 6 shows that O₃_NAT tracks with…"*

**In response to a comment from Reviewer 1, we have also added Supplementary Table 4 that evaluates monthly mean model temperatures with the Global Historical Climatology Network.**

Given model performance findings, would the authors see value in revising the "2-step" contribution analysis (assessing contributions on high-bias days, then assessing contributions on high/all observed days) to a "3-step" contribution where as an intermediate step you also investigated contributions on top-10 modeled days? This could be valuable presuming that the subset of days would differ from top 10 highest bias days.

**Though we did conduct some exploratory analysis using this 3-step method early on, we did not end up pursuing this method in the paper because the highest model days are less relevant to the "real world" and if this method were used throughout our paper, the number of figures would have doubled. As the paper is already lengthy, we choose to focus on the days in the observations when the O₃ NAAQS is most likely to be exceeded.**

**We have, however added text that clarifies the extent to which there is overlap between the highest 10 days in the model and the 10 days with the highest biases:** *"There is at most a 2-6 day overlap between the top 10 O₃_Base days and the top 10 most biased days in 2004-2012 across all regions, but during most years, the overlap is around 0-2 days. We restrict our analysis to examining the top 10 observed O₃ days as these days are most relevant from a policy perspective."* **(lines 137-139).**

Rather than lumping the Mount Bachelor observations (and subsequent pairs) with surface sites in Region 10, it would be interesting to see how model contributions varied as functions of model performance and observation concentration as a standalone site.

**Thanks for this suggestion. We now include a more detailed analysis of Mount Bachelor as a separate standalone section (Section 3.2) that includes new figures (Figure 3 and Supplemental Figure 4).**

**Specific:**

Line 86: "download" should be "downloaded". **FIXED**

Lines 124-127: Would be easier to read, if a new sentence was started w/ "On the days with . . .". **FIXED**

Line 146: "Avian" should be "Aviation". **FIXED**

Line 195: Is the word "summer" missing from this sentence . . . "The model, however, has a maximum in [summer] and underestimates springtime baseline $O_3$"? **Thank you. Section has been edited and this sentence was removed**

Line 205: Are Travis et al. (2016) conclusions regarding 2011 NEI relevant to a model configuration based on 2005 NEI w/ annual scalars?

**We add a comparison with Travis et al. (2016) in lines 226-230:** *"Travis et al. (2016) find that the 3.5 Tg N y-1 NEI 2011 estimate for U.S. fuel NOx emissions is too high and contributes to excessive surface $O_3$. Our simulations include even higher U.S. fuel NOx emissions of 4.4 Tg N y-1 during 2010-2012 (Supplemental Table 3), implying that some portion of the model $O_3$ bias reflects excessively high anthropogenic NOx emissions (Travis et al., 2016)."*

Line 248: Clarify that these monthly averages are MDA8 $O_3$ (not hourly)? **FIXED**

Line 367: Move mention of lack of daily variation in emissions to early section? **Done. Now at lines 179-183)**

Line 396: Same as above, maybe mention this earlier in modeling methodology section? **See lines 179-183)**

**Author response to Reviewer #3**

This paper presents a comprehensive modeling analysis of surface ozone and the various factors that contribute to its variability over the United States. By conducting multiple sensitivity simulation removing various sources for the 2004-2012 period, the authors estimate the influence of different background sources and of U.S. anthropogenic sources on mean surface $O_3$ and high $O_3$ events as a function of region, season, and year.

Two aspects of the paper that I'd like to see more discussion on are listed below:

1) The paper is very detailed with many figures and tables and is one more study on top of a rich set of published work, including by some of the co-authors. The authors often cite previous work, saying it is consistent with their results, but it would be useful to highlight what are the new key contributions from their specific analysis. What new information did the detailed modeling analysis bring to this prolific field?

**We added text with the intent of providing stronger motivation to the introduction in which we highlight the use of sensitivity simulations to help us identify which sources contribute most to the summertime bias and to the highest $O_3$ days (lines 70-87). To our knowledge, the finding that increasing $O_3$ production from temperature-sensitive biogenic emissions might be offsetting some of the gains achieved by reducing anthropogenic ozone precursor emissions is new, and potentially of growing importance as record-setting warm years have been increasing. We believe that our finding that the summertime bias is associated with regionally produced ozone – including both U.S. anthropogenic and components of U.S.**

**background – rather than transported background (either internationally or intercontinentally) is also new. We have also rewritten the conclusions to emphasize these points.**

2) There isn't much discussion on the causes of the large summer bias over the Eastern US and how this bias affects the interpretation of the results.

**To our knowledge, prior studies have not used such a broad set of sensitivity simulations to interpret which sources are contributing most in places and times when the model is most biased against observations. Section 3.3 in the submitted paper is entirely devoted to addressing this point. We thus assume that the reviewer is instead driving at the deeper question of the specific causes of the bias, beyond what we can identify cleanly with the sensitivity simulations. We have added additional discussion in response to reviewer 2 that attempts to address both the causes and how it affects the interpretation of the results.**

**Specifically, we added**

1) **A sentence in the introduction, (lines 77-80) to highlight the key benefit and drawback of using a coarse resolution model:** *"Though coarse resolution global models such as GEOS-Chem will mix emissions into the same grid cell that may remain separate in the real atmosphere, a global model is necessary to quantify background $O_3$ transported intercontinentally, including that produced via oxidation of methane."*

2) **We also added a sentence in the conclusion (lines 476-481) to emphasize a need to confirm our findings with finer scale models:** *"Future work with high-resolution models (e.g., at the regional scale, ideally with boundary conditions that include source attributions from a global model) is needed, along with observational evidence, to quantify the extent to which biogenic VOC and NOx contribute to the highest observed $O_3$ levels in the warm season. The importance of temperature sensitive sources like biogenic VOC and NOx emissions to background $O_3$ imply that in a warmer climate, these background influences on $O_3$ will play an even more important role in driving up $O_3$ levels."*

3) **We now state, in lines 441-449** *"Our finding that BVOC emissions contribute to the summertime surface $O_3$ biases could reflect poor representation of the emissions (and subsequent oxidation chemistry). Earlier work has noted that MEGAN BVOC emissions are too high over California (Bash et al., 2016), Southeast Texas (Kota et al., 2015), the Ozarks in southern Missouri (Carlton and Baker, 2011), and across much of the U.S.A. (Wang et al., 2017). One recent model study uniformly reduced MEGAN isoprene emissions by 20% (Li et al., ACP 2018), but we did not apply any such scaling here. In regions that are highly NOx-sensitive, additional isoprene should not strongly influence $O_3$, as found over southeast Texas (Kota et al., 2015). While not eliminated entirely, the summertime model bias does lessen in the simulation with BVOC emissions set to zero, suggesting that the $O_3$ bias is indeed exacerbated if BVOC emissions are overestimated in the model."*

Discussing this in more detail would strengthen the paper. The authors have one sentence addressing this by referring to the work of Travis et al. (2016) using a more recent version of the GEOS-Chem model. They mention potential errors in anthropogenic NOx emission in the NEI inventory, but Travis et al. use the NEI 2011 inventory while the authors use the NEI 2005 inventory. How different are they? If the NEI NOx inventory is indeed too high, how would that affect the calculation of $O_3$\_USA?

**We now directly compare our NOx emissions to those used in Travis et al., 2016 and include a supplementary table (see above – Supplemental Table 3) providing the NOx emissions applied in each year within the U.S.A. and globally.**

**Lines 226-230: "*Travis et al. (2016) find that the 3.5 Tg N y-1 NEI 2011 estimate for U.S. fuel NOx emissions is too high and contributes to excessive surface $O_3$. Our simulations include even higher U.S. fuel NOx emissions of 4.4 Tg N y-1 during 2010-2012 (Supplemental Table 3), implying that some portion of the model $O_3$ bias reflects excessively high anthropogenic NOx emissions (Travis et al., 2016)."***

They mention meteorological factors associated with boundary layer mixing and cloud cover which would affect the vertical distribution of $O_3$, but Travis et al. used different meteorological fields (GEOS-FP) compared to the MERRA fields used by the authors. It is unclear whether these potential explanations apply in this case. If MERRA meteorology is indeed biased, then that would certainly affect the validity of the relative influence of various sources on the "most-biased" days analysis and on the average MDA8 $O_3$ levels. A discussion of this would be valuable.

**Thanks for this suggestion. We have attempted to address this point by including more discussion of published evaluations of MERRA meteorology:**

1) **Lines 152-157: "*MERRA meteorology captures summer mean surface temperatures to within 1-2 K across U.S. regions and precipitation to within 0.5 mm d-1 except for over the Northern Great Plains where a positive bias exceeds 1 mm d-1, but the variance in summer mean precipitation is lower than observed in some regions (Bosilovich, 2013). While interannual variability in cloudiness observed at weather stations is largely captured by MERRA, the reanalysis generally underestimates cloud cover and thus overestimates observed downward surface shortwave fluxes (Free et al., 2016)."***
2) **Lines 230-232: "*The low bias in cloud cover in the MERRA meteorology and associated overestimate in downward shortwave surface radiation (Free et al., 2016) may also contribute to excessive $O_3$ production in the model."***
3) **We also added our own evaluation of surface temperature over the U.S.A. in the MERRA fields (Supplemental Table 4).**

Minor comments:

Line 154. "Anthropogenic emissions. . . are scaled each year on the basis of economic data". It would be useful to have a bit more discussion on how anthropogenic emissions are scaled over

the continental U.S. which uses 2005 as the baseline. By how much do NOx emissions change over the time period of the simulation 2004-2012.

**Supplemental Table 3 was added to provide the NOx emissions within each year, both globally and within the U.S.A. (Lines 178-183)**

Are these scaling factors taken from the NEI trends report (https://www.epa.gov/air-emissions-inventories/air-pollutant-emissions-trends-data) itself or was independent estimate done?

**The van Donkelaar et al., 2008 describes the standard GEOS-Chem emissions scaling reference. The scale factors use government statistics where available.**

**Edited this sentence (lines 169-170) to include "provided by individual countries, where available"**

Line 195. "a maximum in and underestimate springtime. . ." is "summer" missing after maximum? **Yes. Thank you**

Line 196. While the authors talk about potential causes for the springtime underestimate (stratospheric intrusions), they do not talk about the summertime overestimate, which is quite large.

**See response to general comments above and our additions above regarding anthropogenic NO$_x$ emissions and citations of prior work evaluating MERRA meteorology (temperature, precipitation and cloud cover).**

**Figures and tables edited/added in response to reviewer's comments:**

[Figure]

Figure 1: Monthly 2004-2012 average 24-hour $O_3$ concentrations at Mount Bachelor Observatory. Observations (grey) are the same in both panels. Simulations from the GEOS-Chem model are sampled in the grid cell containing Mount Bachelor at (a) 2.7 km (the height of the Mount Bachelor Observatory) and at (b) the surface: $O_3$_Base (blue), $O_3$_USB (red), $O_3$_NAT (light green), $O_3$_ICT+$CH_4$ (pink), and $O_3$_USA (dark green). The shaded range spans the highest and lowest years.

[Figure]

Supplemental Figure 1: Monthly average of observed (a) daily 24-hour and (b) MDA8 $O_3$ concentrations averaged across 2004-2012 at Mount Bachelor Observatory. Black line shows the average of each month from 2004-2012. Error bars show the standard deviation in the interannual variability in each month. Dashed lines show the concentrations for each individual year.

**Table 1: Change in MDA8 $O_3$ concentrations from 2004-2006 to 2010-2012 on $O_3$_top10obs_JJA days in the observations, $O_3$_Base, $O_3$_USB, $O_3$_USA, and temperature.**

|  | Obs | $O_3$_Base | $O_3$_USB | $O_3$_USA | Temperature (C) |
|---|---|---|---|---|---|
| New England | -6 | -4 | 6 | -10 | 2 |
| NY+NJ | -2 | -4 | 3 | -7 | 1 |
| Mid-Atlantic | 0 | -3 | 4 | -7 | 1 |
| Southeast | -4 | -5 | 2 | -7 | 1 |
| Midwest | -2 | -4 | 2 | -6 | 0 |
| South Central | -6 | -2 | 5 | -7 | 1 |
| Plains | -1 | -2 | 4 | -5 | 1 |
| Mountains + Plains | -4 | -1 | 1 | -2 | -1 |
| Pacific SW | -3 | -4 | 0 | -4 | -1 |
| Pacific NW | -7 | -5 | -4 | -1 | -1 |
| *Average* | *-3* | *-3* | *2* | *-6* | *0* |

**Supplemental Table 1: Monthly average temperature across all days in each season (average of 2004-2012) in (1) GEOS-Chem, in (2) the Global Historical Climatology Network (GHCN) and the Climate Anomaly Monitoring System (CAMS) (in degrees C), and (3) the difference between these values.**

| Region | Model Temperature (C) | | | GHCN+CAMS Temperature (C) | | | Model Temp. Bias | | |
|---|---|---|---|---|---|---|---|---|---|
|  | *MAM* | *JJA* | *SON* | *MAM* | *JJA* | *SON* | *MAM* | *JJA* | *SON* |
| New England | 7 | 19 | 11 | 8 | 20 | 11 | 0 | -1 | 1 |
| NY+NJ | 9 | 21 | 12 | 9 | 21 | 12 | 0 | 0 | 1 |
| Mid-Atlantic | 12 | 23 | 14 | 12 | 23 | 14 | 0 | 0 | 0 |
| Southeast | 17 | 26 | 18 | 17 | 26 | 18 | 0 | 0 | 0 |
| Midwest | 10 | 22 | 12 | 10 | 22 | 12 | 0 | 0 | 0 |
| South Central | 18 | 27 | 19 | 19 | 28 | 20 | -1 | -1 | -1 |
| Plains | 13 | 25 | 13 | 13 | 25 | 13 | 0 | 0 | 0 |
| Mountains + Plains | 7 | 20 | 9 | 7 | 19 | 8 | 0 | 1 | 1 |
| Pacific SW | 14 | 23 | 17 | 14 | 22 | 17 | -1 | 0 | 0 |
| Pacific NW | 8 | 17 | 10 | 8 | 17 | 9 | 0 | 0 | 1 |

**Supplemental Table 2: Global and US emissions totals for 2004-2012.**

| | Emissions | 2004 | 2005 | 2006 | 2007 | 2008 | 2009 | 2010 | 2011 | 2012 |
|---|---|---|---|---|---|---|---|---|---|---|
| Global | Anthropogenic NO with biofuels (Tg N) | 30.3 | 30.2 | 30.1 | 29.9 | 29.5 | 29.0 | 28.8 | 28.8 | 28.8 |
| | Biomass burning (Tg N) | 4.5 | 4.7 | 4.6 | 4.6 | 3.9 | 3.5 | 5.0 | 3.7 | 3.7 |
| | Soil (Tg N) | 9.0 | 9.1 | 8.8 | 8.5 | 8.4 | 8.6 | 8.4 | 8.6 | 9.2 |
| | Lightning (Tg N) | 5.5 | 6.1 | 6.2 | 6.4 | 6.9 | 7.3 | 7.2 | 7.1 | 7.2 |
| | Isoprene (Tg C) | 493.0 | 499.3 | 471.5 | 453.6 | 435.3 | 455.4 | 466.0 | 453.3 | 467.3 |
| US | Anthropogenic NO with biofuels (Tg N) | 6.32 | 6.04 | 5.75 | 5.44 | 5.13 | 4.63 | 4.36 | 4.36 | 4.36 |
| | Biomass burning (Tg N) | 0.02 | 0.06 | 0.06 | 0.07 | 0.04 | 0.04 | 0.05 | 0.12 | 0.12 |
| | Soil (Tg N) | 0.78 | 0.86 | 1.02 | 0.92 | 0.82 | 0.79 | 0.77 | 0.95 | 1.10 |
| | Lightning (Tg N) | 0.86 | 0.86 | 0.77 | 0.75 | 1.10 | 1.13 | 1.13 | 1.28 | 1.31 |
| | Isoprene (Tg C) | 18.1 | 21.5 | 21.9 | 22.0 | 19.3 | 18.3 | 20.2 | 22.0 | 22.4 |

**Additional figures/tables added to paper:**

**Supplemental Table 3: Correlation between (1) $O_3$_Base and $O_3$_USB and (2) $O_3$_Base and $O_3$_USA on the average of $O_3$_top10obs_JJA days from 2004-2012 in each region.**

| Region | Correlation | |
|---|---|---|
| | $O_3$ Base and $O_3$ USB | $O_3$ Base and $O_3$ USA |
| New England | 0.28 | 0.64 |
| NY+NJ | 0.50 | 0.58 |
| Mid-Atlantic | 0.54 | 0.70 |
| Southeast | 0.66 | 0.59 |
| Midwest | 0.75 | 0.76 |
| South Central | 0.71 | 0.72 |
| Plains | 0.80 | 0.75 |
| Mountains + Plains | 0.95 | 0.64 |
| Pacific SW | 0.72 | 0.28 |
| Pacific NW | 0.98 | 0.05 |

[Figure]

**Figure 2: The three 4th highest days in each year (solid dots) that went into the calculation of the three-year average of the 4th highest MDA8 O₃ day (hollow diamond). Error bars show the range between the highest and lowest O₃_top10obs days across each 3-year span (i.e, across 30 total points) occurring between March and October in the (a) Southeast and (b) Mountains and Plains regions in the observations (black), and the O₃_Base (blue) and O₃_USB (red) simulations sampled on the same days as the top 10 observed values.**

[Figure]

**Supplemental Figure 2: Summary information for each region showing the three 4th highest days in each year (solid dots) that went into the calculation of the three-year average of the 4th highest MDA8 O$_3$ day (hollow diamond). Error bars show the range between the highest and lowest O$_3$_top10obs days across each 3-year span (i.e, across 30 total points) occurring between March and October in (a) New England, (b) NY+NJ, (c) Mid-Atlantic, (d) Midwest, (e) South Central, (f) Plains, (g) Pacific SW, and (h) Pacific NW. Observations are shown in black, O$_3$_Base is in blue, and O$_3$_USB is in red.**

[revised manuscript text omitted]
_3$ ~~values. Figure 3 compares modeled and observed monthly mean $O_3$ at Mount Bachelor. The observations peak in springtime and then fall in the summer months. The model, however, has a maximum in and underestimates springtime baseline $O_3$. We infer, consistent with our analysis below, that the model does not resolve springtime high $O_3$ events, possibly reflecting an underestimate of stratospheric influences (see Fiore et al., 2014; Zhang et al., 2011; 2014). The model indicates that $O_3$_USB dominates $O_3$_Base (Figure 3). Even at this baseline site, however, the model indicates that U.S. anthropogenic emissions enhance monthly mean $O_3$ by at least a few ppb (estimated as the difference between $O_3$_Base and $O_3$_USB). .~~ In Supplemental Figure 4, we compare the observed 24-hour and MDA8 $O_3$ concentrations at MBO for 2004-2012. The observed $O_3$ concentrations vary from year to year, and by definition, MDA8 $O_3$ is a few ppb higher than the 24-hour mean mixing ratios. However, the seasonal pattern is similar across both metrics, with a springtime peak, maximum in April, and a secondary summertime peak in July.

[revised manuscript text omitted]

To explore possible drivers of model biases across the different seasons, we evaluate the timing of the highest ten events across each year in the $O_3\_Base$, $O_3\_USB$, and $O_3\_noBVOC$ (BVOCs shut off) simulations for each region (900 events). We bin these 900 events by month and calculate the percentage of the total events that fall within each month. Note that all the top ten days fall between March and October. The standard model ($O_3\_Base$) underestimates the occurrence of high events early in the $O_3$ season (March-June) and overestimates them later in the season (July-September) (Figure 7). While the model indicates that most top ten $O_3$ days fall between July-August (35% each), the observations show that May through August each contain around 15-25% with the maximum in June at 25%. When we examine the highest ten $O_3$ events in the $O_3\_USB$ case (U.S. anthropogenic emissions shut off), we see 5-10% fewer top ten events in July and August (27% in July and 28% in August), suggesting that $O_3\_USA$ is contributing most to the temporal shift (and general summertime overestimate) relative to the observations. The $O_3\_USB$ case does capture some early spring events in April (5%) and May (10%), though still fewer than observed (12% and 17% respectively). In the $O_3\_noBVOC$ case, there are

5-10% more events during April and May than in the O₃_Base case, but the shortage of high spring O₃ events remains. The lack of high events in spring may stem from the springtime underestimate in this model, particularly at high altitude sites (e.g., Figure 5; see also Figures 4 and 6 of Fiore et al. (2014)), and may reflect poor representation of stratospheric O₃ intrusions at the coarse resolution of the CTM (Zhang et al., 2014). The summertime overestimate of high O₃ events is less pronounced in the O₃_noBVOC case than in the O₃_Base case, implying that BVOCs are also contributing to the misplaced seasonal timing of the highest events, either through excessive O₃ production or a missing coincident sink.
[revised manuscript text omitted]
 ~~covered by the 10 highest events (Figure 13, Supplemental Figure 12, Supplemental Figure 13). The annual range in the model ($O_3$_Base) sampled on $O_3$_top10obs days tends to be wider than the observed range (except for a few years in New England and NY+NJ) by as little as a few ppb to as much as 20 ppb. This modeled range overestimate lessens when averaged over three years (Figure 13a, b versus Figure 13c, d). We also include in Figure 12 (and Supplemental Figures 12 and 13) the range of the $O_3$_USB onWhile the three-year averaging period reduces the range in $O_3$_USB on the highest days, variability remains, and over the Mountains and Plains regions thissource influencing these(Figure 13b, d).background.~~, and in the WUS, this interannual variability tends to reflect variations in $O_3$_USB.

**6  Discussion and Conclusions**

As air quality controls decrease U.S. anthropogenic precursor emissions to $O_3$, the relative importance of the background influence on total surface $O_3$ increases. We use $O_3$ MDA8 concentrations spanning 2004-2012 from the EPA AQS, CASTNet, and Mount Bachelor Observatory sites, and  sensitivity simulations from the global GEOS-Chem 3D chemistry transport model to estimate the influence from various individual background sources on $O_3$ in each of the ten EPA regions in the continental U.S.A.  The global scale of the GEOS-Chem model allows us to quantify intercontinental transport (including global methane) in addition to regional natural

525 and anthropogenic sources of $O_3$. The sensitivity simulations span nine years, allowing us to examine the role of these sources in contributing to interannual variability. Our analysis contrasts average- and high-$O_3$ days.

Correlations between monthly averages across 2004-2012 show that the model captures monthly variations from year-to-year, especially during summer (JJA). The model shows substantial variability in simulated U.S. background $O_3$ concentrations from year-to-year, on the order of 10-20 ppb between 2004-2012 in summer (Figure

530 7). We find that the extent to which the current three-year averaging period for assessing compliance with the National Ambient Air Quality Standard for $O_3$ succeeds in smoothing out interannual variability depends on the range in consecutive years, and thus varies by region and time period, but is generally not long enough to completely eliminate the interannual variability in background $O_3$ (Figure 9).

We find substantial biases in the severity (+0-19 ppb in maximum daily 8-hour average (MDA8) $O_3$) and

535 timing of high-$O_3$ events in the model. The model underestimates the frequency of high events in spring, possibly associated with stratospheric intrusions (Fiore et al., 2014; Zhang et al., 2011; 2014). Future efforts would benefit from quantifying the stratospheric (as well as Asian) influence alongside the other background sources we consider. We find a stronger influence of U.S. anthropogenic emissions on regionally averaged MDA8 $O_3$ (up to 30 ppb) from BVOCs (up to 15 ppb) and soil $NO_x$ (up to 10 ppb) on the ten most biased days as compared to average days. We

540 conclude that regional production of $O_3$ is driving the pervasive high positive model bias in summer, as opposed to transported background, although our sensitivity simulations do not allow us to rule out the possibility of a coincident missing sink.

Our finding that BVOC emissions contribute to the summertime surface $O_3$ biases could reflect poor representation of the emissions (and subsequent oxidation chemistry). Earlier work has noted that MEGAN BVOC

545 emissions are too high over California (Bash et al., 2016), Southeast Texas (Kota et al., 2015), the Ozarks in southern Missouri (Carlton and Baker, 2011), and across much of the U.S.A. (Wang et al., 2017). One recent model study uniformly reduced MEGAN isoprene emissions by 20% (Li et al., ACP 2018), but we did not apply any such scaling here. In regions that are highly NOx-sensitive, additional isoprene should not strongly influence $O_3$, as found over southeast Texas (Kota et al., 2015). While not eliminated entirely, the summertime model bias does lessen in

550 the simulation with BVOC emissions set to zero, suggesting that the $O_3$ bias is indeed exacerbated if BVOC emissions are overestimated in the model.

 The model underestimates the frequency of high events in spring. The ten most biased days (considering regionally-averaged MDA8 O₃ values in each of the ten EPA regions) tend to be around 10°C warmer than average days. Our model does not include daily variations in U.S. anthropogenic emissions associated with higher electricity demand on hotter days (e.g., Abel et al., 2017), but we still find that the influence of U.S. anthropogenic emissions on regionally averaged MDA8 O₃ is up to 30 ppb higher on the ten most biased days as compared to average days. The model does include daily variability in temperature-sensitive biogenic emissions and simulates higher than average O₃ 
[revised manuscript text omitted]

[Figure]

815    **Figure 1: Map of the states falling within each EPA region in the continental United States (adapted from *U.S. Environmental Protection Agency*, 2012).**

*Table 2*: The number of observational sites that fall within each EPA region for EPA AQS and CASTNet. (*) We include data from the Mount Bachelor Observatory in the Pacific Northwest region.

| Region | EPA AQS | CASTNet | *Total* |
|---|---|---|---|
| 1. New England | 82 | 7 | 89 |
| 2. New York + New Jersey (NY+NJ) | 61 | 7 | 68 |
| 3. Mid-Atlantic | 138 | 14 | 152 |
| 4. Southeast | 309 | 24 | 333 |
| 5. Midwest | 255 | 18 | 273 |
| 6. South Central | 202 | 5 | 207 |
| 7. Plains | 71 | 2 | 73 |
| 8. Mountains and Plains | 153 | 12 | 165 |
| 9. Pacific Southwest | 325 | 14 | 339 |
| 10. Pacific Northwest | 48 | 6* | 54 |
| *Total* | *1644* | *109* | *1753* |

820

[Figure]

 **Figure 1~~: Frequency distribution of MDA8 O₃ values across all sites in the United States from Jan-Dec (365 or 366 days per year) from 2004-2012 in the (a) Schnell dataset (2014) interpolated to 2° by 2.5°, (b) at individual observational sites, and c) on the 10 most biased days. Concentrations for each day are obtained by averaging across all sites in a region. The model bias is defined as O₃_Base minus observed. The total number of points consists of 9 years x 10 days x 10 regions. The observations are in shown in blue and GEOS-Chem is in orange. The line drawn at 70 ppb in panels (a) and (b) denotes the~~**

 **: Frequency distribution of regionally averaged U.S. MDA8 O₃ values from 2004-**

**2012 in the (a) Schnell and Prather (2017) dataset interpolated to 2° by 2.5° and (b) at individual observational sites prior to averaging over each of the 10 EPA regions (total number of points is 9 years x 365 or 366 days x 10 regions) in the observations (blue) and the GEOS-Chem model (orange). c) As in panel (b) but selecting for the 10 most biased days in each region (total number of points is 9 years x 10 days x 10 regions). The line drawn at 70 ppb in panels (a) and (b) is the current O₃ NAAQS level.**

[Figure]

[Figure]

**Figure 2**

**: Average MDA8 O₃ model bias (O₃_Base – observed) on all days in (a)**  **MAM, (b) JJA, and (c) SON versus on the (d) O₃_top10obs_MAM, (e) O₃_top10obs_JJA, and (f) O₃_top10obs_SON days at each observational site averaged across 2004-2012.**

*Figure 4:*

[Figure]

845

[Figure]

**Figure 3: Monthly 2004-2012 average 24-hour O₃ concentrations at Mount Bachelor Observatory. Observations (grey) are the same in both panels. Simulations from the GEOS-Chem model are sampled in the grid cell containing Mount Bachelor at (a) 2.7 km (the height of the Mount Bachelor Observatory) and at (b) the surface: O₃_Base (blue), O₃_USB (red), O₃_NAT (light green), O₃_ICT+CH4 (pink), and O₃_USA (dark green). The shaded range spans the highest and lowest years.** ÷
850 Percent of total top 10 most biased days from Jan-Dec (9 years x 10 days x 10 regions) that fell within each month in the United States. All the most biased days fell between Mar-Oct.

[Figure]

[Figure]

**Figure 4**: : Multi-year (2004-2012) Mar-Oct average temperature and MDA8 O₃ source contributions estimated with the GEOS-Chem model in the (a) Southeast and (b) Mountain and Plains regions on the 10 most biased days (blue) versus averaged across all days ( yellow). Note that the two  are on  different scales.

860

[Figure]

Figure 7: Percent of total top ten days (9 years x 10 days x 10 regions) from Jan-Dec (365 or 366 days) in the observations, O₃_Base, O₃_USB, and O₃_noBVOC that fell within each month for all sites across the U.S.A. All the top ten days for each simulation fell between Mar-Oct.

865

[Figure]

Figure 8: Correlation between 2004-2012 year-to-year monthly MDA8 O₃ averages for May, July, and September in the observation and in the model (O₃_Base).

[revised manuscript text omitted]

**Supplemental Figures**

Supplemental Table 1: The number of observational sites that fall within each EPA region for EPA AQS and CASTNet. (*) We include data from the Mount Bachelor Observatory in the Pacific Northwest region.

925

| Region | EPA AQS | CASTNet | Total |
|---|---|---|---|
| 1. New England | 82 | 7 | 89 |
| 2. New York + New Jersey (NY+NJ) | 61 | 7 | 68 |
| 3. Mid-Atlantic | 138 | 14 | 152 |
| 4. Southeast | 309 | 24 | 333 |
| 5. Midwest | 255 | 18 | 273 |
| 6. South Central | 202 | 5 | 207 |
| 7. Plains | 71 | 2 | 73 |
| 8. Mountains and Plains | 153 | 12 | 165 |
| 9. Pacific Southwest | 325 | 14 | 339 |
| 10. Pacific Northwest | 48 | 6* | 54 |
| Total | 1644 | 109 | 1753 |

[Figure]

Supplemental Figure 1: Map of the states falling within each EPA region in the continental United States (adapted from U.S. Environmental Protection Agency, 2012).

930

**Supplemental Table 2: Number of EPA AQS sites collecting MDA8 O₃ data during each year from 2004-2012.**

| Number of EPA AQS Sites | |
|---|---|
| **2004** | 1219 |
| **2005** | 1207 |
| **2006** | 1211 |
| **2007** | 1237 |
| **2008** | 1241 |
| **2009** | 1251 |
| **2010** | 1280 |
| **2011** | 1333 |
| **2012** | 1315 |

**Supplemental Table 3: Global and US emissions totals for 2004-2012.**

| | Emissions | 2004 | 2005 | 2006 | 2007 | 2008 | 2009 | 2010 | 2011 | 2012 |
|---|---|---|---|---|---|---|---|---|---|---|
| **Global** | Anthropogenic NO with biofuels (Tg N) | 30.3 | 30.2 | 30.1 | 29.9 | 29.5 | 29.0 | 28.8 | 28.8 | 28.8 |
| | Biomass burning (Tg N) | 4.5 | 4.7 | 4.6 | 4.6 | 3.9 | 3.5 | 5.0 | 3.7 | 3.7 |
| | Soil (Tg N) | 9.0 | 9.1 | 8.8 | 8.5 | 8.4 | 8.6 | 8.4 | 8.6 | 9.2 |
| | Lightning (Tg N) | 5.5 | 6.1 | 6.2 | 6.4 | 6.9 | 7.3 | 7.2 | 7.1 | 7.2 |
| | Isoprene (Tg C) | 493.0 | 499.3 | 471.5 | 453.6 | 435.3 | 455.4 | 466.0 | 453.3 | 467.3 |
| **US** | Anthropogenic NO with biofuels (Tg N) | 6.32 | 6.04 | 5.75 | 5.44 | 5.13 | 4.63 | 4.36 | 4.36 | 4.36 |
| | Biomass burning (Tg N) | 0.02 | 0.06 | 0.06 | 0.07 | 0.04 | 0.04 | 0.05 | 0.12 | 0.12 |
| | Soil (Tg N) | 0.78 | 0.86 | 1.02 | 0.92 | 0.82 | 0.79 | 0.77 | 0.95 | 1.10 |
| | Lightning (Tg N) | 0.86 | 0.86 | 0.77 | 0.75 | 1.10 | 1.13 | 1.13 | 1.28 | 1.31 |
| | Isoprene (Tg C) | 18.1 | 21.5 | 21.9 | 22.0 | 19.3 | 18.3 | 20.2 | 22.0 | 22.4 |

935

[Figure]

**Supplemental Figure 2: Average model bias (model – observed) on the O₃_top10obs days during (a) 2004, (b) 2005, (c) 2006, (d) 2007, (e) 2008, (f) 2009, (g) 2010, (h) 2011, and (i) 2012.**

940

[Figure]

**Supplemental Figure 3: Model bias (model – observed) on the 4th highest MDA8 O₃ day at each observational site averaged for each three-year span. (a) 2004-2006, (b) 2005-2007, (c) 2006-2008, (d) 2007-2009, (e) 2008-2010, (f) 2009-2011, and (g) 2010-2012.**

[Figure]

[Figure]

[Figure]

[Figure]

955

[Figure]

**Supplemental Figure 4: Monthly average of observed (a) daily 24-hour and (b) MDA8 O₃ concentrations averaged across 2004-2012 at Mount Bachelor Observatory. Black line shows the average of each month from 2004-2012. Error bars show the standard deviation in the interannual variability in each month. Dashed lines show the concentrations for each individual year. : Average influence of each sensitivity simulation on MDA8 O3 in each region on the 10 most biased days from Jan-Dec (red) versus averaged across all days (blue). Red circles show the average model bias (O3_Base – observations) on the top 10 model bias days. Blue circles show the model bias averaged across all days. The circles do not vary between subplots. Note that O3_USB and O3_USA are on a different scale than the other plots.**

960

[Figure]

965

**Supplemental Figure 5**: Percent of total top 10 most biased days from Jan-Dec (9 years x 10 days x 10 regions) that fell within each month in the United States. All the most biased days fell between Mar-Oct.

Southeast

[Figure]

Mountain and Plains

970

On highest 10 model bias days
Average across all days
Model bias on highest model bias days
Model bias across all days

975

[Figure]

New England

NY+NJ

On highest 10 model bias days
Average across all days
Model bias on highest model bias days
Model bias across all days

e) Mid-Atlantic

f) Midwest

On highest 10 model bias days
Average across all days
Model bias on highest model bias days
Model bias across all days

South Central

[Figure]

[Figure]

[Figure]

**Supplemental Figure 6: Average influence of each sensitivity simulation on MDA8 O₃ in each region on the 10 most biased days from Mar-Oct (red) versus averaged across all days (blue). Red circles show the average model bias (O₃ Base – observations) on the top 10 model bias days. Blue circles show the model bias averaged across all days. The circles do not vary between subplots.**

[Figure]

**Supplemental Figure 7: Percent of total top ten days (9 years x 10 days x 10 regions) from Jan-Dec (365 or 366 days) in the observations, O$_3$_Base, O$_3$_USB, and O$_3$_noBVOC that fell within each month for all sites across the U.S.A. All the top ten days for each simulation fell between Mar-Oct.**

[Figure]

**Supplemental Figure 8: Correlation between 2004-2012 year-to-year monthly averages for MDA8 O$_3$ in the observation and in the model (O$_3$_Base) for each individual month.**

[Figure]

1000 **Supplemental Figure 9: Average 2004-2012 influence of each sensitivity simulation to O₃_Base in (a) New England, (b) NY+NJ, (c) Mid-Atlantic, (d) Midwest, (e) South Central, (f) Plains, (g) Pacific SW, and (h) Pacific NW on the MDA8 O₃_top10obs_JJA days (red) versus averaged across all days (blue). Error bars show the average concentration on the lowest versus highest year for each sensitivity simulation in each region.**

[Figure]

1005

**Supplemental Figure 10: Average yearly MDA8 $O_3$_top10obs_JJA concentrations for observations (divided by 2 to fit on the same axes; blue dashed line), $O_3$_Base (divided by 2; blue solid line), $O_3$_USB (red), $O_3$_USA (black), $O_3$_NAT (green) MDA8, and temperature (in degrees C; light blue) sampled on the $O_3$_top10obs days in (a) New England, (b) NY+NJ, (c) Mid-Atlantic, (d) Midwest, (e) South Central, (f) Plains, (g) Pacific SW, and (h) Pacific NW.**

1010

**Supplemental Table 4: Monthly average temperature across all days in each season (average of 2004-2012) in (1) GEOS-Chem, in (2) the Global Historical Climatology Network (GHCN) and the Climate Anomaly Monitoring System (CAMS) (in degrees C), and (3) the difference between these values.**

| | Model Temperature (C) | | | GHCN+CAMS Temperature (C) | | | Model Temp. Bias | | |
|---|---|---|---|---|---|---|---|---|---|
| Region | MAM | JJA | SON | MAM | JJA | SON | MAM | JJA | SON |
| New England | 7 | 19 | 11 | 8 | 20 | 11 | 0 | -1 | 1 |
| NY+NJ | 9 | 21 | 12 | 9 | 21 | 12 | 0 | 0 | 1 |
| Mid-Atlantic | 12 | 23 | 14 | 12 | 23 | 14 | 0 | 0 | 0 |
| Southeast | 17 | 26 | 18 | 17 | 26 | 18 | 0 | 0 | 0 |
| Midwest | 10 | 22 | 12 | 10 | 22 | 12 | 0 | 0 | 0 |
| South Central | 18 | 27 | 19 | 19 | 28 | 20 | -1 | -1 | -1 |
| Plains | 13 | 25 | 13 | 13 | 25 | 13 | 0 | 0 | 0 |
| Mountains + Plains | 7 | 20 | 9 | 7 | 19 | 8 | 0 | 1 | 1 |
| Pacific SW | 14 | 23 | 17 | 14 | 22 | 17 | -1 | 0 | 0 |
| Pacific NW | 8 | 17 | 10 | 8 | 17 | 9 | 0 | 0 | 1 |

1015

**Supplemental Table 5: Correlation between (1) $O_3$\_Base and $O_3$\_USB and (2) $O_3$\_Base and $O_3$\_USA on the average of $O_3$\_top10obs\_JJA days from 2004-2012 in each region.**

| | Correlation | |
|---|---|---|
| Region | $O_3$\_Base and $O_3$\_USB | $O_3$\_Base and $O_3$\_USA |
| New England | 0.28 | 0.64 |
| NY+NJ | 0.50 | 0.58 |
| Mid-Atlantic | 0.54 | 0.70 |
| Southeast | 0.66 | 0.59 |
| Midwest | 0.75 | 0.76 |
| South Central | 0.71 | 0.72 |
| Plains | 0.80 | 0.75 |
| Mountains + Plains | 0.95 | 0.64 |
| Pacific SW | 0.72 | 0.28 |
| Pacific NW | 0.98 | 0.05 |

[Figure]

**1020**

**Supplemental Figure 11: Monthly average MDA8 O$_3$_USB concentrations in (a) New England, (b) NY+NJ, (c) Mid-Atlantic, (d) Midwest, (e) South Central, (f) Plains, (g) Pacific SW, and (h) Pacific NW.**

[Figure]

a) O$_3$_BVOC

b ) O$_3$_BVOC

c) O$_3$_BVOC

d) O$_3$_BVOC

1025

e) O$_3$_BVOC

f ) O$_3$_BVOC

g) O$_3$_BVOC

h) O$_3$_BVOC

[Figure]

[Figure]

**Supplemental Figure 12**: Monthly average MDA8 O₃_BVOC concentrations in (a) New England, (b) NY+NJ, (c) Mid-Atlantic, (d) Midwest, (e) South Central, (f) Plains, (g) Pacific SW, and (h) Pacific NW.

1030

[Figure]

**Supplemental Figure** 13: Monthly average MDA8 $O_3$\_$SNO_x$ concentrations in (a) New England, (b) NY+NJ, (c) Mid-Atlantic, (d) Midwest, (e) South Central, (f) Plains, (g) Pacific SW, and (h) Pacific NW.

1035

1040

[Figure]

**Supplemental Figure 14: Anomaly on the MDA8 O₃_top10obs_JJA days of each sensitivity simulation relative to the 2004-2012 average in (a) New England, (b) NY+NJ, (c) Mid-Atlantic, (d) Midwest, (e) South Central, (f) Plains, (g) Pacific SW, and (h) Pacific NW. Each panel shows the anomaly from observations, O₃_Base, O₃_USB, O₃_USA, and temperature (in degrees C).**

[Figure]

1055

**Supplemental Figure 15: Anomaly on the O₃_top10obs_JJA days for each sensitivity simulation relative to the 2004-2012 average in (a) New England, (b) NY+NJ, (c) Mid-Atlantic, (d) Midwest, (e) South Central, (f) Plains, (g) Pacific SW, and (h) Pacific NW. Each panel shows the anomaly from O₃_BVOC, O₃_SNOₓ, O₃_NALNOₓ, O₃_BB, O₃_ICT+CH₄, and O₃_CA+MX.**

[Figure]

1060

[Figure]

[Figure]

**Supplemental Figure 16: Summary information for each region showing the three 4th highest days in each year (solid dots) that went into the calculation of the three-year average of the 4th highest MDA8 O₃ day (hollow diamond). Error bars show the range between the highest and lowest O₃_top10obs days across each 3-year span (i.e, across 30 total points) occurring between March and October in (a) New England, (b) NY+NJ, (c) Mid-Atlantic, (d) Midwest, (e) South Central, (f) Plains, (g) Pacific SW, and (h) Pacific NW. Observations are shown in black, O₃_Base is in blue, and O₃_USB is in red.** regions. (e), (d), (e), (f) show the range of the O₃_top10obs days after averaging over three consecutive years. The solid dots show the 4th highest MDA8 O₃ day for each simulation (a, b) and the annual 4th highest MDA8 O₃ day averaged over three consecutive years.: Range in magnitude of the ten highest MDA8 O₃ values for each year shown as vertical lines in the observations (black), O₃_Base (blue), and O₃_USB (red) in (a) New England, (b) NY+NJ, (c) Mid-Atlantic, (d) Midwest, (e) South Central, (f) Plains, (g) Pacific SW, and (h) Pacific NW. a), (b), (e), (f) show the range on of O₃_top10obs days during

each year between 2004-2012. (c), (d), (e), (f) show the range of the $O_3$_top10obs days for each year. The solid dots show the 4th highest MDA8 $O_3$ day for each simulation (a), (b) and the annual 4th highest MDA8 $O_3$ day.

Supplemental Figure 13: Range in magnitude of the ten highest MDA8 $O_3$ values after averaging over 3 consecutive years in the observations (black), $O_3$_Base (blue), and $O_3$_USB (red)

1080